# EDGE OF STOCHASTIC STABILITY:
# REVISITING THE EDGE OF STABILITY FOR SGD

## ABSTRACT

Recent findings by Cohen et al. (2021) demonstrate that when training neural networks with full-batch gradient descent with step size $\eta$, the largest eigenvalue $\lambda_{\max}$ of the full-batch Hessian consistently stabilizes around $2/\eta$. These results have significant implications for convergence and generalization. This, however, is not the case of mini-batch stochastic gradient descent (SGD), limiting the broader applicability of its consequences. We show that SGD trains in a different regime we term Edge of Stochastic Stability (EoSS). In this regime, what stabilizes at $2/\eta$ is *Batch Sharpness*: the expected directional curvature of mini-batch Hessians along their corresponding stochastic gradients. As a consequence, $\lambda_{\max}$—which is generally smaller than *Batch Sharpness*—is suppressed, aligning with the long-standing empirical observation that smaller batches and larger step sizes favor flatter minima. We further discuss implications for mathematical modeling of SGD trajectories.

## 1 INTRODUCTION

The choice of training algorithm is a key ingredient in the deep learning recipe. Extensive evidence, e.g. (Keskar et al., 2016), indeed shows that performance consistently depends on the optimizer and hyperparameters. What machinery induces this optimizer-dependence is a central question of theory of deep learning.

Cohen et al. (2021; 2024) answered this question for Gradient Descent (GD): it optimizes neural networks in a regime of instability, they termed Edge of Stability (EoS). With a constant step size $\eta$, the highest eigenvalue of the Hessian of the full-batch loss—denoted here as $\lambda_{\max}$—grows until $2/\eta$ and hovers right above, subject to small oscillations (Cohen et al., 2021; 2022; Jastrzębski et al., 2019; 2020; Xing et al., 2018). Although, classical convex optimization theory call this step size "too large", the loss continues to decrease. These works established a number of surprising facts: **(1)** that we require an optimization theory which works in more general scenarios then the classical $\eta < 2/\lambda_{\max}$; **(2)** that NN training happens in a special regime of instability, establishing what the source of it is; **(3)** how geometry of the local landscape around the solution found depends on the choice of hyperparameters.

**Our finding: EoSS.** While real-world training is almost always *mini-batch*—given the large amounts of data—existing EoS analyses **explicitly** do not apply to this case: no curvature-type quantities, such as $\lambda_{\max}$, are known to similarly affect SGD while training neural networks. We bridge this gap by establishing that:

> **Mini-batch SGD trains in a regime of instability akin to EoS** which we term Edge of Stochastic Stability (EoSS). Precisely, *Batch Sharpness*, our notion of curvature,
>
> $$\textit{Batch Sharpness}\,(\theta) \;:=\; \mathbb{E}_{B \sim \mathcal{P}_b}\left[\frac{\nabla L_B(\theta)^\top\,\mathcal{H}(L_B)\,\nabla L_B(\theta)}{\|\nabla L_B(\theta)\|^2}\right], \qquad \text{with } L_B \text{ being loss on the batch } B \text{ sampled from } \mathcal{P}_b. \; \mathcal{H}(\cdot) \text{ Hessian matrix.}$$
>
> hovers around $2/\eta$ and implicitly functions as sharpness for SGD. This implies that:
> **stability for SGD is stability on the mini-batch landscape**

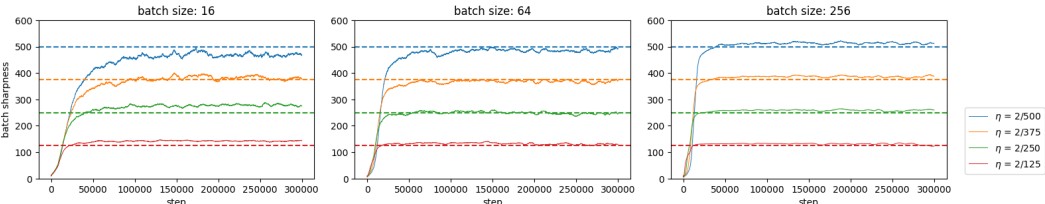

Figure 1: **SGD at EoSS under different step sizes and batch sizes.** MLP on an 8k subset of CIFAR-10 with step size $\eta > 0$. *Batch Sharpness* stabilizes at the $2/\eta$ threshold across varying batch sizes and step sizes.

**Organization and Contributions.** Section 2 reviews related work and outlines the key open questions we tackle. Oscillations are central to these phenomena, as a necessary step, in Section 3 we distinguish SGD oscillations between noise-driven (as in Robbins-Monro–type of stochastic optimization when the step size is kept fixed) and curvature-driven—which are the ones we are interested in. In Section 4, we introduce, properly characterize, and empirically validate the phenomenon of Edge of Stochastic Stability. In Section 5 we give a mathematical treatment of SGD stability. Finally, our results are yet another proof of the fact that the dynamics of noise-injected GD or SDEs and the dynamics of mini-batch SGD are qualitatively different and studying the firsts could be misleading for inducing properties of the second. We discuss this implication in Section 6.

Throughout the rest of this paper $B \subset \mathcal{D}$ denotes a random mini-batch of size $b$ drawn from a fixed sampling distribution $\mathcal{P}_b$. For model parameters $\theta \in \mathbb{R}^d$ let $L_B(\theta) = \frac{1}{b}\sum_{(x_i,y_i) \in B} \ell\big(f_\theta(x_i), y_i\big)$, $L(\theta) = \mathbb{E}_{B \sim \mathcal{P}_b}\big[L_B(\theta)\big]$ be the mini-batch and full-batch losses, respectively, where $\ell$ is the loss function. Write $\mathcal{H}(L_B) = \nabla_\theta^2 L_B(\theta)$ and $\mathcal{H}(L)$ or $\mathcal{H}$ the full-batch Hessian $\nabla_\theta^2 L(\theta)$.

## 2 RELATED WORK

**Progressive sharpening.** Early studies observed that the local shape of the loss landscape changes rapidly at the beginning of the training, by means of growth of different estimators of the curvature (Keskar et al., 2016; Jastrzębski et al., 2019; LeCun et al., 2012; Achille et al., 2017; Jastrzębski et al., 2018; Fort & Ganguli, 2019; Sagun et al., 2016; Fort & Scherlis, 2019). Subsequently, Jastrzębski et al. (2019; 2020) and Cohen et al. (2021) precisely characterized this behavior, demonstrating a steady rise in $\lambda_{\max}$ along GD and SGD trajectories, typically following a brief initial decline. This phenomenon was termed *progressive sharpening* by Cohen et al. (2021).

**Full-batch edge of stability.** Prior research (Goodfellow et al., 2016; Li et al., 2019; Jiang et al., 2019; Lewkowycz et al., 2020) found that large initial learning rates often enhance generalization despite delaying initial loss reduction. Jastrzębski et al. (2020) attributed this effect to a phase transition, termed the break-even point, marking the end of progressive sharpening. Unlike progressive sharpening, this phenomenon is considered to result from algorithmic instability rather than inherent landscape properties. Indeed, Jastrzębski et al. (2019; 2020); Cohen et al. (2021; 2022) demonstrated that this phase transition comes at different points for different algorithms on the same landscapes. Cohen et al. (2021; 2022) later showed that it comes at the instability thresholds, in the case of full-batch optimization algorithms. Precisely, GD and full-batch Adam train in the EoS oscillatory regime (Cohen et al., 2021; 2022), where the $\lambda_{\max}$ stabilizes and oscillates around a characteristic value. The name is due to the fact that, in the case of full-batch GD, the $\lambda_{\max}$ hovers at $2/\eta$ which is the stability threshold for optimizing quadratics. Observations from Cohen et al. (2021; 2022) indicate that, under mean square error (MSE), the bulk of training dynamics occur within this regime, effectively determining $\lambda_{\max}$ of the final solution. Lee & Jang (2023) explained why in this regime $\lambda_{\max}$ often slightly exceeds $2/\eta$: this deviation arises primarily from nonlinearity of the loss gradient, which shifts the required value depending on higher-order derivatives, and the EoS being governed by the Hessian along the gradient direction, rather than $\lambda_{\max}$ alone. A growing body of research analyzes the surprising mechanism underlying EoS dynamics observed during training with GD. Classically, when gradients depend linearly on parameters, divergence occurs locally if $\eta > \frac{2}{\lambda_{\max}}$, as illustrated by one-dimensional quadratic models (Cohen et al., 2021). In contrast, neural networks often converge despite violating this classical stability condition, presumably due

to the problem's non-standard geometry. Damian et al. (2023) propose an explanation under some, empirically tested, assumptions of alignment of third derivatives and gradients.

**Existing EoS work is limited to full-batch methods.** While the empirical behavior of EoS for full-batch algorithms is relatively well-understood, neural networks are predominantly trained using mini-batch methods. As explicitly noted by Cohen et al. (2021, Section 6, Appendices G and H), their observations and analysis do not directly apply to mini-batch training, and EoS in SGD "does not center around the (full-batch) sharpness." We show that the EoS phenomenon does indeed generalize to SGD, and we identify the key quantity governing this generalization (*Batch Sharpness* in Definition 3). We model stability of SGD on the neural networks landscapes: our results show that *SGD is stable if on average the step is stable on the mini-batch landscape—not on the full-batch landscape.*

**What was empirically known for SGD.** In the context of mini-batch algorithms, **(i)** Jastrzębski et al. (2019; 2020) noticed that for SGD the phase transition happens earlier for smaller $\eta$ or smaller batch size $b$, but they did not quantify when. **(ii)** Cohen et al. (2021); Gilmer et al. (2021) established that initialization and architecture choices affect stability of SGD, without providing a definitive condition. **(iii)** When $\lambda_{\max}$ stabilizes, that always happens at a level they could not quantify which is below the $2/\eta$ threshold (Cohen et al., 2021; Keskar et al., 2016), see Figure 2, often without a proper progressive sharpening phase. This leaves the most basic questions open: *In what way the location of convergence of SGD acclimates to the choice of hy-*

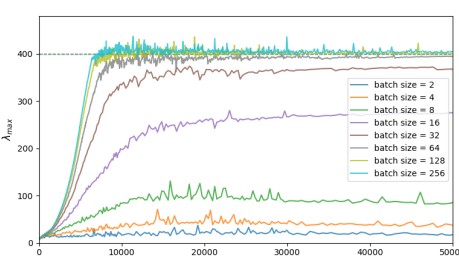

Figure 2: **SGD on CIFAR-10:** $\eta = 1/400$**.** The full-batch Hessian's $\lambda_{\max}$ plateaus *below* $2/\eta$. Smaller batch sizes lead to lower plateau values.

*perparameters? What are the key quantities involved?* To be more specific, can we characterize the training phenomena in **(i), (ii), (iii)** above? What determines them? Does SGD train in an unstable regime?

**Previous works on SGD stability.** A series of works, (Wu et al., 2018; Ma & Ying, 2021; Granziol et al., 2021; Wu et al., 2022; Mulayoff & Michaeli, 2024), studies constant-step-size SGD on quadratic losses via *linear stability* near the minima. From the perspective of Appendix B, these are valid instability criteria for a particular Lyapunov function, but two limitations remain for our purposes: (i) the resulting thresholds are expressed via $d^2$-dimensional operators and are not computable for modern neural networks; and (ii) they do not address whether, and in what sense, SGD on neural networks actually *trains* in an EoS-like regime of instability along its real trajectory (limited by the aforementioned incomputability). For further discussion see Appendix B and L. A number of works (Wu & Su, 2023; Agarwala & Pennington, 2024) showed that for SGD the regime of instability might be governed by the trace of loss Hessian/NTK, further discussed in Section M. Empirically, several works have documented oscillatory SGD dynamics in deep networks (Xing et al., 2018; Cohen et al., 2021; Ahn et al., 2022; Lee & Jang, 2023). However, these works do not establish whether any of those oscillations constitute a regime of instability; in particular, they do not distinguish between noise-driven (Type-1) and curvature-driven (Type-2) oscillations, see Section 3. Our work complements these efforts by (i) placing candidate criteria such as $\lambda_{\max}$, GNI, and Batch Sharpness within a unified instability framework, and (ii) identifying Batch Sharpness as a valid, empirically saturating, and computationally tractable instability criterion for SGD on neural networks.

**Flatness and Generalization.** SGD-trained networks consistently generalize better than GD-trained ones, with smaller batch sizes further enhancing generalization performance (Keskar et al., 2016; LeCun et al., 2012; Jastrzębski et al., 2018; Goyal et al., 2017; Masters & Luschi, 2018; Smith et al., 2021; Beneventano et al., 2024). This advantage has been widely attributed to some notion of flatness of the minima (Jiang et al., 2019; Jastrzębski et al., 2021; Hochreiter & Schmidhuber, 1994; Neyshabur et al., 2017; Wu et al., 2017; Kleinberg et al., 2018; Xie et al., 2020). Training algorithms explicitly designed to find flat minima have indeed demonstrated strong performance across various tasks (Izmailov et al., 2019; Foret et al., 2021). Our result is inherently a result about mini-batch training improving flatness. Specifically, we explain why: *Training with smaller batches constraints*

*the dynamics to areas with smaller eigenvalues of the full-batch Hessian.* This quantifies and characterizes prior observations that SGD tends to locate flat minima and that smaller batch sizes result in reduced Hessian sharpness (Keskar et al., 2016; Jastrzębski et al., 2021).

## 3 PRELIMINARIES: NOISE-DRIVEN *vs* CURVATURE-DRIVEN

The key defining aspect of EoS is about the solutions found by the algorithm adapting to the optimizer's hyperparameters. In the case of full-batch algorithms, this manifests through the emergence of an oscillatory regime. Mini-batch SGD, however, always oscillates because its gradient is noisy and the step size is not annealed. The central question, therefore, is *which* oscillations signal curvature-limited dynamics (EoS-like). We define stable and unstable oscillations based on the induction of catapults.

**Definition 1** (Quadratic instability and Catapults)**.** Consider the quadratic approximation of all the data point loss landscapes $\frac{1}{2}(\theta - x_i)^\top \mathcal{H}_i (\theta - x_i)$. We say that a set of hyperparameters is unstable if the trajectory exits[1] all the compact subsets of the region in which the quadratic approximation holds up to $\mathcal{O}(\eta)$. We say the algorithm experienced a *catapult* when this event happened.

We define *Type-1* (Noise-Driven Oscillation) those that are stable under the definition above, e.g., when we increase the step size and the trajectory re-stabilizes within the neighborhood. We call *Type-2* (Curvature-induced) the oscillations which saturate stability, i.e., the ones for which a small change in the hyperparameters induces a catapult as defined in Definition 1. Interestingly, both types of oscillation involve quantities stabilizing near the critical threshold of $2/\eta$, yet they differ.

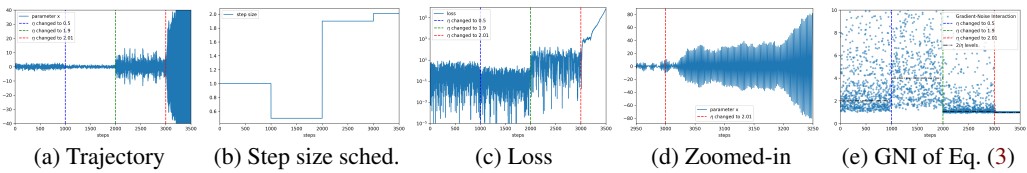

| (a) Trajectory | (b) Step size sched. | (c) Loss | (d) Zoomed-in | (e) GNI of Eq. (3) |

Figure 3: **Quadratics:** Dynamics of SGD on a 1-D quadratic with $N$ datapoints, $L(x) = \frac{1}{2N}\sum_i (x - a_i)^2$, where $a_i \sim \mathcal{N}(0, 1)$. Oscillations are present for any step size. Yet, only when the step size becomes larger than $2/\lambda_{\max} = 2$ (after the red line), the oscillations become unstable (d) and the loss diverges (c). Meanwhile, *GNI* consistently stays at $2/\eta$.

### 3.1 *Type-1* OSCILLATION AND GNI

SGD can wobble around a stationary point simply because gradients vary across batches and the step size is not annealed. This occurs even if the Hessian is small as, with fixed step-size, mini-batch noise has lower-bounded variance. Such noise-driven behavior is well-studied in classical stochastic approximation (Robbins & Monro, 1951; Mandt et al., 2016; Bottou et al., 2018; Mishchenko et al., 2020). We call *Type-1* oscillations any stochastic or chaotic trajectory which does not leave the region defined in Definition 1. We now introduce *GNI*, as a certificate of presence of oscillations:

**Definition 2.** We define G̲radient-N̲oise I̲nteraction (*GNI*):[2]

$$\underline{Gradient}\text{-}\underline{Noise}\ \underline{Interaction}\ (\theta) \quad := \quad \frac{\mathbb{E}_{B \sim \mathcal{P}_b}\left[\nabla L_B(\theta)^\top \mathcal{H}(L)\, \nabla L_B(\theta)\right]}{\|\nabla L(\theta)\|^2} \tag{1}$$

*GNI* is defined by dividing the two terms in the classical descent lemma. *The SGD trajectory oscillates, no matter the reason, if and only if GNI $\approx 2/\eta$*—see Proposition 2 for a more rigorous statement and the proof—indeed:

**Lemma 1.** $\mathbb{E}_{B \sim \mathcal{P}_b}\left[L(\theta_{t+1})\right] \approx L(\theta_t)$ *if and only if*

$$-\eta \|\nabla L(\theta)\|^2 + \frac{\eta^2}{2}\mathbb{E}_{B \sim \mathcal{P}_b}\left[\nabla L_B(\theta)^\top \mathcal{H} \nabla L_B(\theta)\right] \approx 0 \quad \Longleftrightarrow \quad GNI \approx \frac{2}{\eta}. \tag{2}$$

---

[1]This means that either SGD seen as a linear dynamical system is diverging **or** that the re-stabilization would happen be at a level which exits the largest region in which the quadratic approximation holds and so the dynamics *changes region*.

[2]Note that both the Hessian $\mathcal{H}$ and the gradient at the denominator are on the full-batch loss.

Notably, GNI is a quantity that is *centered* around $2/\eta$ *whenever* the trajectory oscillates. No matter the reason of the oscillation, see Figure 5. The regime of oscillation of SGD has been previously documented by measuring the expected total loss decrease by e.g., Cohen et al. (2021, Appendix H), Ahn et al. (2022), and Lee & Jang (2023) that tracked GNI explicitly[3].

## 3.2 SGD Always Oscillates *Type-1*

*Type-1* oscillations generally occur for SGD with fixed step size—even for simple quadratics[4].

**Proposition 1.** *Assume $L_B$ are quadratic. Around a local minimum $\theta^*$, fix $\eta > 0$ such that $\|(I - \eta\mathcal{H})^2 + \frac{\eta^2}{b}\mathbb{E}_B[\mathcal{H}(L_B)^2 - \mathcal{H}^2]\|_2^2 < 1$. Then the trajectory of SGD settles in a stationary distribution $\theta \sim \pi$ characterized by Type-1 oscillations but not Type-2 and satisfying*

$$\frac{\mathbb{E}_{\theta\sim\pi}\left[\mathbb{E}_{B\sim\mathcal{P}_b}\left[\nabla L_B(\theta)^\top \mathcal{H} \nabla L_B(\theta)\right]\right]}{\mathbb{E}_{\theta\sim\pi}\left[\|\nabla L(\theta)\|^2\right]} = \frac{2}{\eta}\left[1 + \mathcal{O}(\eta)\right], \tag{3}$$

*Independently of the moments of the Hessians $\mathcal{H}$ and $\mathcal{H}(L_B)$.*

See Appendix E for a formal statement and a proof. Crucially, the appearance of some quantity—GNI, as defined below—being $2/\eta$ implies the system is oscillating, not *why*. It does not mean, in principle, that the landscape or the curvature **adapted** to the hyper parameters. In this case (of *Type-1*), $2/\eta$ is about the ratio between the covariance of the gradients and the size of the full-batch gradient. Importantly, in this setting by perturbing the hyper parameters the system does not show catapults (as defined in Definition 1). When the size of oscillations increases (bigger step or smaller batch) the dynamics just increases the size of the oscillations—quickly restabilizing.

## 3.3 *Type-2*: Curvature-Driven Oscillation

Once the *local*, or *perceived*, curvature saturates with respect to the hyperparameters, the updates become unstable in a manner analogous to the classic EoS (Cohen et al., 2021). We define *Type-2* oscillation the trajectories for which a small perturbation of the hyperparameters induces a Catapult.

**Instability criteria, catapults, and EO(S)S.** As we further formalize in Appendix B, we view a training algorithm with fixed hyperparameters $h$ as a stochastic dynamical system $(\theta_t)_{t\geq 0}$ and summarize its local stability near a region $U$ by a scalar *instability criterion* $f(\theta)$ with threshold $c$ (Definition 4): if $f(\theta_0) > c$, the trajectory leaves every compact subset of $U$ in finite time. In the deterministic quadratic full-batch case this recovers the classical condition $\lambda_{\max}(\nabla^2 L) \leq 2/\eta$: once $\lambda_{\max} > 2/\eta$, gradient descent is linearly unstable. We say that SGD trains at the *Edge of (Stochastic) Stability*—analogously, shows *Type-2* oscillations—when such a criterion empirically *saturates* along the trajectory, i.e. $f(\theta_t) \approx c$ (up to $O(\eta poly(\log(\eta)))$) for extended periods.

On the local quadratic approximation around $\theta_t$, as we prove in Appendices B.2 and H, divergence has three equivalent manifestations: (i) breaking a valid instability criterion $f > c$; (ii) a *catapult* in the sense of Definition 1 (the quadratic trajectory leaves every compact subset of the trusted region $U_t$); and (iii) a *loss spike of sufficient size* on that quadratic model. If $f(\theta_t)$ is monotone in a destabilizing hyperparameter direction (e.g. $\eta \uparrow$ or $b \downarrow$) and is near $c$, Lemma 2 implies that a small perturbation of $h$ pushes $f > c$ and forces catapults / loss spikes on the quadratic approximation. We call an oscillatory trajectory *Type-2* precisely when this happens: the dynamics oscillates (so $GNI(\theta_t) \approx 2/\eta$, cf. Lemma 1), and a curvature-based instability criterion is saturated so that small destabilizing perturbations reliably induce catapults. This is what we mean by *curvature-driven* oscillations.

**Batch Sharpness as the curvature criterion.** For mini-batch SGD, different Lyapunov functions lead to different scalar statistics of the Hessians (depending on various moments or cumulants of the random mini-batch Hessians), and there is no reason a priori for all of them to saturate. The mathematical treatment in Appendix B and Section 5 and the empirical in Section 4 single out *Batch Sharpness* as the relevant instability criterion:

---

[3]In their notations $\operatorname{tr}(HS_b)/\operatorname{tr}(S_n)$. See Appendix C for further comparison with previous work.

[4]See Figure 3 and Appendix D

**Definition 3** (Batch Sharpness). We define

$$\textit{Batch Sharpness}\,(\theta) \quad := \quad \mathbb{E}_{B\sim\mathcal{P}_b}\left[\frac{\nabla L_B(\theta)^\top\,\mathcal{H}(L_\mathbf{B})\,\nabla L_B(\theta)}{\|\nabla L_\mathbf{B}(\theta)\|^2}\right]. \qquad (4)$$

The SGD update follows $\nabla L_B(\theta)$, and *Batch Sharpness* is the expected Rayleigh quotient of $\mathcal{H}(L_B)$ along these directions. It measures the *average directional curvature of the mini-batch landscapes along the steps SGD actually takes*, in contrast to *GNI* (Definition 2), which mixes mini-batch gradients with the full-batch Hessian.

## 4 SGD TYPICALLY OCCURS AT THE EoSS

We characterize here the phenomenon of the Edge of Stochastic Stability. We verify the emergence of EoSS across of a range of step sizes, batch sizes and architectures (Figure 6 and Appendix Q); datasets (CIFAR-10 and SVHN, Appendix R); and dataset sizes (8k and 32k subsets, Figure 7).

**1. Stabilization of *Batch Sharpness*.** SGD typically traind in an EoS-like regime:

> *SGD tends to train in a regime we call Edge of Stochastic Stability. Precisely, after a phase of progressive sharpening,* Batch Sharpness *reaches a stability level of* $2/\eta$, *and hovers there.*

In particular, the level of plateau of *Batch Sharpness* is $2/\eta$ independent of the batch size (Figure 1). Importantly, *Type-1* oscillations happen throughout most of the training as highlighted by the quantity of Proposition 1, see Figure 5, but they do not impact progressive sharpening which leads to the second phase of EoSS stabilization and *Type-2* oscillations. Importantly, analogously to EoS, training continues and the loss continues to decrease while *Batch Sharpness* is constrained by the step size magnitude.

**2. Stabilization of $\lambda_{\max}$ and *GNI*.** Crucially, stabilization of *Batch Sharpness* around $2/\eta$ happens while *GNI* has stabilized at $2/\eta$ already, and induces a corresponding stabilization of $\lambda_{\max}$. However, $\lambda_{\max}$ consistently settles at a lower level, due to a batch-size–dependent gap between the two. This is also influenced by the specific optimization trajectory, Figures 6 and 7. See Section I for factors determining their gap.

**3. Catapults.** Unlike in EoS, in the EoSS regime what stabilizes is an *expectation* of a quantity which the algorithm sees one observation at time. Occasionally, a sequence of sampled batches exhibits anomalously high sharpness—that is too high for the stable regime—and steps overshoot, triggering a catapult (Figure 4). This causes a spike in the loss (Section B.2), after which the trajectory either diverges or re-enters a stable region. If in this region *Batch Sharpness* is strictly less than $2/\eta$, this leads to a renewed phase of progressive sharpening, eventually returning to the EoSS regime. This aligns with, and maybe explains, previous observations about catapult behaviors, e.g., (Lewkowycz et al., 2020; Zhu et al., 2024).

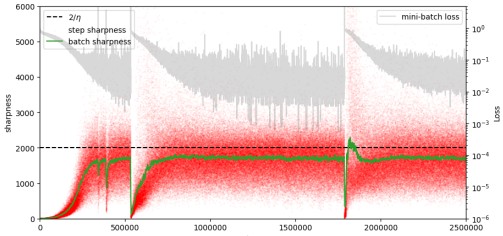

Figure 4: **Catapults at EoSS.** During EoSS, randomness in batch sampling might cause catapults, leading to renewed PS, and EoSS again. Notations follow Figure 6.

### 4.1 *Batch Sharpness* GOVERNS EoSS

Following Cohen et al. (2021) and the discussion in Section 3, we track how the training dynamics change when perturbing the hyperparameters mid-training. Overall, we find that *Batch Sharpness* governs EoSS behavior—mirroring how $\lambda_{\max}$ operates in the full-batch EoS—while the full-batch $\lambda_{\max}$ lags behind or settles inconsistently, underlining the mini-batch nature of SGD stability, see Appendix I. Increasing the step size $\eta$ or decreasing the batch size $b$ triggers a *catapult* spike in all the quantities in considerations and the training loss, before *Batch Sharpness* re-stabilizes near the

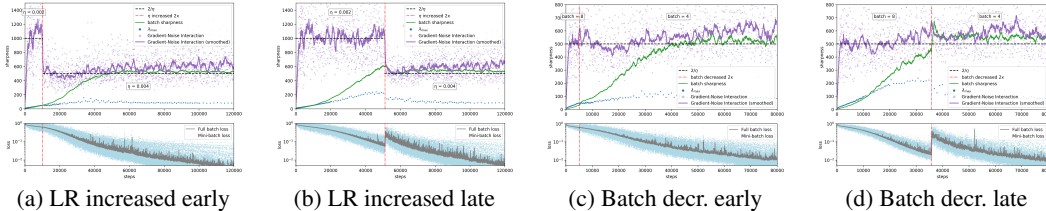

(a) LR increased early     (b) LR increased late     (c) Batch decr. early     (d) Batch decr. late

Figure 5: **(1)** The whole training happens with *Type-1* oscillations (see Proposition 1, $GNI \approx 2/\eta$), however, **(2)** *GNI* being $2/\eta$ does not govern *Type-2* oscillations—in particular, highlighting the difference in the two types of oscillations. **(3)** *Batch Sharpness* is instead an indicator of *Type-2* oscillations, as illustrated by the fact that catapults happen only when the shift in hyperparametes occurs **after** *Batch Sharpness* reaches $2/\eta$.

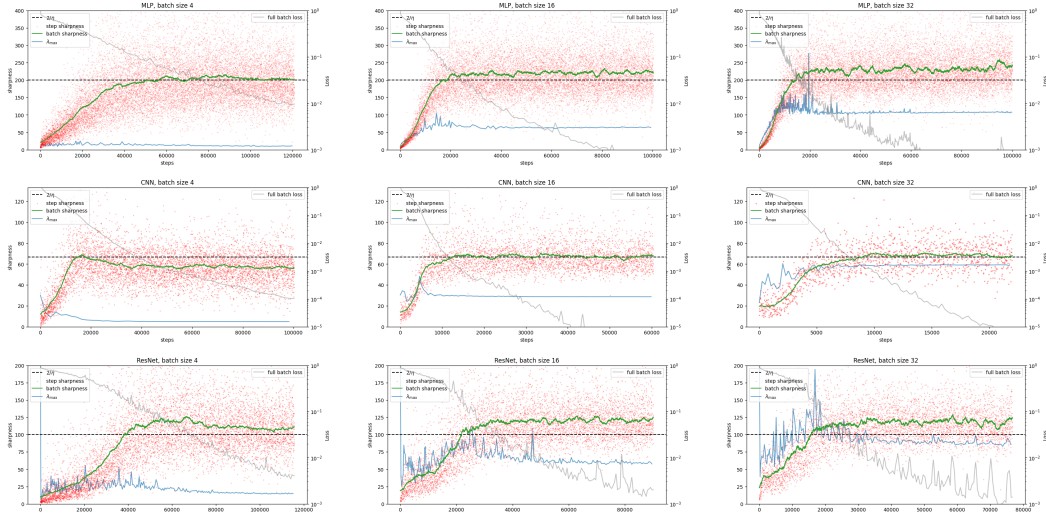

Figure 6: **Comparing different sharpness measures.** Red: *step sharpness*, observed sharpness on the current step's mini batch—essentially *Batch Sharpness* without the expectation; Green: *Batch Sharpness* (Definition 3); Blue: full-batch $\lambda_{\max}$. Top row: MLP (2 hidden layers of width 512); middle: 5-layer CNN; bottom: ResNet-14; all trained on an 8k subset of CIFAR-10.

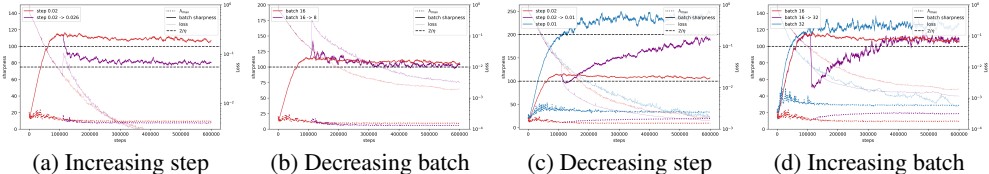

(a) Increasing step     (b) Decreasing batch     (c) Decreasing step     (d) Increasing batch

Figure 7: **Effects of changing step size or batch size in EoSS.** *Catapults:* (a) Increasing the step size $\eta$ causes a catapult spike before Batch Sharpness re-settles at the new $2/\eta$. (b) Decreasing the batch size $b$ increases *Batch Sharpness* and causes a catapult. *Restarting PS:* (c) Decreasing $\eta$ prompts renewed progressive sharpening. (d) Increasing $b$ lowers *Batch Sharpness* and re-starts progressive sharpening. The experiments are conducted on a 32k subset of CIFAR-10 to ensure sufficient complexity remains in the dataset, which is necessary for observing renewed progressive sharpening, consistent with observations by Cohen et al. (2021).

updated threshold $2/\eta$, see Figures 7a and 7b. This therefore pushes $\lambda_{\max}$ lower. Conversely, reducing $\eta$ raises the $2/\eta$ threshold. Analogously, increasing the batch size leaves $\lambda_{\max}$ unchanged but reduces *Batch Sharpness*. These changes prompt a new phase of progressive sharpening, see Figures 7c and 7d. Notice that, *instantaneously*, the change in batch size does not change the full-batch loss landscape, but only changes the mini-batch landscapes—the fact that this causes a catapult/restarts PS is an indicator that it is indeed the mini-batch landscape (and therefore *Batch Sharpness*) that governs the stability/instability of SGD. Here, $\lambda_{\max}$ also rises, but ultimately stabilizes at a lower value than if the entire training had run with the smaller step size/larger. Again, if stability was governed by $\lambda_{\max}$, this step-size adjustment would have had the same effect as starting from scratch with the new step size.

## 5    ON STABILITY

The previous section empirically demonstrated that mini-batch SGD generally settles into the EoSS regime, where *Batch Sharpness* hovers around $2/\eta$. Importantly, there exist many stochastic notions of stability, depending on different moments of the *random variable* $\mathcal{H}(L_B)$, see Appendix B. Some depend on quantities that can not be computed in high-dimensional experiments, others do not saturate empirically or their saturation does not induce EoSS, see Appendix M. In classical (full-batch) gradient descent, the condition $\eta < 2/\lambda_{\max}$ guarantees local stability by preventing divergence along the direction of the largest eigenvalue of a fixed Hessian. Here, is *Batch Sharpness* at $2/\eta$ *saturating the stability regime*? We answered positively empirically by showing that when you perturb hyperparameters you have explosions, see Figure 7. This proves empirically that we are at the Edge of Stability according to Definition 1. We show this mathematically in this section. Analogously, *is Batch Sharpness hovering at* $2/\eta$ *the cause of* EoSS *or a byproduct of something else happening?*. We already established empirically that the stabilization of $\lambda_{\max}$ is a byproduct of it, see Figure 7 and Section I. We establish this causality proving Theorem 1 below. Precisely, Theorem 1 shows that the trajectory is unstable with respect to Definition 1 when *Batch Sharpness* is bigger than $2/\eta$. We proceed to discuss the meanings of *Batch Sharpness*, how it relates to previous criteria of instability, and why it has that form.

### 5.1    *Type-2* IS ABOUT *Batch Sharpness*

The next theorem implies that SGD is unstable on quadratics if *Batch Sharpness* is bigger than $2/\eta$.

**Theorem 1.** *If Batch Sharpness is strictly bigger than* $(2 + \epsilon)/\eta$ *then* $\mathbb{E}[\|\nabla L_B(\theta_{t+1})\|^2/\|\nabla L_B(\theta_t)\|^2] > (1 + \epsilon)^2$. *On the second-order Taylor approximation, the norm of the mini-batch gradients increases exponentially with the SGD step and the trajectory is unstable in the sense of Definition 1.*

The proof relies on Jensen and Cauchy-Schwarz inequalities, see Appendix G. The use of these inequalities is its main limitation—we can not show the if and only if. However, Theorem 1 is the first (in)stability results that relies on a quantity we can efficiently estimate or compute in high-dimensional settings as neural networks. Note indeed that stability for quadratics is classically established by checking when $\mathbb{E}[\|\theta\|^2]$ diverges and when does not, see (Ma & Ying, 2021; Mulayoff & Michaeli, 2024) and Appendix L. *Batch Sharpness* is not directly related to these proofs and to the size of $\mathbb{E}[\|\theta\|^2]$.

### 5.2    CONNECTING BATCH SHARPNESS WITH EARLIER NOTIONS

**In the full-batch case.**    In the case of deterministic GD, EoS and EoSS are equivalent. If $\lambda_{\max} \geq 2/\eta$, then *Batch Sharpness*—the Rayleigh quotient—quickly becomes bigger than $2/\eta$ as the gradients align with the top-eigenvectors. Viceversa, for full-batch, $\lambda_{\max} \geq$ *Batch Sharpness*.

**Batch Sharpness governs instability.**    Theorem 1 allows to claim that *Batch Sharpness* generalizes $\lambda_{\max}$ to the mini-batch case. Precisely, it generalizes that as it is a valid instability certificate. Indeed, we showed that when either of those cross $2/\eta$ the system becomes unstable.

**Stability on the mini-batch landscape.**    The descent lemma on $L_B$ shows that one SGD step on the mini-batch landscape is locally stable iff its directional curvature is below $2/\eta$:

$$\frac{\nabla L_B(\theta)^\top \mathcal{H}(L_B) \nabla L_B(\theta)}{\|\nabla L_B(\theta)\|^2} \leq \frac{2}{\eta} \quad \Longleftrightarrow \quad -\eta\|\nabla L_B(\theta)\|^2 + \frac{\eta^2}{2} \nabla L_B(\theta)^\top \mathcal{H}(L_B) \nabla L_B(\theta) \leq 0. \tag{5}$$

*Batch Sharpness* at $2/\eta$ thus can be interpret as average such local stability.

**Alignment does not come for free.**    $\lambda_{\max}$ is *also* the averaged directional sharpness in the direction of the gradients for full-batch GD, because when it crosses $2/\eta$, the gradients align with the top eigenvectors. In the case of SGD, the gradients on different batches do not align though, as we show in Appendix K. Batch Sharpness is thus a natural generalization of $\lambda_{\max}$ as the averaged directional sharpness. Moreover, we observe how, in some large batch size cases, *Batch Sharpness* hovers at $2/\eta$ but $\lambda_{\max}$ is higher, as it does not govern stability due to mis-alignment of the gradients—e.g., Figure 44.

# 6 IMPLICATIONS: HOW NOISE-INJECTED GD DIFFERS FROM SGD

**SGD vs. Noisy Gradient Descent.** A common belief is that SGD's regularization stems from its "noisy" gradients, which find flatter minima. Our analysis highlights how the noise in the Hessians as crucial. To test this, we compare mini-batch SGD (batch size 16) against three noisy GD variants—see details in Appendix J: **(i)** *Gaussian reweighting on the samples (Wu et al., 2020)* which maintains the noise structure in the Hessians; **(ii)** *Isotropic/Anisotropic diagonal noise (Zhu et al., 2019)*; and SDE dynamics (Li et al., 2017). As shown in Figure 8, only noise which maintains the higher moments of the Hessian(s) (and thus implicitly preserves the mini-batch landscape structure) leads to an EoSS-like regime with $\lambda_{\max}$ stabilizing well below $2/\eta$. Classical analyses of neural network optimization

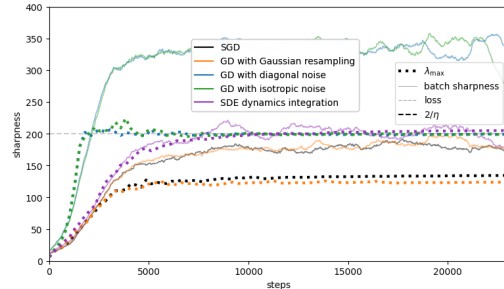

Figure 8: **SGD vs. Noisy GD vs SDE.** Only noise preserving the mini-batch structure of SGD leads to $\lambda_{\max}$ plateauing below $2/\eta$ (akin to EoSS and as observed by (Keskar et al., 2016)). Noise injection fails to reproduce this behavior even with the same covariance SGD's.

often assume noisy trajectories on a single, static, landscape. This is a further proof that the community has to be careful when modeling SGD as noise-injected GD or SDEs.

> Standard SDE, noise-injected GD, or analogous approximations of SGD cannot describe the solution found by SGD or its behavior under the assumption of progressive sharpening. Indeed, they typically ignore any statistics of the Hessians except for the mean.

# 7 CONCLUSIONS, LIMITATIONS, AND FUTURE WORK

**Conclusions.** We have addressed the longstanding question of *if* and *how* mini-batch SGD enters a regime reminiscent of the "Edge of Stability" previously observed in full-batch methods. Contrary to the usual focus on the global Hessian's top eigenvalue, we uncovered that *Batch Sharpness*—the expected directional curvature of the mini-batch landscape in the direction of its own gradient—consistently rises (progressive sharpening) and then hovers around $2/\eta$, independent of batch size. This behavior characterizes a new regime "Edge of Stochastic Stability", which explains how mini-batch training can exhibit catapult-like surges and settle into flatter minima even when the full-batch Hessian remains below $2/\eta$. Our analysis clarifies why smaller batch sizes and larger step sizes both constrain the final curvature to a lower level, thereby linking these hyperparameters to flatter solutions and often improved generalization. Furthermore, we show that this phenomenon depends on the noise injected into the Hessians by mini-batch optimizers, highlighting important limitations of SDE-based approximations. Overall, the EoSS framework unifies several empirically observed effects—catapult phases, dependence on batch size, and progressive sharpening—under a single perspective focused on the *mini-batch* landscape and its directional curvature.

**Limitations.** (*i*) We have tested only image-classification tasks, leaving open whether similar phenomena arise in NLP, RL, or other domains. (*ii*) Our experiments mainly use fixed step sizes and standard architectures, so very large-scale or large-batch settings remain less explored. (*iii*) We have not analyzed momentum-based or adaptive methods (e.g. Adam), even though full-batch EoS has been seen there (Cohen et al., 2022).

**Future Work.** Beyond addressing these limitations, several directions remain: Understanding (*i*) *where* $\lambda_{\max}$ stabilizes; (*ii*) how EoSS and EoS affect performances and properties of the neural network, e.g. (Lyu et al., 2023; Arora et al., 2022; Ahn et al., 2023; Zhu et al., 2023; Wang et al., 2022; Beneventano & Woodworth, 2025); (*iii*) consequently if it is benign effect or not; (*iv*) what the *other* sources of instability are there in the (pre-)training; (*v*) better describing the phenomenon of progressive sharpening; and (*vi*) understanding its causes.

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

CONTENTS

# A  ACKNOWLEDGEMENT OF LLMS USAGE

We acknowledge the use of DeepSeek, Claude Code, Codex, and ChatGPT for code assistance. We used ChatGPT and Claude for text editing suggestions, proof-reading, and LaTeX editing help.

# B  THEORY PRELIMINARIES: A FRAMEWORK FOR INSTABILITY

In this section, we give a minimal formal framework for the way we deal with *(in)stability* in the rest of the paper. Our goal is to isolate (i) *instability criteria*—sufficient certificates for divergence of a (stochastic) dynamical system—and (ii) ways to recognize the regime where such a criterion *empirically saturates*, which we call the Edge of (Stochastic) Stability (Eo(S)S). We explain why, on the quadratic approximation of the loss, the following three viewpoints are equivalent for us:

- breaking an instability criterion,
- observing a *catapult* (a large excursion out of the approximately quadratic region),
- and observing a spike of appropriate size in the loss.

This justifies why, in experiments, we both perturb hyperparameters and look for loss spikes to diagnose EoSS and what we consider catapults.

## B.1  DEFINING INSTABILITY

**On the notion of stability.** Different notions of stability are in principle possible for training algorithms. If for every data point $z_i$ there exist $\mathcal{H}_i \in \mathbb{R}^{d \times d}$ and $x_i \in \mathbb{R}^d$ such that the loss on $z_i$ is $L_i(\theta) = \frac{1}{2}(\theta - x_i)^\top \mathcal{H}_i(\theta - x_i)$, we can consider the evolution of, e.g., the squared distance to a solution $\theta^*$, or more generally any Lyapunov function $V(\theta)$. Typical stability requirements have the form

$$\mathbb{E}[V(\theta_{t+1})|\theta_t] \leq V(\theta_t), \quad \mathbb{E}\left[\frac{V(\theta_{t+1})}{V(\theta_t)}\,\Big|\,\theta_t\right]^{\,5} \leq 1, \quad \text{or} \quad \lim_{t\to\infty}\frac{1}{t}\log\big(V(\theta_t)\big) \leq 0, \quad \text{etc.} \quad (6)$$

In all these cases one can expand $\theta_{t+1}$ in a Taylor series around $\theta_t$ and obtain an equivalent inequality that can be expressed in terms of a *scalar quantity built from the Hessians*—generally termed *notion of curvature* (for instance, an eigenvalue, an operator norm, or a combination of low-order cumulants of the distribution of $\{\mathcal{H}_i\}_i$). As an example, non-expansion of the second moment of the parameters near a minimum—*linear stability*—was studied by Wu et al. (2018); Ma & Ying (2021); Mulayoff & Michaeli (2024) and boils down to a bound on an operator of the form

$$\big\|\mathbb{E}_{B\sim\mathcal{P}_b}\big[(I - \eta H(L_B))^{\otimes 2}\big]\big\| \leq 1. \quad (7)$$

We refer to Appendix L for a detailed comparison to linear stochastic stability.

**Criteria for instability.** We now isolate the abstract notion of stability, or better, certificate of instability, that will be used throughout the paper. Crucially, we define an instability criterion only through *sufficiency* for divergence. We do not require necessity for divergence (equivalently, sufficiency for stability), which is the type of condition typically required for convergence proofs and is, for example, the nature of the condition in Wu et al. (2018).

**Definition 4** (Instability criterion). Consider a training algorithm (a discrete-time dynamical system) $(\theta_t)_{t\geq 0}$ on a parameter space $\Theta \subseteq \mathbb{R}^d$ with fixed hyperparameters $h$ (e.g. learning rate, batch size). Let $U \subseteq \Theta$ be an open set (typically, a region where a given local approximation of the loss is trusted). Let $f : U \to \mathbb{R}$ and $c \in \mathbb{R}$. We say that $f$ is a *valid instability criterion with threshold $c$* for the algorithm on $U$ if the following holds:

$$f(\theta_0) > c \quad \Longrightarrow \quad (\theta_t)_{t\geq 0} \text{ leaves every compact subset of } U.$$

That is, for any compact $K \subset U$ containing $\theta_0$, there exists a finite time $T$ such that $\theta_T \notin K$. We say that the instability criterion $f$ is *saturated* at $\theta$ if $f(\theta)$ is (approximately) equal to $c$; in practice, up to an $O(\eta)$ tolerance.

---

[5] This is the one we find in our empirical work, with $V(\theta) = \|\nabla L_B(\theta)\|^2$.

In words: an instability criterion is a scalar quantity $f$ together with a threshold $c$ such that crossing $f > c$ is *sufficient* to force divergence from the region $U$ we trust as a local model. For a specific $f$ we generally want the *smallest* such $c$, which depends both on $f$ and on the underlying dynamical system (algorithm, data, loss, architecture). A canonical example is full-batch EoS, where $U = \mathbb{R}^d$, $f(\theta) = \lambda_{\max}(\nabla^2 L(\theta))$ and $c = 2/\eta$: crossing $\lambda_{\max} > 2/\eta$ makes GD linearly unstable on any compact set.

> The question if an optimizer acts at the *Edge of (Stochastic) Stability* thus becomes:
>
> *Are there criteria of instability, e.g., Eqs. (6, 7), that empirically saturate during training?*

**Quadratic and deterministic case.**   To connect what above with Cohen et al. (2021), in the case in which we use a full-batch method (deterministic) over a quadratic loss $L$, the criteria of instability obtainable from a Lyapunov function as in Eq. (6) simplify up to higher order in $\eta$ as

    **(i)** only full-batch quantities appear in the Taylor expansion, no higher cumulants of the Hessians;

    **(ii)** shortly after reaching instability, the gradient aligns with the eigenvector of the highest eigenvalue $\lambda_{\max}$ of the Hessian.

In particular, GD is an asymptotic stable (and Lyapunov stable) linear dynamical system if and only if $\lambda_{\max} \leq 2/\eta$. Cohen et al. (2021) empirically showed GD trains neural networks in the regime in which the inequality $\lambda_{\max} \leq 2/\eta$ is saturated, Damian et al. (2023) showed this mathematically under the assumption of progressive sharpening, making precise that saturation means that $\lambda_{\max} - 2/\eta \in [-c\eta|\log(\eta)|, +c\eta|\log(\eta)|]$ for some constant $c$.

### B.2   Identifying a Regime of Instability

But how do we find if such an instability criterion is saturated? In practice, we diagnose this through the observation of *catapults* and *loss spikes* when the hyperparameters are perturbed. In this subsection we formalize this connection on the local quadratic approximation.

On the Taylor approximation of the loss, the underlying property is *divergence of the dynamics on* $U_t$, and the three diagnostics introduced—which we use throughout the article—are just different ways to detect it.

> The three phenomena below are *equivalent manifestations of divergence on the quadratic approximation* rather than three logically independent assumptions:
>
> Breaking an instability criterion $\iff$ Catapult (Def. 5) $\iff$ Loss spike of sufficient size.

This equivalence—which we prove in Appendix H, in this precise sense, explains why, in practice, we use both catapults and loss spikes as our main empirical signatures of instability at the EO(S)S.

**Catapults on the quadratic approximation.**   Fix a time $t$ and a point $\theta_t$. Assume that for each $i$ the per-sample loss $L_i$ is twice differentiable with $\beta$-Lipschitz Hessian in a neighborhood of $\theta_t$, and let $\widetilde{L}_i(\theta) := \frac{1}{2}(\theta - x_i)^\top \mathcal{H}_i (\theta - x_i)$ be the second-order Taylor approximation of $L_i$ at $\theta_t$. Let $U_t$ denote an open neighborhood of $\theta_t$ where the quadratic approximation is accurate, e.g. where $\left| \mathcal{H}(L_i(\theta)) - \mathcal{H}_i \right| \leq C\eta$ for all $\theta \in U_t$ and all $i$ and some constant $C > 0$ depending on $\beta$ and the local geometry. Consider SGD (or GD) run on the quadratic model $\widetilde{L}(\theta) := \frac{1}{N} \sum_i \widetilde{L}_i(\theta)$ with the same hyperparameters as the original dynamics.

**Definition 5** (Catapults on the quadratic model)**.** We say that the algorithm *experiences a catapult at time $t$* if, when run on $\widetilde{L}$ from initialization $\theta_t$, the resulting trajectory $(\theta_s)_{s \geq t}$ leaves every compact subset of $U_t$ in finite time.

This definition is deliberately phrased in the same language as Definition 4: a catapult is precisely a divergence event for the quadratic dynamics on the region where the approximation is trusted.

**Hyperparameter perturbations and saturation.**   Let $f$ be a valid instability criterion as in Definition 4, with threshold $c$. In practice, $f$ and $c$ depend on hyperparameters $h$ (e.g. $f(\theta; \eta, b)$ and $c = c(\eta, b)$). We are interested in settings where:

(a) $f(\cdot; h)$ is a valid instability criterion for the quadratic model on $U_t$;

(b) $f(\theta_t; h)$ is *monotone* in some destabilizing direction in $h$ (e.g. increasing $\eta$ or decreasing $b$ increases $f(\theta_t; h) - c(h)$);

(c) along the observed trajectory at hyperparameters $h_0$, $f(\theta_t; h_0)$ *saturates*, i.e. $f(\theta_t; h_0) \approx c(h_0)$.

**Lemma 2** (Tight instability criterion $\Rightarrow$ catapult under perturbation). *Assume $h_0$ saturates $f$ at $\theta_t$. Under assumptions (a)–(c) above, any sufficiently small destabilizing perturbation of $h_0$ (e.g. $\eta \uparrow$ or $b \downarrow$) produces hyperparameters $h$ such that $f(\theta_t; h) > c(h)$. By validity of the criterion, the quadratic-model trajectory from $\theta_t$ leaves every compact subset of $U_t$ in finite time, i.e., the algorithm experiences a catapult at time $t$ in the sense of Definition 5.*

Thus a instability criterion that is monotone in a hyperparameter gives a concrete way to test whether we are at the EO(S)S: if it saturates along training, then a small destabilizing perturbation must trigger a catapult on the quadratic model. Conversely, if a quantity empirically saturates but small destabilizing perturbations do *not* lead to catapults, this quantity cannot be a valid instability criterion for the dynamics of interest.

**From catapults to loss spikes.** On the quadratic model $\widetilde{L}$ and within $U_t$, the loss is a quadratic form in $\theta - \theta_t$. If a catapult occurs at time $t$, the quadratic trajectory leaves every compact subset of $U_t$, so $\|\theta_s - \theta_t\|$ eventually exceeds any fixed radius $R$ for which $B_R(\theta_t) \subset U_t$. In particular, before (or as) the iterate exits such a ball, $\widetilde{L}(\theta_s)$ must increase by at least a fixed factor $\mathcal{O}_\eta(1)$ compared to $\widetilde{L}(\theta_t)$. We refer to such an increase as a *loss spike of sufficient size* on the quadratic model[6].

Conversely, a loss spike of sufficiently large ($\mathcal{O}_\eta(1)$) relative size on the quadratic model cannot occur if the dynamics remains linearly stable and confined to a fixed compact subset of $U_t$: in that regime, the quadratic dynamics is a contraction in a suitable norm, and both $\|\theta_s - \theta_t\|$ and $\widetilde{L}(\theta_s)$ remain uniformly controlled. Thus such a spike implies that the quadratic dynamics is divergent on $U_t$ in the sense above, and therefore that some valid instability criterion for the quadratic model must be broken along the trajectory.

### B.3 GUIDING QUESTIONS FOR EoSS

Understanding *if and how* SGD trains at the EoSS is thus inherently linked to:

- what scalar Hessian-based statistic (if any) saturates as a valid instability criterion (Q1);
- how progressive sharpening and self-stabilization act on the moments/cumulants of the Hessian distribution and thereby select that statistic (Q2);
- and whether this statistic is computable and usable in high-dimensional practice (Q3).

The rest of the paper answers these questions, at least partially: we will show that Batch Sharpness is a valid and tractable instability criterion for SGD on the quadratic approximation, that it empirically saturates in the practice of neural network training. Importantly, *Batch Sharpness* is the first proposed such quantity which does not fail any of these desiderata.

**Stochastic and non-quadratic case.** The discussion above shows that for deterministic gradient descent on a quadratic loss, all the usual stability notions collapse to the same scalar quantity

$$f(\theta) = \lambda_{\max}(\nabla^2 L(\theta)), \qquad f(\theta) \le 2/\eta$$

and training at the edge of stability means that this inequality saturates. In contrast, for *stochastic* training on a *non-quadratic* loss, the situation is substantially more delicate: different Lyapunov functions lead to different scalar quantities built from the Hessians (e.g. different moments or cumulants over $\{\mathcal{H}_i\}_i$), and there is no reason a priori for them to agree or to saturate at the same values. Note for instance, that in (7) the first two cumulants appear, while in *Batch Sharpness* the first three. We organize the rest of the paper around three guiding questions.

---

[6]We are imprecise on purpose here, because the size required for the spike to be a catapult depends in practice on the *global* geometry. As we see in Corollary 1 and Figure 3 for quadratics, when you perturb a stable step size to a new stable step size, the loss jumps by $\mathcal{O}\big(\eta/(2 - \eta\lambda_{\max})\big)$ before restabilizing, that size is not a catapult spike.

**(Q1) Existence and (non-)uniqueness of a criterion.** In principle, there may exist infinitely many instability criteria that saturate along the SGD trajectory.[7] This leads to the first question:

*Is there a "distinguished" scalar Hessian-based quantity that both (i) is a valid instability criterion in the sense of Definition 4 and (ii) empirically saturates at the EoSS for SGD?*

Even if such a quantity exists, it need not be unique in principle; part of our contribution is to show that *Batch Sharpness* is one such quantity and to argue that it is preferred over several alternatives (e.g. GNI).

**(Q2) Mechanism: progressive sharpening and self-stabilization.** Different candidate criteria depend on different statistics of the Hessian distribution. For example, the operator in (7) only involves the first and second cumulants of $\{\mathcal{H}_i\}_i$, whereas Batch Sharpness (even in the quadratic case) also depends on the third, and $\lambda_{\max}$ depends only on the first. Which of these is actually driven to saturation by the dynamics depends on how *Progressive Sharpening* and *self-stabilization* act on these statistics:

- progressive sharpening pushes up certain Hessian statistics during training;
- self-stabilization constrains them by preventing blow-up.

This leads to the second question:

*Which scalar Hessian-based statistic does the combination of progressive sharpening and self-stabilization actually push to a saturated instability threshold in practice?*

If, for instance, progressive sharpening mostly increases the third cumulant while leaving the first two relatively unchanged, we should expect Batch Sharpness (which depends on the third cumulant) to be the relevant instability criterion, rather than quantities that only see the first or second cumulant.

**(Q3) Computability and usefulness in high dimension.** Finally, some natural instability criteria are not efficiently estimable in high dimension. For instance, conditions based directly on operators like (7) or the exact instability criterion of Mulayoff & Michaeli (2024) live in a $d^2$-dimensional space and are infeasible to evaluate in modern neural networks. This motivates the third question:

*Is there a valid instability criterion for SGD on neural networks that is both empirically saturating and computationally tractable in high dimension?*

In practice, this means looking for criteria that can be estimated in roughly $\mathcal{O}(d \cdot poly(\log(d)))$ time.

## C  COMPARISON WITH PREVIOUS EMPIRICAL WORK

Lee & Jang (2023) introduce several quantities crucial for understanding neural network training dynamics. Below, we discuss the relationships among $\lambda_{\max}$, *Batch Sharpness*, and Interaction-Aware Sharpness (IAS, Lee & Jang (2023)), emphasizing that a comprehensive theory of mini-batch dynamics should explain their distinct plateau timings and interconnected behaviors. We conjecture that a complete theory of stochastic gradient descent (SGD) dynamics would elucidate these metrics' precise interrelations and their different plateau timings.

**Interaction-Aware Sharpness.** Lee & Jang (2023) introduce Interaction-Aware Sharpness (IAS), denoted $\|\mathcal{H}\|_{S_b}$:

$$\|\mathcal{H}\|_{S_b} \quad := \quad \frac{\mathbb{E}_{B\sim\mathcal{P}_b}\big[\nabla L_B(\mathbf{x})^\top \mathcal{H} \nabla L_B(\mathbf{x})\big]}{\mathbb{E}_{B\sim\mathcal{P}_b}\big[\|\nabla L_B\|^2\big]}.$$

This quantity shares structural similarities with both *Batch Sharpness* (Definition 3) and the *Gradient-Noise Interaction* (Proposition 1), differing from the latter only in the denominator. The

---

[7]For instance, in Section 4 we show that $\lambda_{\max} \leq 2/\eta$ does not saturate for batch sizes smaller than a problem-dependent critical batch size. It is an open problem to characterize the level at which $\lambda_{\max}$ stabilizes in that regime.

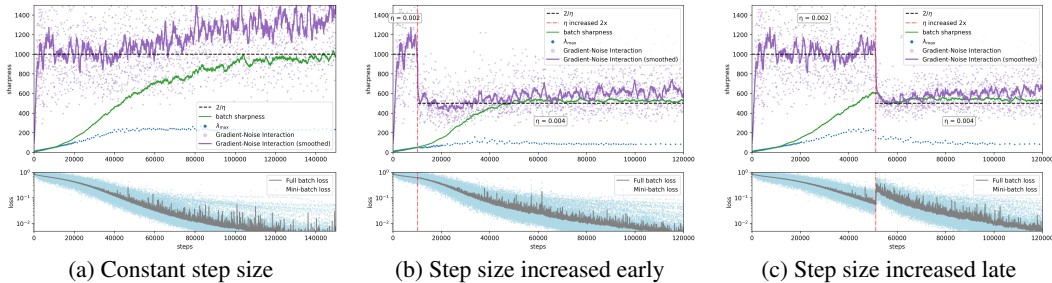

(a) Constant step size      (b) Step size increased early      (c) Step size increased late

Figure 9: We demonstrate that the saturation of *GNI* does not govern a sharpness-related regime of instability typical of Type-2 oscillations - and in particular, highlighting the difference in the two types of oscillations. When we double the step size after *batch sharpness* is at least half of $2/\eta$ threshold (so that it is beyond the new $2/\eta$ level), training exhibits a catapult surge in the loss (c). But if we make the same change *before* batch sharpness crosses that level—despite *GNI* already saturating—no catapult occurs. (b)

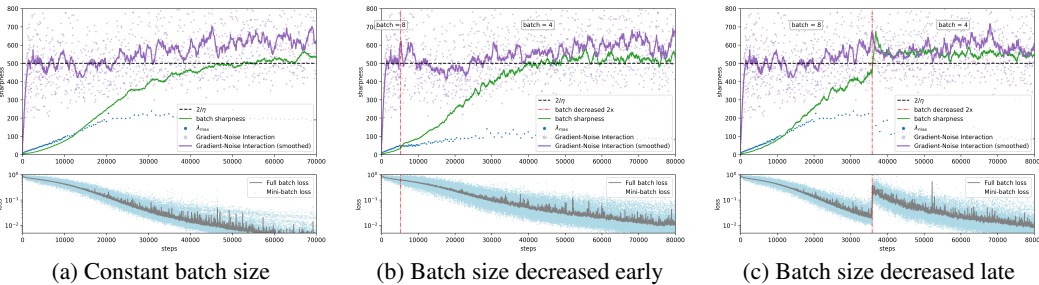

(a) Constant batch size      (b) Batch size decreased early      (c) Batch size decreased late

Figure 10: Similarly, reducing the batch size only triggers catapults if batch sharpness, not *GNI*, exceeds the threshold.

key distinction from *Batch Sharpness* lies in which Hessian is evaluated: IAS measures the directional curvature of the **full-batch** loss landscape $L$ along mini-batch gradient directions, while *Batch Sharpness* measures the directional curvature of **mini-batch** loss landscape $L_B$ along their corresponding gradients. This distinction is crucial, as mini-batch Hessians vary with batch selection while the full-batch Hessian remains fixed.

Notably, with full-batch GD, IAS serves as a directional alternative to the maximal Hessian eigenvalue, $\lambda_{\max}$, introduced by Cohen et al. (2021). IAS aligns closely with the $2/\eta$ threshold, unlike $\lambda_{\max}$, which often remains slightly above this threshold during EoS, especially at the beginning of it. Since IAS measures *directional* curvature, we have $\|\mathcal{H}\|_{S_n} \leq \lambda_{\max}$. Consequently, in the mini-batch setting, IAS stabilizes below $2/\eta$, consistent with empirical observations from Jastrzęb-ski et al. (2019; 2020); Cohen et al. (2021) and our Figure 2. Notably, when $B = n$, our *Batch Sharpness* coincides with IAS rather than $\lambda\max$, reinforcing the interpretation of *Batch Sharpness* as the relevant metric stabilizing at $2/\eta$ even under full-batch conditions.

**Relation to Gradient-Noise Interaction.** Another metric from Lee & Jang (2023) is defined as:

$$\frac{\text{tr}(HS_b)}{\text{tr}(S_n)} = \frac{\mathbb{E}_{B \sim \mathcal{P}_b}\left[\nabla L_B(\mathbf{x})^\top \mathcal{H} \nabla L_B(\mathbf{x})\right]}{\|\nabla L\|^2}$$

which coincides exactly with our definition of GNI (Proposition 1). As detailed in Section 3 and Appendix D, the stabilization of GNI around $2/\eta$ signals the presence of oscillations, at least Type-1 oscillations. Lee & Jang (2023) provide extensive empirical evidence demonstrating that neural networks spend much of their training within this oscillatory regime (see also Figures 9a and 10a). This contrasts traditional theoretical analyses (Bottou et al. (2018); Mandt et al. (2016)), which consider oscillations only near the manifold of minima.

**Distinguishing oscillation types.** It is crucial to note that GNI around $2/\eta$ does not inherently indicate instability. As clarified in Sections 3, 4 and Appendix E, not all oscillations are inherently unstable. Figures 5, 9b, 10b illustrate that altering hyperparameters when GNI is around $2/\eta$ typically does not trigger instability (catapult-like divergence), contrary to expectations if the system was in an EoS-like regime of instability. Instead, as shown in Figures 5, 9c, 10c, *Batch Sharpness* more reliably predicts a regime of instability. Additionally, Figure 11 highlights GNI's independence from progressive sharpening, a necessary precursor to Type-2 (curvature-driven) oscillations and EoS-like instabilities, as detailed in Appendix F.

**Missing Progressive Sharpening.** Extensively, both in our experiments and in the ones of Lee & Jang (2023), GNI grows to $2/\eta$ in a few initial steps (and sometimes from the very beginning if the intialization size is large) without ever being in subject to a phase of progressive sharpening unlike *Batch Sharpness* and $\lambda_{\max}$. The phase of growth of GNI is generally short and independent of the size, the behavior, and the phase in which *Batch Sharpness* and $\lambda_{\max}$ are.

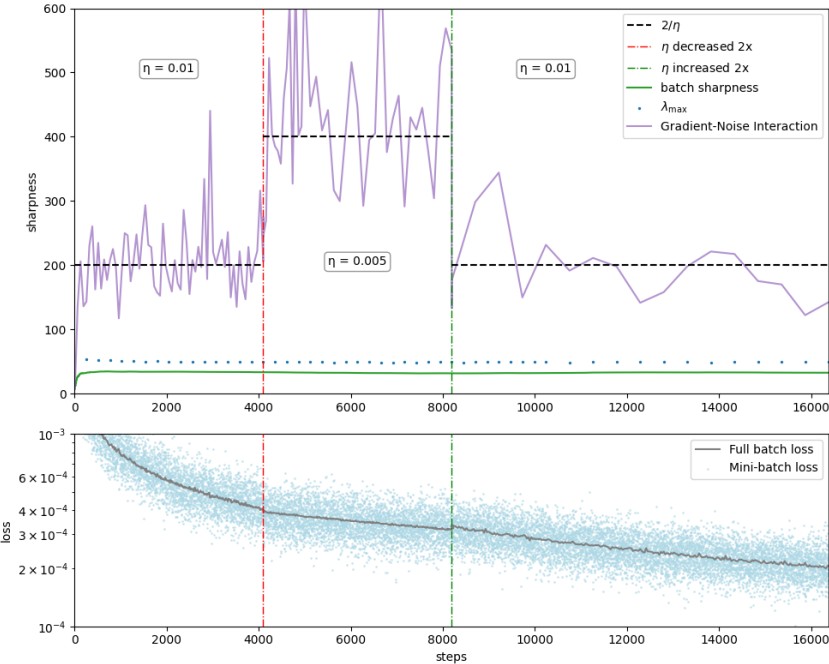

Figure 11: We construct a 32k-point "easy" CIFAR-10, where we "pull apart" all the 10 classes, so the classes become linearly separable. In this case, there is virtually no "learning" to be done, and therefore, there is barely any progressive sharpening happening (as established Cohen et al. (2021), progressive sharpening does not happen if the dataset "is not complex enough"). Yet, *GNI* still stabilizes at the initial level of $2/\eta$. More importantly, when we decrease and then increase the step size, the *GNI* measure restabilizes to the corresponding new thresholds, while $\lambda_{\max}$ does not change. That means that *GNI* is *independent of the curvature of the loss landscape* and is unrelated to progressive sharpening, and thus Type-2 oscillations and EoS-like instability regimes.

## D  ON THE TWO TYPES OF OSCILLATIONS IN SGD DYNAMICS

A fundamental challenge in analyzing SGD compared to GD stems from the inherent oscillations induced by mini-batch gradient noise. This appendix, together with Appendix C (also see proofs in E and G), extends the discussion in Section 3 by formally distinguishing between two distinct types of oscillations: noise-driven (Type-1) and curvature-driven (Type-2). This distinction is crucial because Type-1 oscillations occur independently of the loss landscape's curvature and thus do not exert a regularizing effect on the sharpness of the final solution. In contrast, Type-2 oscillations are directly caused by landscape curvature and induce an implicit regularization effect by discouraging convergence towards sharp minima.

We begin with a minimalistic example to illustrate the nature of Type-1

## D.1    A MINIMALISTIC QUADRATIC EXAMPLE.

Here we show mathematically what we see empirically in Figure 3 (the simplified version—only two data points). Consider a regression problem with two datapoints, 1 and $-1$, and a linear model $f(x) = x$ under the quadratic loss. The (scaled) full-batch loss is given by:

$$L(x) \;=\; \frac{1}{4}(x-1)^2 \;+\; \frac{1}{4}(x+1)^2.$$

Batch-1 SGD updates with step-size $0 < \eta < 2$ result in oscillatory behavior around the optimum $x = 0$ due entirely to gradient noise, with amplitude approximately $\sqrt{\frac{\eta}{2-\eta}}$. Crucially, the Hessian in this example is small ($\frac{d^2 L}{dx^2} = 1$), demonstrating that these persistent oscillations are entirely noise-driven (Type-1).

Formally, the SGD update is:

$$x_{t+1} = x_t - \eta \nabla \ell_{i_t} = (1-\eta)x_t + \eta \xi_t$$

where $l_{i_t}$s are the individual datapoint losses, and $\xi_t$s are i.i.d Rademacher random variables. Thus, we obtain the first two moments explicitly:

$$\mathbb{E}[x_t] = (1-\eta)\mathbb{E}[x_{t-1}] = (1-\eta)^t x_0$$

$$\mathbb{E}[x_t^2] = (1-\eta)^2 \mathbb{E}[x_{t-1}^2] + \eta^2 = (1-\eta)^{2t} x_0 + \frac{\eta^2}{1-(1-\eta)^2}\left(1-(1-\eta)^{2t}\right)$$

This implies convergence in expectation for $0 < \eta < 2$, with a limiting variance given by:

$$\lim_{t \to \infty} \mathbb{E}[x_t^2] = \frac{\eta}{2-\eta}$$

and divergence for $\eta > 2$.

A key observation is that increasing $\eta$ to any value $\eta_1 < 2$ merely changes the amplitude of oscillations to $\sqrt{\frac{\eta_1}{2-\eta_1}}$ without triggering any catapult-like behavior. The only step size for which we observe Type-2 (curvature-driven) oscillations and an EoS-like[8] instability is precisely $\eta = 2$, where the dynamics effectively become a random walk, and any larger step size leads to divergence.

> *Crucially, when $\eta < 2$ oscillations occur persistently on the full-batch loss, despite the individual steps on the mini-batch loss remaining stable.*

The oscillation is due to the fact that the mini-batch loss landscape shifts from step to step, not to the fact that the steps are unstable.

## D.2    PROOF OF LEMMA 1

We propose here the formal version of Lemma 1.

**Proposition 2** (Loss increment and GNI). *Assume $L$ is three times continuously differentiable and its Hessian is $L_2$–Lipschitz in a neighborhood of $\theta$. Then for any mini–batch $B$ and step size $\eta > 0$ small enough we have*

$$\mathbb{E}_B\big[L(\theta - \eta \nabla L_B(\theta)) - L(\theta) \,\big|\, \theta\big] = -\eta \|\nabla L(\theta)\|^2 + \frac{\eta^2}{2}\, \mathbb{E}_B\big[\nabla L_B(\theta)^\top \mathcal{H}(\theta)\, \nabla L_B(\theta)\big] + \mathcal{O}(\eta^3), \tag{8}$$

---

[8]The key difference between these oscillations and genuine EoS behavior in neural networks is that, in the quadratic case, the full-batch loss does not decrease, making this scenario inherently less informative. In contrast, neural networks exhibit a surprising, albeit non-monotonic, decrease in loss within this instability regime, an effect arising from the multidimensional nature of their optimization landscape (Damian et al., 2023)

*where the $\mathcal{O}(\eta^3)$ constant depends only on $L_2$ and an upper bound on $\|\nabla L_B(\theta)\|$. Equivalently,*

$$\mathbb{E}_B\big[L(\theta_{t+1}) - L(\theta_t) \,|\, \theta_t\big] = -\eta\|\nabla L(\theta_t)\|^2\Big(1 - \frac{\eta}{2}\,\mathrm{GNI}(\theta_t) + \mathcal{O}(\eta^2)\Big), \tag{9}$$

*with*

$$\mathrm{GNI}(\theta) := \frac{\mathbb{E}_B[\nabla L_B(\theta)^\top \mathcal{H}(\theta)\,\nabla L_B(\theta)]}{\|\nabla L(\theta)\|^2}.$$

*In particular, there exists $c > 0$ such that for all sufficiently small $\eta$: if*

$$\big|\mathrm{GNI}(\theta_t) - 2/\eta\big| \geq c\,\eta,$$

*then the sign of the expected loss increment satisfies*

$$\mathrm{sign}\,\mathbb{E}[L(\theta_{t+1}) - L(\theta_t) \,|\, \theta_t] = \mathrm{sign}\,\big(2/\eta - \mathrm{GNI}(\theta_t)\big).$$

*Proof of Proposition 2.* Fix $\theta$ and a mini–batch $B$. Write a third–order Taylor expansion of $L$ around $\theta$:

$$L(\theta - \eta\nabla L_B(\theta)) = L(\theta) - \eta\nabla L(\theta)^\top \nabla L_B(\theta) + \frac{\eta^2}{2}\,\nabla L_B(\theta)^\top \mathcal{H}(\theta)\,\nabla L_B(\theta) + R(\eta, \nabla L_B(\theta)),$$

where $\nabla L_B(\theta) := \nabla L_B(\theta)$ and $\mathcal{H}(\theta)$ is the full–batch Hessian. The third–order remainder satisfies the standard bound $|R(\eta, \nabla L_B(\theta))| \leq CL_2\eta^3\|\nabla L_B(\theta)\|^3$ for some numerical $C$, because the Hessian is $L_2$–Lipschitz.

Now take the expectation over $B \sim \mathcal{P}_b$. Using $\mathbb{E}_B[\nabla L_B(\theta)] = \nabla L(\theta)$ we obtain

$$\mathbb{E}_B\big[L(\theta - \eta\nabla L_B(\theta)) - L(\theta) \,|\, \theta\big] = -\eta\|\nabla L(\theta)\|^2 + \frac{\eta^2}{2}\,\mathbb{E}_B\big[\nabla L_B(\theta)^\top \mathcal{H}(\theta)\,\nabla L_B(\theta)\big] + \mathcal{O}(\eta^3),$$

which is (8).

Divide the right–hand side by $-\eta\|\nabla L(\theta)\|^2$ to get

$$\mathbb{E}_B[L(\theta_{t+1}) - L(\theta_t) \,|\, \theta_t] = -\eta\|\nabla L(\theta_t)\|^2\left(1 - \frac{\eta}{2}\,\mathrm{GNI}(\theta_t) + \mathcal{O}(\eta^2)\right),$$

since

$$\mathrm{GNI}(\theta_t) = \frac{\mathbb{E}_B[\nabla L_B(\theta)^\top \mathcal{H}(\theta_t)\,\nabla L_B(\theta)]}{\|\nabla L(\theta_t)\|^2}.$$

The last claim (sign agreement) follows immediately: for sufficiently small $\eta$, the $\mathcal{O}(\eta^2)$ term is dominated whenever $|\mathrm{GNI} - 2/\eta| \geq c\eta$ for a fixed constant $c > 0$. $\qquad\square$

# E  PROOF OF PROPOSITION 1

## E.1  SETUP AND NOTATION FOR PROPOSITION 1

Let

$$L(\theta) \;=\; \frac{1}{n}\sum_{i=1}^{n}\ell_i(\theta)$$

be three-times continuously differentiable, and let $\theta^\star$ be a (possibly non-isolated) local minimiser. Denote the full-batch Hessian at $\theta^\star$ by

$$\mathcal{H} \;:=\; \nabla^2 L(\theta^\star) \succeq 0.$$

For each sample $i$, define its (local) Hessian at $\theta^\star$,

$$\mathcal{H}_i \;:=\; \nabla^2\ell_i(\theta^\star),$$

so that $\mathcal{H} = \frac{1}{n}\sum_{i=1}^{n}\mathcal{H}_i$. We also define the (single-sample) gradient noise covariance at $\theta^\star$ as

$$\Sigma_g \;:=\; \mathbb{E}_i\big[\nabla\ell_i(\theta^\star)\,\nabla\ell_i(\theta^\star)^\top\big].$$

We decompose the parameter space as

$$\mathbb{R}^d \;=\; E_+ \oplus E_0, \qquad E_+ := \mathrm{Im}(\mathcal{H}), \quad E_0 := \ker(\mathcal{H}),$$

with associated orthogonal projectors $P_+$ and $P_0$. We will only require control of the dynamics in $E_+$ and assume that gradient noise in the flat subspace $E_0$ is not too large.

For each iteration $t$, a mini-batch $B_t$ of size $b$ is drawn (with or without replacement) and the SGD update is

$$\theta_{t+1} \;=\; \theta_t \;-\; \eta\,\nabla L_{B_t}(\theta_t), \qquad \nabla L_{B_t}(\theta) := \frac{1}{b}\sum_{i\in B_t}\nabla\ell_i(\theta).$$

Finally, define the Kronecker–sum operator

$$\mathcal{K} : \mathbb{R}^{d\times d} \to \mathbb{R}^{d\times d}, \qquad \mathcal{K}(X) := \mathcal{H}X + X\mathcal{H}.$$

On $E_+ \otimes E_+$, $\mathcal{K}$ is positive definite and has a Moore–Penrose pseudoinverse $\mathcal{K}^\dagger$.

We work under the following assumptions in a neighbourhood of $\theta^\star$.

**(A1) Local quadratic approximation.** Each $\ell_i$ is twice differentiable with $L_2$–Lipschitz Hessian near $\theta^\star$, and admits the Taylor expansion

$$\nabla\ell_i(\theta) \;=\; \nabla\ell_i(\theta^\star) \;+\; \mathcal{H}_i\,(\theta - \theta^\star) \;+\; R_i(\theta),$$

where $\|R_i(\theta)\| = \mathcal{O}(\|\theta - \theta^\star\|^2)$ uniformly in $i$.

**(A2) Compatible noise in flat directions.** The gradient noise covariance in the flat subspace is small:

$$\|P_0\,\Sigma_g\,P_0\| \;\lesssim\; \eta,$$

so that the iterates do not perform an unbounded random walk along $\ker(\mathcal{H})$.

**(A3) Linear stability of the linear dynamics.** The SGD on the quadratic approximation is linearly stable on $E_+$, i.e.

$$\rho\mathbb{E}\big[(I - \eta\,\mathcal{H}(L_B))^{\otimes 2}\big]_{|E_+} \;<\; 1.$$

*Remark* 1 (Remarks on the assumptions).

*Exact vs. Lipschitz Hessian (on (A1))*  When each $\ell_i$ is strictly quadratic, the local linearity

$$\nabla\ell_i(x) = \nabla\ell_i(x^\star) + \mathcal{H}_i(x - x^\star), \qquad \mathcal{H}_i := \nabla^2\ell_i(x^\star),$$

holds exactly, and $\mathcal{H} = \frac{1}{n}\sum_i \mathcal{H}_i$. In the general case, if $\nabla^2\ell_i$ is $L_2$–Lipschitz in a neighborhood of $x^\star$, a second–order Taylor expansion gives a remainder $\mathcal{O}(\|x - x^\star\|^2)$. For sufficiently small $\eta$, the SGD iterates typically remain in an $\mathcal{O}(\sqrt{\eta})$–neighborhood of $x^\star$, so these higher–order terms contribute only $\mathcal{O}(\eta^2)$ corrections in the discrete Lyapunov equation, which are dominated by the main $\mathcal{O}(\eta)$ term in Proposition 3.

*Small drift in flat directions (on (A2))*  The requirement $P_0\Sigma_g P_0 = 0$ can be relaxed to $\|P_0\Sigma_g P_0\| \le \delta$. A standard discrete–Lyapunov analysis shows that the stationary covariance $\Sigma_x$ remains finite provided $\delta = \mathcal{O}(\eta)$: roughly, if $\|P_0\Sigma_g P_0\|$ is at most a constant multiple of $\eta$ times the curvature scale on $E_+$, then the null–space covariance $\Sigma_x^{00}$ grows at most on the same $\mathcal{O}(\eta)$ scale as the covariance on $\mathrm{Im}(\mathcal{H})$. If instead $P_0\Sigma_g P_0$ is large, the dynamics executes an (uncontrolled) random walk along $E_0$, and no finite stationary covariance exists in those directions.

*Linear stability and spectral gap (on (A3))*  The condition

$$\rho\Big(\mathbb{E}\big[(I - \eta\,\mathcal{H}(L_B))^{\otimes 2}\big]_{|E_+}\Big) < 1$$

is the standard linear–stability condition for the second moment of SGD on a quadratic objective (cf. Ma & Ying (2021); Wu et al. (2018)). It ensures that the linearized error dynamics on $E_+$ is mean–square contractive and that the discrete Lyapunov equation

$$\Sigma_x = \mathbb{E}\big[(I - \eta\,\mathcal{H}(L_B))\,\Sigma_x\,(I - \eta\,\mathcal{H}(L_B))^\top\big] + \frac{\eta^2}{b}\,\Sigma_g$$

has a unique finite solution. As the spectral radius $\rho$ approaches $1$ from below, the spectral gap $1 - \rho$ controls both the mixing time and the size of the stationary covariance, with $\|\Sigma_x\| = \mathcal{O}((1-\rho)^{-1})$. In our use of Proposition 3 and Corollary 1 we implicitly assume that the step size $\eta$ is chosen so that the dynamics remains in this linearly stable regime on the time scales of interest, i.e. $\rho < 1$ (and often $\rho$ bounded away from 1), so that the $\mathcal{O}(\eta)$ expansion and the GNI $\approx 2/\eta$ law are accurate.

### E.2 FORMAL VERSION OF PROPOSITION 1

**Proposition 3** (Gradient–Noise Interaction at a stable stationary regime). *Assume (A1) and (A2) above, and run mini-batch SGD with fixed batch size $b$ and fixed step size $\eta$ satisfying the linear stability condition (A3).*

*Then the linearised error process $\Delta_t := \theta_t - \theta^\star$ admits a unique stationary covariance matrix $\Sigma_x$ on $E_+$, given by*

$$\Sigma_x \;=\; \frac{\eta}{b}\,\mathcal{K}^\dagger(\Sigma_g) \;+\; \mathcal{O}(\eta^2), \tag{10}$$

*where the $\mathcal{O}(\eta^2)$ term depends only on $L_2$, $\|\Sigma_g\|$ and $(\lambda_{\min}^+)^{-1}$.*

*Moreover, if $\theta \sim \pi$ is distributed according to this stationary law, and $B$ is an independent fresh mini-batch of size $b$, then*

$$\frac{\mathbb{E}_{\theta\sim\pi}\,\mathbb{E}_B\big[\nabla L_B(\theta)^\top\,\mathcal{H}\,\nabla L_B(\theta)\big]}{\mathbb{E}_{\theta\sim\pi}\big[\|\nabla L(\theta)\|^2\big]} \;=\; \frac{2}{\eta}\,\big(1+\mathcal{O}(\eta)\big). \tag{11}$$

*In particular, to leading order in $\eta$, the Gradient–Noise Interaction*

$$\mathrm{GNI}(\theta) \;:=\; \frac{\mathbb{E}_B\big[\nabla L_B(\theta)^\top\,\mathcal{H}\,\nabla L_B(\theta)\big]}{\|\nabla L(\theta)\|^2}$$

*is centred at $2/\eta$ under the (linearly) stable stationary distribution, and this leading behaviour is independent of the individual Hessians $\{\mathcal{H}_i\}_i$, depending on them only through $\Sigma_g$ and $\mathcal{H}$.*

**Corollary 1** (Stable $\eta$–changes induce only bounded loss jumps in the Type-1 regime). *Assume (A1)–(A3) and let $\eta_0, \eta_1 > 0$ be two step sizes such that the linearized SGD dynamics on the quadratic approximation is linearly stable on $E_+$ for both $\eta_0$ and $\eta_1$:*

$$\rho\Big(\mathbb{E}\big[(I - \eta_k\,\mathcal{H}(L_B))^{\otimes 2}\big]_{|E_+}\Big) \;<\; 1, \qquad k \in \{0,1\}.$$

*Consider the SGD trajectory $(\theta_t)$ that is run with step size $\eta_0$ up to some time $T$, and with step size $\eta_1$ for all $t \geq T$.*

*Then:*

(i) *For each $k \in \{0,1\}$ the linearized error process $\Delta_t^{(\eta_k)} := \theta_t - \theta^\star$ admits a unique stationary covariance $\Sigma_x(\eta_k)$ on $E_+$ satisfying*

$$\Sigma_x(\eta_k) \;=\; \frac{\eta_k}{b}\,\mathcal{K}^\dagger(\Sigma_g) \;+\; \mathcal{O}(\eta_k^2),$$

*as in Proposition 3.*

(ii) *Let $A_1 := \mathbb{E}[(I - \eta_1\mathcal{H}(L_B))^{\otimes 2}]_{|E_+}$ and $\rho_1 := \rho(A_1) < 1$. There exist constants $C_1, C_2 < \infty$, depending only on $\Sigma_g$ and $\rho_1$ (but not on $T$), such that for all $s \geq 0$,*

$$\mathbb{E}\big[\|\theta_{T+s} - \theta^\star\|^2\big] \;\leq\; \frac{C_1}{1 - \rho_1}, \qquad \mathbb{E}\big[L(\theta_{T+s}) - L(\theta^\star)\big] \;\leq\; \frac{C_2}{1 - \rho_1}.$$

*In particular, after switching from $\eta_0$ to $\eta_1$ the loss trajectory remains uniformly bounded and converges to the finite stationary level*

$$L_\infty(\eta_1) \;:=\; \mathbb{E}_{\theta\sim\pi_{\eta_1}}\big[L(\theta) - L(\theta^\star)\big] \;=\; \frac{1}{2}\,\mathrm{tr}\big(\mathcal{H}\,\Sigma_x(\eta_1)\,\mathcal{H}\big),$$

*which itself is of order $\mathcal{O}((1-\rho_1)^{-1})$.*

*Thus, in the Type-1 (noise-driven) regime, any change of step size that preserves linear stability (A3) can produce at most a finite "jump" in the expected loss, of the same order as the new stationary level $L_\infty(\eta_1)$, but cannot generate catapult-like divergence.*

### E.3 PROOF OF PROPOSITION 3

*Proof of Proposition 3.* We proceed in four steps. Throughout the proof we work on the subspace $E_+ = \text{Im}(\mathcal{H})$; all covariances and operators are implicitly restricted to $E_+$ (the flat subspace $E_0$ is controlled by Assumption **(A2)** and does not contribute to the quantities involving $\mathcal{H}$).

**Step 1: Linearised dynamics on the quadratic approximation.**

By Assumption **(A1)**, near $\theta^\star$ each per-sample loss $\ell_i$ admits the expansion

$$\nabla \ell_i(\theta) = \nabla \ell_i(\theta^\star) + \mathcal{H}_i(\theta - \theta^\star) + R_i(\theta),$$

where $\|R_i(\theta)\| = \mathcal{O}(\|\theta - \theta^\star\|^2)$ uniformly in $i$. Let us denote the single-sample gradient at $\theta^\star$ by

$$g_i := \nabla \ell_i(\theta^\star),$$

so that $\Sigma_g = \mathbb{E}_i[g_i g_i^\top]$.

For each iteration $t$, a mini-batch $B_t$ of size $b$ is drawn and the SGD update is

$$\theta_{t+1} = \theta_t - \eta \nabla L_{B_t}(\theta_t), \qquad \nabla L_{B_t}(\theta) := \frac{1}{b} \sum_{i \in B_t} \nabla \ell_i(\theta).$$

Define the error vector

$$\Delta_t := \theta_t - \theta^\star.$$

Then

$$\nabla L_{B_t}(\theta_t) = \frac{1}{b} \sum_{i \in B_t} \Big[ g_i + \mathcal{H}_i \Delta_t + R_i(\theta_t) \Big]$$

$$=: \underbrace{\xi_t}_{\text{zero mean}} + \underbrace{\mathcal{H}_{B_t}}_{\text{batch Hessian}} \Delta_t + r_t,$$

where

$$\xi_t := \frac{1}{b} \sum_{i \in B_t} g_i, \qquad \mathcal{H}_{B_t} := \frac{1}{b} \sum_{i \in B_t} \mathcal{H}_i, \qquad r_t := \frac{1}{b} \sum_{i \in B_t} R_i(\theta_t).$$

By construction we have

$$\mathbb{E}[\xi_t] = 0, \qquad \mathbb{E}[\xi_t \xi_t^\top] = \frac{1}{b} \Sigma_g,$$

and (using $\nabla L(\theta^\star) = 0$)

$$\mathbb{E}[\mathcal{H}_{B_t}] = \mathcal{H}.$$

The exact SGD recursion can therefore be written as

$$\Delta_{t+1} = \Delta_t - \eta \nabla L_{B_t}(\theta_t) = (I - \eta \mathcal{H}_{B_t})\Delta_t - \eta \xi_t - \eta r_t. \tag{12}$$

For the purposes of the leading-order analysis, it is convenient to first *ignore* the nonlinear remainders $r_t$ and consider the purely linearised dynamics on the quadratic approximation (i.e. we replace each $\ell_i$ by its quadratic Taylor polynomial at $\theta^\star$). On this quadratic model we have $R_i \equiv 0$, hence $r_t \equiv 0$, and (12) becomes

$$\Delta_{t+1} = C_t \Delta_t - \eta \xi_t, \qquad C_t := I - \eta \mathcal{H}_{B_t}. \tag{13}$$

Note that in this linearised model the random matrices $C_t$ and the noise vectors $\xi_t$ are independent of $\Delta_t$ (they depend only on the batch $B_t$ and the fixed Hessians $\{\mathcal{H}_i\}_i$).

We will first solve the covariance structure of the linear recursion (13), and then argue that restoring the remainder $r_t$ only introduces $\mathcal{O}(\eta^2)$ corrections.

**Step 2: Discrete Lyapunov equation and existence of a stationary covariance.**

Let $\Sigma_t := \mathbb{E}[\Delta_t \Delta_t^\top]$ denote the covariance of $\Delta_t$ under the linear recursion (13). Using independence of $C_t$ and $\xi_t$ from $\Delta_t$, we compute

$$\Sigma_{t+1} = \mathbb{E}\big[\Delta_{t+1}\Delta_{t+1}^\top\big]$$

$$= \mathbb{E}\Big[(C_t \Delta_t - \eta \xi_t)(C_t \Delta_t - \eta \xi_t)^\top\Big]$$

$$= \mathbb{E}\big[C_t \Delta_t \Delta_t^\top C_t^\top\big] - \eta \mathbb{E}\big[C_t \Delta_t \xi_t^\top\big] - \eta \mathbb{E}\big[\xi_t \Delta_t^\top C_t^\top\big] + \eta^2 \mathbb{E}\big[\xi_t \xi_t^\top\big].$$

Conditioning on $\Delta_t$ and using $\mathbb{E}[\xi_t \mid \Delta_t] = 0$, the two cross-terms vanish:

$$\mathbb{E}\big[C_t \Delta_t \xi_t^\top\big] = \mathbb{E}\big[C_t \Delta_t \mathbb{E}[\xi_t^\top \mid \Delta_t]\big] = 0, \quad \mathbb{E}\big[\xi_t \Delta_t^\top C_t^\top\big] = 0.$$

Thus we obtain

$$\Sigma_{t+1} = \mathbb{E}\big[C_t \Sigma_t C_t^\top\big] + \eta^2 \frac{1}{b} \Sigma_g. \tag{14}$$

Assuming that the linear recursion admits a stationary distribution on $E_+$, we denote the stationary covariance by

$$\Sigma_x := \lim_{t \to \infty} \Sigma_t$$

and it must satisfy the discrete Lyapunov equation

$$\Sigma_x = \mathbb{E}\big[C_t \Sigma_x C_t^\top\big] + \eta^2 \frac{1}{b} \Sigma_g. \tag{15}$$

To show that such a $\Sigma_x$ exists and is unique on $E_+$, we vectorise (15). Recall that for any matrices $A, X, B$ of compatible dimensions we have

$$\mathrm{vec}(AXB) = (B^\top \otimes A)\,\mathrm{vec}(X),$$

where $\otimes$ denotes the Kronecker product. Applying this to $C_t \Sigma_x C_t^\top$ we get

$$\mathrm{vec}\big(C_t \Sigma_x C_t^\top\big) = (C_t \otimes C_t)\,\mathrm{vec}(\Sigma_x).$$

Taking expectations in (15) and using linearity of $\mathrm{vec}(\cdot)$ we obtain

$$\mathrm{vec}(\Sigma_x) = T\,\mathrm{vec}(\Sigma_x) + \eta^2 \frac{1}{b}\,\mathrm{vec}(\Sigma_g), \qquad T := \mathbb{E}[C_t \otimes C_t]. \tag{16}$$

Rearranging gives

$$\big(I - T\big)\,\mathrm{vec}(\Sigma_x) = \eta^2 \frac{1}{b}\,\mathrm{vec}(\Sigma_g). \tag{17}$$

Assumption **(A3)** states that the linearised SGD on the quadratic approximation is linearly stable on $E_+$, i.e.

$$\rho\Big(\mathbb{E}\big[(I - \eta \mathcal{H}(L_B))^{\otimes 2}\big]_{|E_+}\Big) < 1.$$

In our notation this means precisely that the restriction of $T$ to $E_+ \otimes E_+$ has spectral radius strictly less than 1. Hence $I - T$ is invertible on $E_+ \otimes E_+$ and the vector equation (17) has a unique solution there, which corresponds to the unique stationary covariance $\Sigma_x$ on $E_+$.

**Step 3: Small-stepsize expansion and explicit form of $\Sigma_x$.**

We now compute the leading behaviour of $\Sigma_x$ as a function of $\eta$ for small $\eta$. Recall that

$$C_t = I - \eta \mathcal{H}_{B_t}, \qquad \mathcal{H}_{B_t} = \frac{1}{b} \sum_{i \in B_t} \mathcal{H}_i.$$

Thus

$$C_t \otimes C_t = (I - \eta \mathcal{H}_{B_t}) \otimes (I - \eta \mathcal{H}_{B_t}) = I \otimes I - \eta(\mathcal{H}_{B_t} \otimes I + I \otimes \mathcal{H}_{B_t}) + \eta^2 (\mathcal{H}_{B_t} \otimes \mathcal{H}_{B_t}).$$

Taking expectations and using $\mathbb{E}[\mathcal{H}_{B_t}] = \mathcal{H}$, we obtain

$$T = I - \eta K + \eta^2 M, \tag{18}$$

where $K$ is the Kronecker-sum operator

$$K := \mathcal{H} \otimes I + I \otimes \mathcal{H},$$

and $M$ is the operator defined by

$$M := \mathbb{E}[\mathcal{H}_{B_t} \otimes \mathcal{H}_{B_t}].$$

Substituting (18) into (17) gives

$$\big(I - T\big)\mathrm{vec}(\Sigma_x) = \big(\eta K - \eta^2 M\big)\mathrm{vec}(\Sigma_x) = \eta^2 \frac{1}{b}\,\mathrm{vec}(\Sigma_g).$$

On $E_+ \otimes E_+$ the operator $K$ is positive definite (its eigenvalues are $\lambda_i + \lambda_j$ where $\lambda_i, \lambda_j > 0$ are eigenvalues of $\mathcal{H}$), hence invertible. Restricting to $E_+ \otimes E_+$, we can rewrite this as

$$\left(K - \eta M\right)\text{vec}(\Sigma_x) = \eta \frac{1}{b}\, \text{vec}(\Sigma_g). \tag{19}$$

For sufficiently small $\eta$, the operator $K - \eta M$ remains invertible and admits a Neumann-series expansion of its inverse. More precisely, on $E_+ \otimes E_+$ we have

$$(K - \eta M)^{-1} = K^{-1} + \eta K^{-1} M K^{-1} + \mathcal{O}(\eta^2),$$

where the $\mathcal{O}(\eta^2)$ term is understood in operator norm (depending on $\|K^{-1}\|$ and $\|M\|$). Applying this to (19) we obtain

$$\text{vec}(\Sigma_x) = (K - \eta M)^{-1} \left(\eta \frac{1}{b}\, \text{vec}(\Sigma_g)\right)$$

$$= \eta \frac{1}{b}\left(K^{-1} + \eta K^{-1} M K^{-1} + \mathcal{O}(\eta^2)\right) \text{vec}(\Sigma_g)$$

$$= \eta \frac{1}{b} K^{-1} \text{vec}(\Sigma_g) + \mathcal{O}(\eta^2).$$

Rewriting in matrix form and denoting by $\mathcal{K}(X) = \mathcal{H}X + X\mathcal{H}$ the corresponding Kronecker-sum operator on matrices, this says that on $E_+$

$$\Sigma_x = \frac{\eta}{b}\, \mathcal{K}^\dagger(\Sigma_g) + \mathcal{O}(\eta^2),$$

where $\mathcal{K}^\dagger$ is the Moore–Penrose inverse of $\mathcal{K}$ on $E_+ \otimes E_+$. This proves (10) in the statement of the proposition for the quadratic model.

The effect of the nonlinear remainders $r_t$ can be treated as follows: by Assumption **(A1)**, $\|r_t\| = \mathcal{O}(\|\Delta_t\|^2)$, and the stationary covariance $\Sigma_x$ of the linear system is $\mathcal{O}(\eta)$, so typical $\|\Delta_t\|^2$ is $\mathcal{O}(\eta)$ and the additive perturbation $-\eta r_t$ to the dynamics has magnitude $\mathcal{O}(\eta^2)$. Its contribution to the noise covariance in the Lyapunov equation is thus $\mathcal{O}(\eta^4)$, which in turn produces a $\mathcal{O}(\eta^2)$ perturbation to $\Sigma_x$. Therefore the formula (10) remains valid up to an $\mathcal{O}(\eta^2)$ error for the true (non-quadratic) dynamics.

**Step 4: Gradient–Noise Interaction ratio.**

We now compute the ratio in (11). Fix $\theta = \theta^\star + \Delta$ and consider a fresh mini-batch $B$. On the quadratic approximation we have

$$\nabla L(\theta) = \mathcal{H}\Delta,$$

and

$$\nabla L_B(\theta) = \frac{1}{b} \sum_{i \in B} (g_i + \mathcal{H}_i \Delta) = \underbrace{\xi_B}_{\text{zero mean}} + \underbrace{\mathcal{H}_B}_{\text{batch Hessian}} \Delta,$$

where $\xi_B := \frac{1}{b} \sum_{i \in B} g_i$ and $\mathcal{H}_B := \frac{1}{b} \sum_{i \in B} \mathcal{H}_i$. Conditioning on $\Delta$ and using $\mathbb{E}_B[\xi_B \mid \Delta] = 0$, we get

$$\mathbb{E}_B\left[\nabla L_B(\theta)^\top \mathcal{H} \nabla L_B(\theta) \,\big|\, \Delta\right] = \mathbb{E}_B\left[(\mathcal{H}_B \Delta + \xi_B)^\top \mathcal{H}(\mathcal{H}_B \Delta + \xi_B) \,\big|\, \Delta\right]$$

$$= \Delta^\top \mathbb{E}_B\left[\mathcal{H}_B^\top \mathcal{H} \mathcal{H}_B\right]\Delta + \mathbb{E}_B\left[\xi_B^\top \mathcal{H}\, \xi_B\right] \tag{20}$$

(the cross term vanishes because $\mathbb{E}_B[\xi_B \mid \Delta] = 0$). Taking expectation over $\theta \sim \pi$ (the stationary law of the linear system) and recalling $\Sigma_x = \mathbb{E}_\pi[\Delta\Delta^\top]$ and $\mathbb{E}_B[\xi_B \xi_B^\top] = \frac{1}{b}\Sigma_g$, we obtain

$$N := \mathbb{E}_{\theta \sim \pi} \mathbb{E}_B\left[\nabla L_B(\theta)^\top \mathcal{H} \nabla L_B(\theta)\right]$$

$$= \text{tr}\left(\mathcal{H}\Sigma_g\right)\frac{1}{b} + \text{tr}\left(\mathbb{E}_B[\mathcal{H}_B^\top \mathcal{H} \mathcal{H}_B]\Sigma_x\right). \tag{21}$$

Similarly, the denominator is

$$D := \mathbb{E}_{\theta \sim \pi}\left[\|\nabla L(\theta)\|^2\right] = \mathbb{E}_{\theta \sim \pi}\left[\Delta^\top \mathcal{H}^2 \Delta\right] = \text{tr}\left(\mathcal{H}^2 \Sigma_x\right). \tag{22}$$

To relate $N$ and $D$, we use the Lyapunov equation (15) for the quadratic model. Expanding the right-hand side of (15), we have

$$\mathbb{E}\left[C_t \Sigma_x C_t^\top\right] = \mathbb{E}\left[(I - \eta\mathcal{H}_{B_t})\Sigma_x(I - \eta\mathcal{H}_{B_t})^\top\right]$$
$$= \Sigma_x - \eta\mathbb{E}\left[\mathcal{H}_{B_t}\Sigma_x\right] - \eta\mathbb{E}\left[\Sigma_x\mathcal{H}_{B_t}\right] + \eta^2\mathbb{E}\left[\mathcal{H}_{B_t}\Sigma_x\mathcal{H}_{B_t}\right].$$

Using $\mathbb{E}[\mathcal{H}_{B_t}] = \mathcal{H}$, this simplifies to

$$\mathbb{E}\left[C_t \Sigma_x C_t^\top\right] = \Sigma_x - \eta(\mathcal{H}\Sigma_x + \Sigma_x\mathcal{H}) + \eta^2\mathbb{E}\left[\mathcal{H}_{B_t}\Sigma_x\mathcal{H}_{B_t}\right].$$

Substituting this back into (15) and cancelling $\Sigma_x$ on both sides, we obtain

$$0 = -\eta(\mathcal{H}\Sigma_x + \Sigma_x\mathcal{H}) + \eta^2\mathbb{E}\left[\mathcal{H}_{B_t}\Sigma_x\mathcal{H}_{B_t}\right] + \eta^2\frac{1}{b}\Sigma_g.$$

Dividing by $\eta$ yields the identity

$$\mathcal{H}\Sigma_x + \Sigma_x\mathcal{H} = \eta\mathbb{E}\left[\mathcal{H}_{B_t}\Sigma_x\mathcal{H}_{B_t}\right] + \eta\frac{1}{b}\Sigma_g. \tag{23}$$

Now multiply both sides of (23) on the left by $\mathcal{H}$ and take traces. Using the cyclicity of the trace and the fact that $\mathcal{H}$ is symmetric, we get

$$\mathrm{tr}\left(\mathcal{H}(\mathcal{H}\Sigma_x + \Sigma_x\mathcal{H})\right) = \mathrm{tr}\left(\mathcal{H}^2\Sigma_x\right) + \mathrm{tr}\left(\mathcal{H}\Sigma_x\mathcal{H}\right)$$
$$= 2\,\mathrm{tr}\left(\mathcal{H}^2\Sigma_x\right) = 2D, \tag{24}$$

and

$$\mathrm{tr}\left(\mathcal{H}\left(\eta\mathbb{E}[\mathcal{H}_{B_t}\Sigma_x\mathcal{H}_{B_t}] + \eta\frac{1}{b}\Sigma_g\right)\right) = \eta\,\mathrm{tr}\left(\mathcal{H}\mathbb{E}[\mathcal{H}_{B_t}\Sigma_x\mathcal{H}_{B_t}]\right) + \eta\frac{1}{b}\mathrm{tr}(\mathcal{H}\Sigma_g). \tag{25}$$

Equating (24) and (25) (they come from the two sides of (23)) gives

$$2D = \eta\,\mathrm{tr}\left(\mathcal{H}\mathbb{E}[\mathcal{H}_{B_t}\Sigma_x\mathcal{H}_{B_t}]\right) + \eta\frac{1}{b}\mathrm{tr}(\mathcal{H}\Sigma_g). \tag{26}$$

Comparing (21) and (26), we see that

$$N = \frac{1}{b}\mathrm{tr}(\mathcal{H}\Sigma_g) + \mathrm{tr}\left(\mathbb{E}[\mathcal{H}_{B_t}^\top\mathcal{H}\,\mathcal{H}_{B_t}]\Sigma_x\right) = \frac{2}{\eta}D.$$

Therefore, on the quadratic approximation we have the *exact* identity

$$\frac{N}{D} = \frac{2}{\eta}. \tag{27}$$

Restoring the higher-order terms $R_i(\theta)$ from Assumption **(A1)** introduces an additional third-order remainder in the Taylor expansion of the loss over one SGD step. For a Lipschitz Hessian with constant $L_2$, standard Taylor estimates give a per-step remainder of order $\mathcal{O}(\eta^3\|\nabla L_B(\theta_t)\|^3)$, which is $\mathcal{O}(\eta^3)$ in expectation under the stationary law (since the stationary covariance scales like $\mathcal{O}(\eta)$ by (10)). This adds a term of order $\mathcal{O}(\eta^3)$ to the stationarity condition $\mathbb{E}[L(\theta_{t+1}) - L(\theta_t)] = 0$, and hence contributes only a factor $\mathcal{O}(\eta)$ to the ratio $N/D$. Thus

$$\frac{N}{D} = \frac{2}{\eta}\left(1 + \mathcal{O}(\eta)\right),$$

which is precisely (11). This completes the proof. $\qquad\square$

## F ON THE TWO TYPES OF OSCILLATIONS IN NNs

**Differentiating Oscillations in Neural Network Optimization** Our analytical treatment of SGD on one-dimensional quadratic objectives in Appendix D.1 leverages the simplicity of having a single curvature measure–the second derivative–which facilitates a precise landscape characterization and explicit stability conditions. However, extending this analysis to multidimensional quadratics

already introduces significantly more intricate dynamics, necessitating advanced analytical frameworks as developed by (Wu et al., 2018; Ma & Ying, 2021; Mulayoff & Michaeli, 2024). Transitioning further to neural network optimization increases this complexity dramatically, since training predominantly occurs away from the manifold of minima, including the EoS-like instabilities themselves (as evidenced by the continuous reduction in loss)—and therefore requires to go beyond linear stability of quadratics near the manifold of minima.

Given the current absence of robust theoretical tools to comprehensively analyze such dynamics, distinguishing between curvature-driven and noise-driven oscillations necessitates empirical experimentation. Specifically, we probe the dynamics by systematically varying hyperparameters (e.g., step size or batch size), as illustrated in Figure 3, allowing us to differentiate curvature-induced (Type-2) oscillations from purely noise-induced (Type-1) oscillations (Figure 5).

**Type-2 Oscillations Are Unique to NN Optimization** This complexity inherent in neural network optimization is not merely an analytical inconvenience; rather, it is intrinsically tied to the emergence and significance of Type-2 oscillations and EoS-style phenomena. Notably, Type-2 oscillations emerge naturally[9] only in the case of neural network optimization, but not in the case of quadratic objectives. In the one-dimensional quadratic scenario analyzed previously, curvature-driven oscillations require the step size to precisely match the stability threshold $2/\lambda_{\max}$, or exceed it, in which case we have divergence—in either case, it means that optimization of quadratics does naturally enter a regime of instability. In contrast, neural network optimization uniquely exhibits *progressive sharpening*, a third-order derivative phenomenon (Damian et al., 2023), where curvature naturally increases during training. This progressive increase in curvature means that training with a fixed step size can transition into an EoS-like regime of instability without any explicit adjustment of the hyperparameters, and stay there due to self-stabilization effects (Damian et al., 2023). Hence, Type-2 oscillations emerge naturally and robustly within neural network training dynamics due to this intrinsic change of the loss landscape. Consequently, Type-2 oscillations and EoS-like regimes are fundamentally driven by progressive sharpening, which does not happen in quadratics, making it a purely neural network optimization phenomena.

## F.1 On the Importance of Type-2 Oscillations Compared to Type-1

Noise-induced (Type-1) oscillations are not unstable when introducing slight perturbations (increase step size or decrease batch size), as showcased in Figure 9 and 10. Therefore, they do not constitute an EoS-type phenomena, where slight perturbations do cause divergence ("complete" divergence as long as we consider just the quadratic terms and can ignore higher terms — the fact that it doesn't fully diverge is exactly the higher-terms effect). Instead, after a perturbation, noise-induced oscillations quickly re-stabilize at a higher level.

Crucially, a lack of such divergence means that noise-induced oscillations wouldn't exhibit the self-stabilization mechanism of Damian et al. (2023) characteristic of EoS (differing it from classical convex optimization). Moreover, as shown in the quadratic example and in the proofs, noise-induced oscillations happen for any quadratic, for a wide range of step sizes, making them inherently "unsurprising", while EoS is a beyond-quadratic phenomena (and, as far as we know, a deep-learning-specific phenomena), as it relies on both progressive sharpening and the aforementioned self-stabilization, both being an effects of higher order terms. And the reason why we care specifically about effects of beyond-quadratic terms is specifically the adaptation of the landscape to the hyper-parameters, which is, by definition, an effect of higher order terms. That is the reason we specifically care about curvature-driven oscillations.

Now, with all of the above, GNI, being an indicator of those noise-induced oscillations, is therefore not an indicator of EoS-like regime. This is despite the fact that GNI in SGD comes from the same place as $\lambda_{\max}$/Rayleigh quotient in GD — i.e. from the descent lemma; yet, it does not mean that the two quantities serve the same role. Instead, it is the presence of the natural noise in SGD that makes the analysis much more complex. Instead, GNI has its usefulness as a measure of the level of noise coming from SGD. That is, noise-induced oscillations are influenced by the Hessian, but are also strongly influenced by the ratio between the noise covariance and the norm of full batch

---

[9]We define an as emerging *naturally* if it arises inherently from the training dynamics, and not a result of precisely-selected hyperparameters or initializations, reflecting a fundamental characteristic of the optimization process itself. Formally, it needs to happen over a range of hyperparameter choices and initializations.

gradient, with the latter being the leading cause of change. In particular, GNI is decoupled from the Hessian, and can change drastically without any change of landscape sharpness, as showcased in our experiments. Lastly, another important consequence of EoS is that the landscape adapts to the hyper-parameters (rather than the other way around in classical optimization). With GNI being decoupled from the Hessian, GNI being at 2/eta is not an indication of landscape adopting to the hyper-parameters, as is the case with $\lambda_{\max}$ being at $2/\eta$ during GD.

# G GRADIENTS EXPLODE ABOVE THE EoSS: PROOF OF THEOREM 1

## G.1 PART 1: EXPLOSION

**Setting.** For each minibatch index $i$, let $L_i : \mathbb{R}^d \to \mathbb{R}$ be twice differentiable with (possibly index–dependent) positive semidefinite Hessian $H_i$ that is *constant in $\theta$* (quadratic model). Write

$$Y_i(\theta) := \nabla L_i(\theta) = H_i(\theta - x_i), \qquad \| \cdot \| = \| \cdot \|_2.$$

At iteration $t$, stochastic gradient descent (SGD) draws $j_t \sim \mathcal{P}_b$ independently of the past and performs

$$\theta_{t+1} = \theta_t - \eta\, Y_{j_t}(\theta_t), \qquad \eta > 0.$$

We define basic statistics evaluated at $\theta_t$:

$$r_i(\theta_t) := \frac{Y_i(\theta_t)^\top H_i Y_i(\theta_t)}{\|Y_i(\theta_t)\|^2} \; \in [0, \infty).$$

Note that *Batch Sharpness* is

$$Batch\ Sharpness(\theta) := \mathbb{E}_i\big[r_i(\theta)\big].$$

All randomness lives on a probability space $(\Omega, \mathcal{F}, \mathbb{P})$. At each step $t \geq 0$, SGD draws $j_t \sim \mathcal{P}_b$ i.i.d. and independent of the $\sigma$–algebra of the past

$$\mathcal{F}_t := \sigma\big(\theta_0, j_0, \ldots, j_{t-1}\big).$$

We assume

$$\Lambda := \operatorname*{ess\,sup}_{i \sim \mathcal{P}_b} \|H_i\| < \infty \quad \text{and} \quad \mathbb{E}_{i \sim \mathcal{P}_b}\big[\|Y_i(\theta)\|^2\big] < \infty \text{ for all } \theta \text{ reached by SGD}.$$

(These ensure that the Rayleigh quotients below are well defined and integrable.) We adopt the convention that, for any $i$ and $\theta$ with $Y_i(\theta) = 0$,

$$\frac{Y_i(\theta)^\top H_i Y_i(\theta)}{\|Y_i(\theta)\|^2} := 0, \qquad \frac{\|Y_i(\theta - \eta Y_i(\theta))\|^2}{\|Y_i(\theta)\|^2} := 1.$$

We provide a rigorous instability statement: a multiplicative explosion controlled solely by *Batch Sharpness*.

**Proposition 4** (On–batch multiplicative explosion under *Batch Sharpness*$> 2/\eta$ for quadratics.)**.** *For each $t$ define the on–batch factor*

$$R_t := \mathbb{E}_{j_t}\left[ \frac{\|Y_{j_t}(\theta_{t+1})\|^2}{\|Y_{j_t}(\theta_t)\|^2} \;\Big|\; \mathcal{F}_t \right].$$

*Then, for all $t$,*

$$R_t \;\geq\; \Big(1 - \eta\, Batch\ Sharpness(\theta_t)\Big)^2.$$

*Consequently, if there exist indices $t_0, \ldots, t_0 + T - 1$ and a constant $\epsilon > 0$ such that*

$$Batch\ Sharpness(\theta_t) \;\geq\; \frac{2 + \epsilon}{\eta} \qquad \text{for all } t_0 \leq t \leq t_0 + T - 1,$$

*then*

$$\mathbb{E}\left[ \prod_{s=t_0}^{t_0+T-1} \frac{\|Y_{j_s}(\theta_{s+1})\|^2}{\|Y_{j_s}(\theta_s)\|^2} \right] \;\geq\; (1 + \epsilon)^{2T},$$

*i.e., the product of on–batch factors grows exponentially in expectation.*

**Beyond quadratic.** Beyond the quadratic model: with $\theta$–dependent Hessians, the identity $Y_i(\theta_{t+1}) = (I - \eta H_i)Y_i(\theta_t)$ incurs Taylor remainders of order $O(\eta\|Y_i\|\,\|\nabla H_i\|)$; standard Lipschitz–Hessian assumptions then yield perturbative versions of Proposition 4 with additional $O(\eta^3)$ terms. We do not invoke these here to keep the statements exact.

**Proof of Proposition 4.** Fix $t$ and condition on $j_t = i$. By the quadratic model,
$$Y_i(\theta_{t+1}) = H_i\big(\theta_t - \eta Y_i(\theta_t) - x_i\big) = (I - \eta H_i)\,Y_i(\theta_t).$$
Hence
$$\frac{\|Y_i(\theta_{t+1})\|^2}{\|Y_i(\theta_t)\|^2} = 1 - 2\eta\,r_i(\theta_t) + \eta^2\,q_i(\theta_t), \qquad q_i(\theta_t) := \frac{Y_i(\theta_t)^\top H_i^2 Y_i(\theta_t)}{\|Y_i(\theta_t)\|^2}.$$
By Cauchy–Schwarz in the $H_i$–inner product, $q_i(\theta_t) \geq r_i(\theta_t)^2$. Therefore, pointwise in $i$,
$$\frac{\|Y_i(\theta_{t+1})\|^2}{\|Y_i(\theta_t)\|^2} \;\geq\; \big(1 - \eta\,r_i(\theta_t)\big)^2.$$
Averaging over $j_t$ and applying Jensen's inequality to the convex map $x \mapsto (1 - \eta x)^2$ yields
$$R_t \;\geq\; \mathbb{E}_i\big[(1 - \eta r_i(\theta_t))^2\big] \;\geq\; \big(1 - \eta\,\mathbb{E}_i[r_i(\theta_t)]\big)^2 = \big(1 - \eta\,Batch\ Sharpness(\theta_t)\big)^2.$$
If $Batch\ Sharpness(\theta_t) \geq (2 + \epsilon)/\eta$ for each $t \in [t_0, t_0 + T - 1]$, then $R_t \geq (1 + \epsilon)^2 > 1 + 2\epsilon$ deterministically given $\mathcal{F}_t$.

Now, let
$$Z_s \;:=\; \frac{\|Y_{j_s}(\theta_{s+1})\|^2}{\|Y_{j_s}(\theta_s)\|^2}, \qquad \mathcal{F}_s \;:=\; \sigma(\theta_0, j_0, \ldots, j_{s-1}), \qquad U_t \;:=\; \prod_{r=t_0}^{t_0+t-1} Z_r \;\;(U_0 := 1).$$
From the single–step bound proved above we have, for each $s$,
$$\mathbb{E}[\,Z_s \mid \mathcal{F}_s\,] \;\geq\; \big(1 - \eta\,Batch\ Sharpness(\theta_s)\big)^2 \;\geq\; (1 + \epsilon)^2.$$
By the tower property,
$$\mathbb{E}[U_{t+1}] \;=\; \mathbb{E}\big[\,U_t\,Z_{t_0+t}\,\big] \;=\; \mathbb{E}\big[\,U_t\,\mathbb{E}[Z_{t_0+t} \mid \mathcal{F}_{t_0+t}]\,\big] \;\geq\; \mathbb{E}\big[\,U_t\,\gamma_{t_0+t}\,\big].$$
$S(\theta_s) \geq 2/\eta + \varepsilon\,\eta$ (with $\varepsilon > 0$), then $\gamma_s = (1 - \eta S(\theta_s))^2 \geq (1 + \varepsilon\eta^2)^2 =: \gamma^2$, hence
$$\mathbb{E}[U_{t+1}] \;\geq\; \gamma^2\,\mathbb{E}[U_t] \quad\Longrightarrow\quad \mathbb{E}[U_T] \;\geq\; \gamma^{2T} \;=\; \big(1 + \epsilon\big)^{2T}.$$
we obtain the stated exponential lower bound. $\qquad\square$

**Discussion after Proposition 4.** The proof uses only: (i) the exact *closure* $Y_i(\theta_{t+1}) = (I - \eta H_i)Y_i(\theta_t)$ (quadratic model), (ii) Cauchy–Schwarz ($q_i \geq r_i^2$), and (iii) Jensen's inequality. No cross–batch interaction is needed; hence the result holds assumption–free, and it naturally yields a multiplicative, path–wide instability witness.

### G.2 Part 2: SGD with Replacement

In Proposition 4 we showed that if *Batch Sharpness* is bigger than $2/\eta$ for prolonged time a certain quantity explodes exponentially fast. What is missing is to show that if *Batch Sharpness* is bigger than $2/\eta$ on the quadratic setting at one step $t_0$, then it will be also at the following time steps.

**Lipschitz drift of batch sharpness in the quadratic model.** Fix a minibatch $i$ and write $r_i(\theta) := \frac{Y_i(\theta)^\top H_i Y_i(\theta)}{\|Y_i(\theta)\|^2}$ with $Y_i(\theta) = H_i(\theta - x_i)$ and $H_i = H_i^\top \succeq 0$. Along any segment in parameter space on which $\|Y_i(\theta)\| \geq g_{\min} > 0$ and $\|H_i\| \leq \Lambda$, the map $\theta \mapsto r_i(\theta)$ is Lipschitz with
$$\|\nabla_\theta r_i(\theta)\| \;\leq\; \frac{4\,\|H_i\|^2}{\|Y_i(\theta)\|} \;\leq\; \frac{4\Lambda^2}{g_{\min}},$$
hence, for any $\theta, \theta'$ in that segment,
$$|r_i(\theta') - r_i(\theta)| \;\leq\; \frac{4\Lambda^2}{g_{\min}}\,\|\theta' - \theta\|.$$
Averaging over $i \sim \mathcal{P}_b$ yields the Lipschitz bound for the *Batch Sharpness* $:= \mathbb{E}_i[r_i(\theta)]$:
$$|Batch\ Sharpness(\theta') - Batch\ Sharpness(\theta)| \;\leq\; L_S\,\|\theta' - \theta\|, \qquad L_S := \frac{4\Lambda^2}{g_{\min}}. \tag{28}$$

**Single-time margin implies a uniform window margin.** Assume that along the SGD trajectory on $[t_0, t_0 + T]$ we have the uniform bounds $\|H_i\| \leq \Lambda$ for all $i$ and $\|Y_i(\theta_t)\| \geq g_{\min} > 0$ for all $i, t$, and that the per-step move is bounded by $\|\theta_{t+1} - \theta_t\| = \eta\|Y_{j_t}(\theta_t)\| \leq \eta G_{\max}$ for some $G_{\max} > 0$ (these three constants define the quadratic region under consideration). If at time $t_0$

$$Batch\ Sharpness(\theta_{t_0}) \geq \frac{2 + \epsilon}{\eta} \qquad (\epsilon > 0),$$

then for every $k$ with $0 \leq k \leq T_\star := \left\lfloor \frac{\epsilon}{2\eta\,L_S\,G_{\max}} \right\rfloor$,

$$Batch\ Sharpness(\theta_{t_0 + k}) \geq \frac{2 + \epsilon/2}{\eta}. \tag{29}$$

*Proof.* By (28) and the step bound, $Batch\ Sharpness(\theta_{t+1}) \geq Batch\ Sharpness(\theta_t) - L_S\,\eta G_{\max}$. Iterating $k$ times gives $Batch\ Sharpness(\theta_{t_0+k}) \geq Batch\ Sharpness(\theta_{t_0}) - k\,L_S\,\eta G_{\max} \geq \frac{2}{\eta} + \epsilon\eta - k\,L_S\,\eta G_{\max}$. If $k \leq \epsilon/(2\eta L_S G_{\max})$ this yields (29). $\qquad\square$

**Consequence for the product (plug into the tower/induction step).** On the whole window $t \in \{t_0, \ldots, t_0 + T_\star - 1\}$ we thus have $S(\theta_t) \geq (2 + \epsilon/2)/\eta$, hence from the one-step bound $\mathbb{E}[Z_t \mid \mathcal{F}_t] \geq (1 - \eta S(\theta_t))^2$ and the tower/induction argument,

$$\mathbb{E}\left[ \prod_{s=t_0}^{t_0 + T_\star - 1} Z_s \right] \geq \left(1 + \frac{\epsilon}{2}\right)^{2T_\star}.$$

This upgrades a *single-time* margin at $t_0$ into an explicit *uniform window* of length $T_\star$ over which the exponential lower bound holds.

### G.3 PART 2 FOR SGD *Without* REPLACEMENT

**RR setting and remaining-set sharpness.** Fix an epoch with a finite pool $\mathcal{I} = \{1, \ldots, n\}$. In random reshuffling (RR), within an epoch we draw a uniform random permutation of $\mathcal{I}$ and visit each index exactly once. Let $R_t \subseteq \mathcal{I}$ denote the *remaining* set at step $t$ (those not yet visited in the current epoch) and let $m_t := |R_t|$. Define the *remaining-set sharpness* at $\theta_t$ by

$$S_{\mathrm{rem}}(\theta_t) := \frac{1}{m_t} \sum_{i \in R_t} r_i(\theta_t), \qquad r_i(\theta) := \frac{Y_i(\theta)^\top H_i Y_i(\theta)}{\|Y_i(\theta)\|^2}.$$

Let $\mathcal{F}_t^{\mathrm{RR}} := \sigma(\theta_0$, the permutation prefix up to step $t-1)$; conditionally on $\mathcal{F}_t^{\mathrm{RR}}$, $j_t$ is uniform over $R_t$.

**Step 1: One-step RR bound.** In the quadratic model, $Y_{j_t}(\theta_{t+1}) = (I - \eta H_{j_t})Y_{j_t}(\theta_t)$ and

$$\frac{\|Y_{j_t}(\theta_{t+1})\|^2}{\|Y_{j_t}(\theta_t)\|^2} = 1 - 2\eta\,r_{j_t}(\theta_t) + \eta^2\,q_{j_t}(\theta_t) \geq (1 - \eta\,r_{j_t}(\theta_t))^2, \quad q_i(\theta) := \frac{Y_i(\theta)^\top H_i^2 Y_i(\theta)}{\|Y_i(\theta)\|^2} \geq r_i(\theta)^2,$$

by Cauchy–Schwarz. Averaging uniformly over $R_t$ and using convexity of $x \mapsto (1 - \eta x)^2$ gives

$$\mathbb{E}\left[ \frac{\|Y_{j_t}(\theta_{t+1})\|^2}{\|Y_{j_t}(\theta_t)\|^2} \,\middle|\, \mathcal{F}_t^{\mathrm{RR}} \right] \geq \frac{1}{m_t} \sum_{i \in R_t} (1 - \eta r_i(\theta_t))^2 \geq \left(1 - \eta\,S_{\mathrm{rem}}(\theta_t)\right)^2. \tag{30}$$

**Step 2: Persistence of a single-time margin (high probability).** Assume the quadratic region bounds used in Part 1: $\|H_i\| \leq \Lambda$ for all $i$, $\|Y_i(\theta_t)\| \geq g_{\min} > 0$ for all $i, t$, and $\|\theta_{t+1} - \theta_t\| = \eta\|Y_{j_t}(\theta_t)\| \leq \eta G_{\max}$. As established there,

$$|\,r_i(\theta') - r_i(\theta)\,| \leq \frac{4\Lambda^2}{g_{\min}} \|\theta' - \theta\|$$

$$|\,Batch\ Sharpness(\theta') - Batch\ Sharpness(\theta)\,| \leq L_S\|\theta' - \theta\|, \quad L_S := \frac{4\Lambda^2}{g_{\min}}. \tag{31}$$

Suppose at the beginning of the epoch (time $t_0$) we have a single-time margin

$$Batch\ Sharpness(\theta_{t_0}) \geq \frac{2}{\eta} + \varepsilon\,\eta \qquad (\varepsilon > 0). \tag{32}$$

Fix integers $K \in \{1, \ldots, n-1\}$ and define the minimal remaining size $m_{\min} := n - K$. For any $\delta \in (0,1)$, with probability at least $1 - \delta$ over the RR permutation,

$$\max_{0 \leq k \leq K} \left| \frac{1}{|R_{t_0+k}|} \sum_{i \in R_{t_0+k}} r_i(\theta_{t_0}) - Batch\ Sharpness(\theta_{t_0}) \right| \leq \Delta_K(\delta), \qquad \Delta_K(\delta) := \Lambda \sqrt{\frac{2\log(2K/\delta)}{m_{\min}}}, \tag{33}$$

(Hoeffding–Serfling inequality; we only use the simple range bound $r_i(\theta_{t_0}) \in [0, \Lambda]$). Combining (31) and (33), for each $k \in \{0, \ldots, K\}$ we obtain on the same event

$$S_{\mathrm{rem}}(\theta_{t_0+k}) \geq \underbrace{\frac{1}{|R_{t_0+k}|} \sum_{i \in R_{t_0+k}} r_i(\theta_{t_0})}_{\text{finite-pop mean at } t_0} - L_S\,\|\theta_{t_0+k} - \theta_{t_0}\| \geq Batch\ Sharpness(\theta_{t_0}) - \Delta_K(\delta) - L_S\,\eta G_{\max}\,k. \tag{34}$$

Hence, if $K$ and $\delta$ are chosen so that

$$\Delta_K(\delta) + L_S\,\eta G_{\max}\,K \leq \frac{\varepsilon}{2}\,\eta, \tag{35}$$

then (32) and (34) imply the uniform window margin

$$S_{\mathrm{rem}}(\theta_{t_0+k}) \geq \frac{2}{\eta} + \frac{\varepsilon}{2}\,\eta \qquad \text{for all } k \in \{0, \ldots, K\}, \quad \text{with probability at least } 1 - \delta. \tag{36}$$

**Step 3: Exponential growth over the RR window.** On the event (36), the one-step bound (30) yields

$$\mathbb{E}\big[Z_{t_0+k} \,\big|\, \mathcal{F}_{t_0+k}^{\mathrm{RR}}\big] \geq \big(1 - \eta\,S_{\mathrm{rem}}(\theta_{t_0+k})\big)^2 \geq \big(1 + (\varepsilon/2)\,\eta^2\big)^2 \quad \text{for all } k \in \{0, \ldots, K-1\},$$

where $Z_t := \frac{\|Y_{j_t}(\theta_{t+1})\|^2}{\|Y_{j_t}(\theta_t)\|^2}$. By the tower property and induction,

$$\mathbb{E}\left[\prod_{s=t_0}^{t_0+K-1} Z_s\right] \geq (1 - \delta)\big(1 + (\varepsilon/2)\,\eta^2\big)^{2K}.$$

In words: a single-time margin (32) at the start of the epoch persists, with high probability under RR, over a whole window whose length $K$ is explicitly controlled by the drift budget (31) and the finite-population deviation (33). On that window the product of on–batch factors explodes exponentially.

# H  PROOF OF THE EQUIVALENCE OF SECTION B.2

In this appendix we make precise the informal statement in Section B.2 that, on the local quadratic approximation of the loss, the following three viewpoints are equivalent:

  (i) breaking a valid instability criterion (Definition 4);
 (ii) experiencing a catapult on the quadratic model (Definition 5);
(iii) observing a loss spike of "sufficient" size.

Throughout we work on the local quadratic model introduced in Section B, and make explicit the mild conditions under which the equivalence holds.

## H.1  SETTING AND BASIC ASSUMPTION

Fix a time $t$ and a point $\theta_t$, and consider the quadratic model $\widetilde{L}$ of the loss around $\theta_t$ as defined in Section B. We recall that $\widetilde{L}$ is of the form

$$\widetilde{L}(\theta) \;=\; \frac{1}{N}\sum_{i=1}^{N}\widetilde{L}_i(\theta), \qquad \widetilde{L}_i(\theta) := \frac{1}{2}(\theta - x_i)^{\top}\mathcal{H}_i(\theta - x_i)$$

for some matrices $\mathcal{H}_i$ and points $x_i$.

We continue to denote by $U_t$ an open neighborhood of $\theta_t$ on which the quadratic approximation is accurate, in the sense described informally in Section B. For the arguments below, we only need the following simple condition.

**Assumption 1** (Coercive quadratic model on $U_t$). There exists a symmetric positive semi-definite matrix $\widehat{\mathcal{H}}$ and constants $0 < \mu \leq L < \infty$ such that:

  (a) $\widetilde{L}(\theta) = \frac{1}{2}(\theta - \theta^{\star})^{\top}\widehat{\mathcal{H}}(\theta - \theta^{\star})$ for some $\theta^{\star}$ (i.e. $\widetilde{L}$ is exactly quadratic);
  (b) all non-zero eigenvalues of $\widehat{\mathcal{H}}$ lie in $[\mu, L]$;
  (c) the connected component of $\theta_t$ in the sublevel set $\{\theta : \widetilde{L}(\theta) \leq R\}$ is contained in $U_t$ for all $R$ in a neighborhood of $\widetilde{L}(\theta_t)$.

Assumption 1 is satisfied, for instance, when the full-batch Hessian of the original loss at $\theta_t$ is positive definite in the relevant directions and the neighborhood $U_t$ is chosen small enough. It implies in particular that:

  • for any $R < \infty$, the sublevel set $\{\theta : \widetilde{L}(\theta) \leq R\}$ is compact;
  • the level sets of $\widetilde{L}$ define a family of nested compacts around $\theta_t$ inside $U_t$.

## H.2  BREAKING A VALID INSTABILITY CRITERION $\Longleftrightarrow$ CATAPULT

We first clarify the relationship between valid instability criteria (Definition 4) and catapults (Definition 5).

Recall that Definition 4 is stated for a discrete-time dynamical system $(\theta_s)_{s \geq 0}$ on a parameter space $\Theta$, with an open set $U \subseteq \Theta$, a scalar map $f : U \to \mathbb{R}$ and a threshold $c \in \mathbb{R}$. It says that $f$ is a *valid instability criterion with threshold $c$ on $U$* if

$$f(\theta_0) > c \quad \Longrightarrow \quad (\theta_s)_{s \geq 0} \text{ leaves every compact subset of } U \text{ in finite time.}$$

On the quadratic model at time $t$, Definition 5 simply re-uses this notion with $U := U_t$ and $\theta_0 := \theta_t$:

> We say that the algorithm *experiences a catapult at time $t$* if, when run on $\widetilde{L}$ from initialization $\theta_t$, the resulting trajectory $(\theta_s)_{s \geq t}$ leaves every compact subset of $U_t$ in finite time.

Thus, formally:

**Lemma 3** (Breaking a valid criterion gives a catapult). *Let $f : U_t \to \mathbb{R}$ be a valid instability criterion with threshold $c$ for the quadratic dynamics on $U_t$ in the sense of Definition 4. Fix a time $t$ and consider the trajectory of the quadratic model initialized at $\theta_t$.*

*If $f(\theta_t) > c$, then the quadratic trajectory experiences a catapult at time $t$ in the sense of Definition 5.*

*Proof.* Apply Definition 4 with $U := U_t$ and initial condition $\theta_0 := \theta_t$. Since $f(\theta_t) > c$ and $f$ is a valid instability criterion on $U_t$, we have that the trajectory $(\theta_s)_{s \geq t}$ leaves every compact subset of $U_t$ in finite time. But this is precisely the definition of a catapult at time $t$ on the quadratic model (Definition 5). $\qquad \square$

Conversely, if a catapult occurs at time $t$, one can always *construct* a (possibly highly non-smooth) instability criterion which is broken at $\theta_t$:

**Lemma 4** (Catapult implies existence of a valid criterion). *Assume that the quadratic trajectory from $\theta_t$ experiences a catapult on $U_t$ in the sense of Definition 5. Then there exists a map $f : U_t \to \mathbb{R}$ and a threshold $c \in \mathbb{R}$ such that:*

> *(a) $f$ is a valid instability criterion with threshold $c$ for the quadratic dynamics on $U_t$;*
> *(b) $f(\theta_t) > c$.*

*Proof.* Define $f : U_t \to \mathbb{R}$ by

$$f(\theta_0) := \begin{cases} 1, & \text{if the quadratic trajectory initialized at } \theta_0 \\ & \quad \text{leaves every compact subset of } U_t \text{ in finite time;} \\ 0, & \text{otherwise.} \end{cases}$$

Set $c := \frac{1}{2}$. Then, by construction, $f(\theta_0) > c$ if and only if the trajectory initialized at $\theta_0$ leaves every compact subset of $U_t$ in finite time. Hence $f$ is a valid instability criterion with threshold $c$ in the sense of Definition 4. Since, by assumption, the trajectory from $\theta_t$ experiences a catapult, we have $f(\theta_t) = 1 > c$, as desired. $\qquad \square$

Lemmas 3 and 4 make precise the first equivalence in the box of Section B.2: on the quadratic model, "breaking a (valid) instability criterion" is just another way to state the occurrence of a catapult, and conversely any catapult defines at least one (possibly non-unique) valid instability criterion which is broken.

### H.3 CATAPULTS $\Longleftrightarrow$ LOSS SPIKES OF SUFFICIENT SIZE

We now formalize the relationship between catapults and loss spikes for the quadratic model under Assumption 1.

**Definition 6** (Loss spike of size $\alpha$ on the quadratic model). Fix $t$ and $\theta_t$, and let $\widetilde{L}$ be the quadratic model as above. For $\alpha > 1$ we say that the quadratic trajectory $(\theta_s)_{s \geq t}$ has a *loss spike of relative size at least $\alpha$ at time $t$* if there exists a time $s \geq t$ such that

$$\widetilde{L}(\theta_s) \geq \alpha \, \widetilde{L}(\theta_t).$$

We say that it has loss spikes of *arbitrarily large* relative size if this holds for all $\alpha > 1$.

Under Assumption 1, sublevel sets of $\widetilde{L}$ are compact, and they form a convenient family of compact sets to test the definition of catapult.

**Lemma 5** (Catapult $\Longleftrightarrow$ unbounded quadratic loss). *Under Assumption 1, the following are equivalent for the quadratic trajectory $(\theta_s)_{s \geq t}$:*

> *(i) the trajectory experiences a catapult at time $t$ on $U_t$, i.e. it leaves every compact subset of $U_t$ in finite time;*
> *(ii) the quadratic loss along the trajectory is unbounded above, i.e. $\sup_{s \geq t} \widetilde{L}(\theta_s) = +\infty$.*

*In particular, under Assumption 1, a catapult is equivalent to the existence of loss spikes of arbitrarily large relative size in the sense of Definition 6.*

*Proof. (i) $\Rightarrow$ (ii).* Assume the trajectory experiences a catapult on $U_t$. Suppose for contradiction that $\sup_{s \geq t} \widetilde{L}(\theta_s) \leq R$ for some finite $R$. Then the whole trajectory is contained in the sublevel set

$$K_R := \big\{ \theta \in U_t : \widetilde{L}(\theta) \leq R \big\}.$$

By Assumption 1(b)–(c), $K_R$ is compact and contained in $U_t$. Hence the trajectory *never* leaves $K_R$, which contradicts the definition of a catapult (it must leave *every* compact subset of $U_t$ in finite time). Thus $\sup_{s \geq t} \widetilde{L}(\theta_s) = +\infty$.

*(ii) $\Rightarrow$ (i).* Conversely, assume that $\sup_{s \geq t} \widetilde{L}(\theta_s) = +\infty$. Let $K \subset U_t$ be any compact set containing $\theta_t$. Compactness and continuity of $\widetilde{L}$ imply that

$$R_K := \sup_{\theta \in K} \widetilde{L}(\theta) < \infty.$$

Since $\sup_{s \geq t} \widetilde{L}(\theta_s) = +\infty$, there exists a time $s_K \geq t$ with $\widetilde{L}(\theta_{s_K}) > R_K$, hence $\theta_{s_K} \notin K$.

It remains to check that the trajectory eventually *stays* outside $K$: but this follows immediately from the fact that once $\widetilde{L}(\theta_s) > R_K$, the iterate cannot re-enter $K$ without violating the definition of $R_K$ as the supremum of $\widetilde{L}$ on $K$. Thus, for every compact $K \subset U_t$ containing $\theta_t$, there is a finite time $s_K$ such that $\theta_s \notin K$ for all $s \geq s_K$. This is exactly the definition of a catapult on $U_t$. □

Combining Lemma 5 with Definition 6 immediately yields:

**Corollary 2** (Catapult $\Longleftrightarrow$ loss spikes of sufficient size). *Under Assumption 1, for the quadratic trajectory $(\theta_s)_{s \geq t}$ the following are equivalent:*

(i) *the trajectory experiences a catapult at time $t$ on $U_t$;*

(ii) *for every $\alpha > 1$ there exists $s \geq t$ such that $\widetilde{L}(\theta_s) \geq \alpha \widetilde{L}(\theta_t)$, i.e. the trajectory has loss spikes of arbitrarily large relative size in the sense of Definition 6.*

From the point of view of Section B.2, we can now interpret a *loss spike of sufficient size* as a spike whose relative height exceeds any upper bound that would be compatible with bounded (linearly stable) dynamics on the quadratic model. In particular, in regimes where the quadratic dynamics is linearly stable and confined to a given compact subset of $U_t$, the loss is uniformly bounded and only admits spikes of bounded relative size; Corollary 1 in Appendix D quantifies this explicitly in a one-dimensional example.

### H.4 PUTTING THE PIECES TOGETHER

We can now summarize the equivalence more succinctly.

**Theorem 2** (Equivalence of the three viewpoints on the quadratic model). *Work on the quadratic approximation $\widetilde{L}$ at time $t$ under Assumption 1. For the quadratic trajectory $(\theta_s)_{s \geq t}$ on $U_t$, the following statements are equivalent:*

(i) *there exists a valid instability criterion $f : U_t \to \mathbb{R}$ with threshold $c$ in the sense of Definition 4 such that $f(\theta_t) > c$;*

(ii) *the trajectory experiences a catapult at time $t$ on $U_t$ in the sense of Definition 5;*

(iii) *the quadratic loss along the trajectory has loss spikes of arbitrarily large relative size, i.e. for every $\alpha > 1$ there exists $s \geq t$ with $\widetilde{L}(\theta_s) \geq \alpha \widetilde{L}(\theta_t)$.*

*Proof.* (i) $\Rightarrow$ (ii) is Lemma 3. (ii) $\Rightarrow$ (i) is Lemma 4.

(ii) $\Leftrightarrow$ (iii) is exactly Lemma 5 together with the definition of loss spikes (Definition 6) and Corollary 2. □

Theorem 2 formalizes the slogan in Section B.2: on the quadratic approximation, the underlying property is *divergence of the dynamics on $U_t$*, and the three diagnostics we use throughout the paper—breaking an instability criterion, observing a catapult, and observing a large enough loss spike—are just different ways of detecting the same phenomenon.

# I   ON THE FATE OF $\lambda_{\max}$

In this section we examine how $\lambda_{\max}$ behaves once EoSS is reached and clarify its relationship to *Batch Sharpness*. A key aspect of the original EoS analysis is, indeed, that the controlling quantity—the largest eigenvalue of the full-batch Hessian $\lambda_{\max}$—has an immediate geometric interpretation. There exists an extensive literature about $\lambda_{\max}$ size and role in neural networks, and it is a main ingredient of any proof of convergence. The EoSS picture replaces $\lambda_{\max}$ with *Batch Sharpness*, a statistic whose connection to generalization and role in optimization theory is largely unexplored.

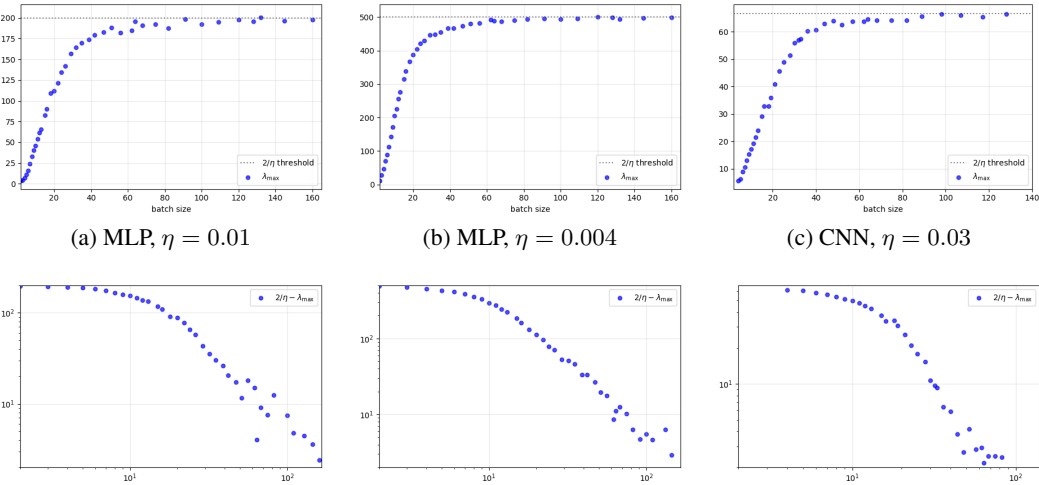

(a) MLP, $\eta = 0.01$      (b) MLP, $\eta = 0.004$      (c) CNN, $\eta = 0.03$

Figure 12: **Stabilisation level of $\lambda_{\max}$ across step sizes and architectures. Top:** final-epoch $\lambda_{\max}$ vs. batch size. **Bottom:** log–log plots of the gap $2/\eta - \lambda_{\max}$ for the same runs. All experiments use CIFAR-10 8k.

## I.1   EMPIRICAL FACTS

Below the phenomena we extensively observe in vision classification tasks trained with MSE, ablating on batch sizes, step sizes, architectures, datasets. See Figure 6 for a good reference of what generally goes on.

- **Fact 1: Progressive Sharpening.** $\lambda_{\max}$ increases at most as long as *Batch Sharpness* increases.
- **Fact 2: Phase Transition.** Once *Batch Sharpness* plateaus at $2/\eta$, $\lambda_{\max}$ stops increasing. If it moves, it only decreases from this time on.
- **Fact 3: Path-dependence.** If changes to hyper parameters are made, *Batch Sharpness* changes abruptly or restart growing and $\lambda_{\max}$ also changes. Stabilization of both happen as *Batch Sharpness* reaches $2/\eta$. The trajectory of $\lambda_{\max}$ is not fully determined by the size of hyper parameters (see Figure 7). That is, the level of $\lambda_{\max}$ is *path-dependent*: it inherits the history of progressive sharpening up to the moment EoSS is reached.
- **Fact 4: Smaller batches $\Rightarrow$ flatter minima.** Across every setting we tested, reducing the batch size monotonically decreases the plateau level of $\lambda_{\max}$. This aligns with the long-standing empirical observation that smaller batches locate flatter minima see, e.g., Keskar et al. (2016); Jastrzębski et al. (2021)).
- **Fact 5: A critical batch size marks the SGD $\rightarrow$ GD crossover.** Each curve in Figure 12 exhibits a bend at $b \approx b_c(\eta)$: for $b < b_c$ the plateau falls rapidly with $b$, while for $b > b_c$ it

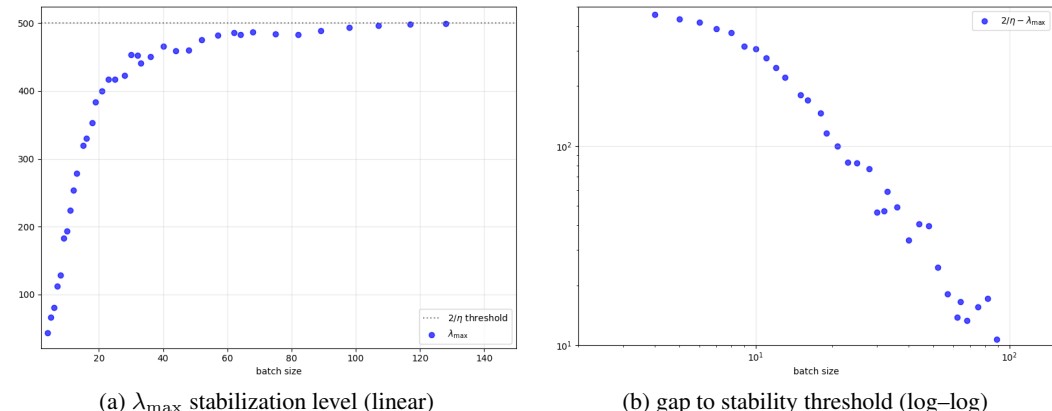

(a) $\lambda_{\max}$ stabilization level (linear)  (b) gap to stability threshold (log–log)

Figure 13: **Baseline MLP: stabilization of $\lambda_{\max}$ as a function of batch size**. Baseline MLP (2 hidden layers, width 512) trained on an 8k-subset of CIFAR-10 with step size 0.004 until convergence. **(a)** Final $\lambda_{\max}$ (linear axes). Smaller batches settle to flatter minima. For batch sizes *below* the *critical batch size* $b_c$ the level of stabilization is significantly below the $2/\eta$ level of full-batch, indicating strong *implicit regularization*. Moreover, the curve is steep, making the the final landscape sensitive to the choice of batch size. For *larger* batches ($b > b_c$) the slope flattens and $\lambda_{\max}$ plateaus close to $2/\eta$, so the dynamics resemble full-batch GD, implicit regularization is **weak**. **(b)** Log–log plot of the gap $2/\eta - \lambda_{\max}$, used to test for any power-law decay.

flattens and approaches the full-batch value. This $b_c$ corresponds to the regime in which the mini-batch landscapes approximate *well enough* the full-batch landscape, restoring GD-like dynamics (Appendix I.2).

- **Fact 6: No universal power law.** From *static* analysis, one would expect a scaling $2/\eta - \lambda_{\max} = O(b^{-\alpha})$ for some $\alpha$. The log–log plots (bottom row of Figure 12) show no robust straight-line behaviour, ruling out such law for any possible exponent $-\alpha$.

### I.2  CRITICAL BATCH SIZE

We can characterize two regimes for the stabilization levels (see Figure 13):

**(i) Small-batch regime** ($b \leq b_c$): $\lambda_{\max}$ stabilizes well *below* the full-batch threshold $2/\eta$, signaling strong implicit regularization by SGD. The stabilization level rises steeply with batch size, so even modest changes in $b$ materially affect the final curvature of the loss landscape of the solution

**(ii) Large-batch regime** ($b \geq b_c$): the growth of $\lambda_{\max}$ with $b$ becomes much slower and the curve asymptotically approaches $2/\eta$ from below, mirroring full-batch gradient descent and reflecting weak implicit regularization.

The *critical batch size* $b_c$ is therefore the point at which the training dynamics cross over into a full-batch-like regime. Works as Zhang et al. (2024) study the following notion of *critical batch size*: "the point beyond which increasing batch size may result in computational efficiency degradation". Likewise, works focusing on generalization performance depending on the batch size (Masters & Luschi, 2018) identify a cut-off batch sizes above which test performance degrades significantly. We conjecture there may be a relation between these quantities and leave a systematic investigation to future work.

### I.3  WHY $2/\eta - C/b^\alpha$ FAILS.

From linear stability analyses near the manifold of minima (Wu et al., 2018; Ma & Ying, 2021; Granziol et al., 2021; Mulayoff & Michaeli, 2024) or random matrix theory (together with the fact that we have *Batch Sharpness* stabilize at $2/\eta$) one would expect to have a law of the form $\lambda_{\max} \approx 2/\eta - O(1/b^\alpha)$. Log-log plots of the gap $2/\eta - \lambda_{\max}$ in Figure 13b shows no robust power law (for the lack of any linear dependency), invalidating this prediction (see also Figures 14-20). Importantly, this does not invalidate the findings of those theories, instead showcases the insufficiency of a *static*

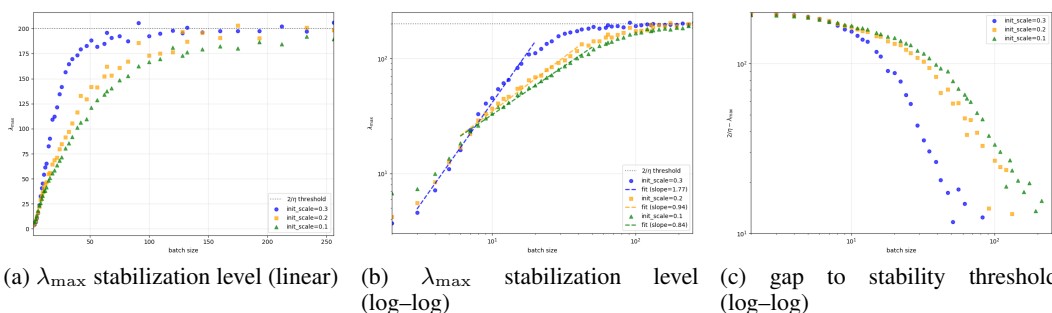

(a) $\lambda_{\max}$ stabilization level (linear)  (b) $\lambda_{\max}$ stabilization level (log–log)  (c) gap to stability threshold (log–log)

Figure 14: **Effect of weight-scale at initialization on the EoSS stabilization of $\lambda_{\max}$.** We train the same network and dataset under identical hyperparameters, varying only a global rescaling ($\times 0.1, 0.2, 0.3$ of He) of the initial weights. (a) Final-epoch $\lambda_{\max}$ as a function of batch size (linear axes). Smaller batches always converge to flatter solutions, yet the absolute level—and the critical batch size at which the curve begins to approach the full-batch limit $2/\eta$ (horizontal dashed line)—shift markedly with the initialization scale. This demonstrates that the landscape geometry at convergence is already seeded by early-training choices. (b) Same data in log-log scale. The three curves exhibit distinct slopes, ruling out a single power-law exponent and confirming strong path-dependence. Linear fit is provided to the linear portion (c) Log–log plot of the gap ,$2/\eta - \lambda_{\max}$,. The absence of a straight line contradicts the prediction $2/\eta - \lambda_{\max} \propto b^{-\alpha}$ that follows from linear stability analyses near a minimum.

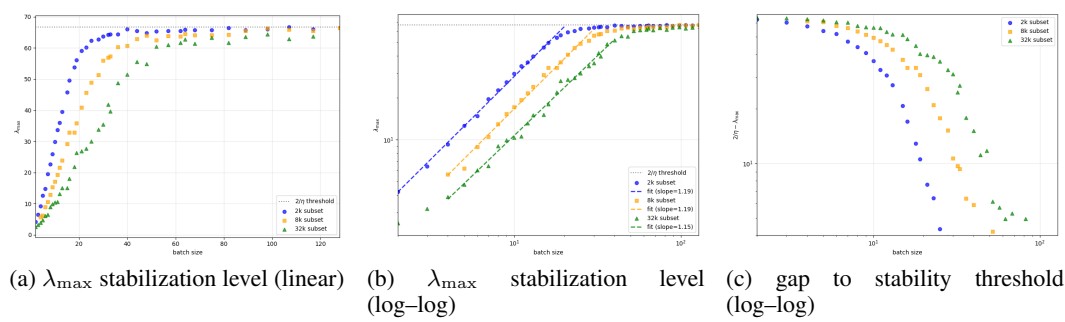

(a) $\lambda_{\max}$ stabilization level (linear)  (b) $\lambda_{\max}$ stabilization level (log–log)  (c) gap to stability threshold (log–log)

Figure 15: **Varying dataset size alters the EoSS plateau of $\lambda_{\max}$ for a CNN.** We use the same setup as Fig. 14 but instead varying the number of training examples (2k, 8k, 32k). Larger datasets drive $\lambda_{\max}$ to lower plateaus—i.e. flatter minima—and push the critical batch size (the knee toward the full-batch limit $2/\eta$) to higher b, as expected from b/N scaling. Plateau heights also differ from the MLP results in Fig. 14 or 13, highlighting architectural sensitivity. Panel order and axes mirror Fig. 14; see that caption for sub-plot details.

analysis. Indeed, those estimates are taken from changing the batch size *statically*, without making any training steps. In particular, linear stability analisys does accommodate virtually any law, as long as there is change in alignment between the mini-batch gradients. The fact that the static law does not apply means that there is a change to the alignment also happening. Therefore, as will be discussed further in detail, the fact that these estimates do not apply means that to give faithful description of the loss landscape at convergence one has to undertake an analysis that is path-dependent.

### I.4 Conclusion & Outlook: Why Path-Dependence Matters

With all of the above, we arrive at a **negative answer** to the question posed at the start:

> *There is no single, path-independent law that fixes the stabilization level of $\lambda_{\max}$ from basic hyper-parameters alone.*

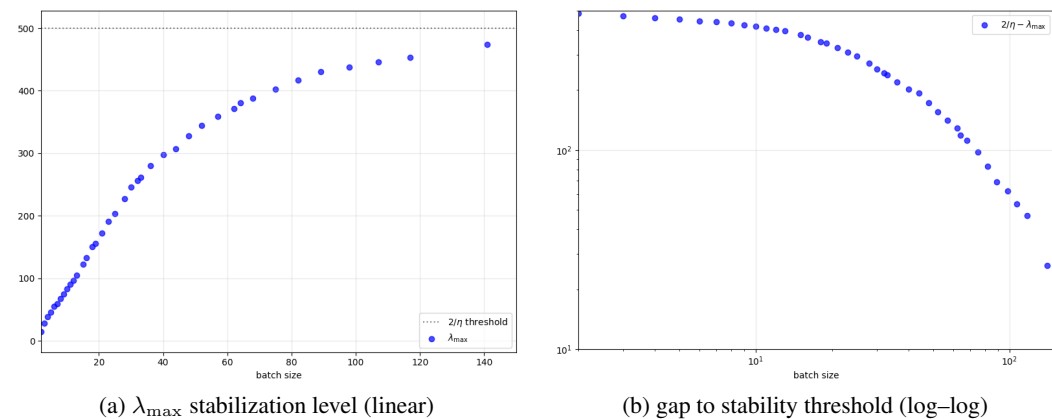

(a) $\lambda_{\max}$ stabilization level (linear)      (b) gap to stability threshold (log–log)

Figure 16: **Level of stabilization of $\lambda_{\max}$.** Same setup as Fig. 13 but the initial weights are rescaled by $1/3$; see Fig. 14 for the broader effect of initialization. (See Fig. 13 for sub-plot explanations.)

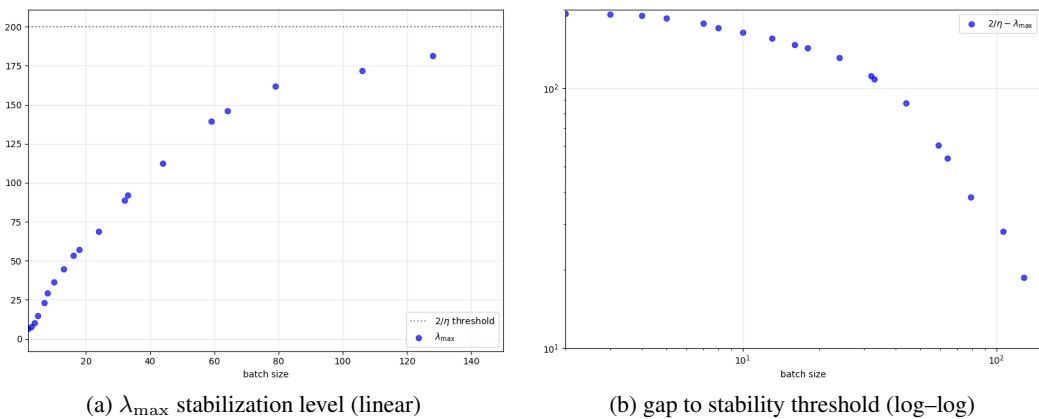

(a) $\lambda_{\max}$ stabilization level (linear)      (b) gap to stability threshold (log–log)

Figure 17: **Level of stabilization of $\lambda_{\max}$.** Identical to Fig. 13 except for a larger step size of $0.01$. (See Fig. 13 for sub-plot explanations.)

## I.5 IMPLICATIONS AND OPEN QUESTIONS

The findings above lead to the following main conclusions.

(C1) $\lambda_{\max}$ *is not the stability limiter for mini-batch training.* *Batch Sharpness* governs EoSS; $\lambda_{\max}$ follows. $\lambda_{\max}$ is capped from above by the value it reaches at the phase transition characterized by *Batch Sharpness* reaching $2/\eta$. This and Facts 1—3 above imply that:

> The stabilization of $\lambda_{\max}$ is a *by-product* of EoSS, not the quantity that governs it.

(C2) *A theory of $\lambda_{\max}$ has to account for the* correct *progressive sharpening.* By fixing the model and changing batch size $b$, the gap between the maximal eigenvalue of $\mathbb{E}\big[\lambda_{\max}(\mathcal{H}(L_B))\big]$ and $\lambda_{\max} = \lambda_{\max}(\mathcal{H}) = \lambda_{\max}\mathbb{E}[\mathcal{H}(L_B)]$ scales as $1/b$. Any theory that keeps the parameter vector fixed and only varies $b$, or anyways leads to a power law, misses the path-dependent descent that determines where training arrives and where $\lambda_{\max}$ stabilizes. Facts 3 and 6 thus imply that analysis of $\lambda_{\max}$ is insufficient *if* it does not account for **(1)** the precise and correct effect of progressive sharpening on the higher moments of the Hessian and **(2)** the correct alignment between mini-batch steps and Hessians.

Quantifying the plateau of $\lambda_{\max}$ is thus still an (important) open problem. A complete account will require a dynamical theory *through* the progressive-sharpening phase and beyond. Not just properties at its endpoint as for full-batch methods.

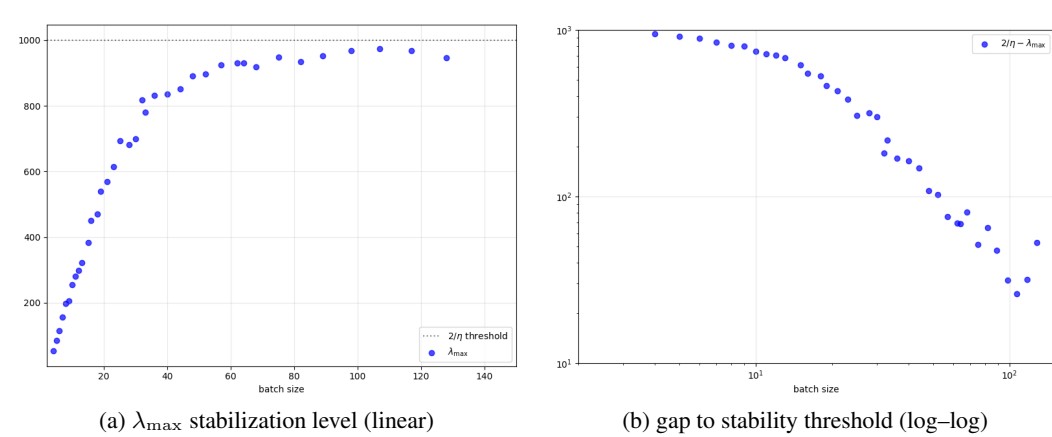

(a) $\lambda_{\max}$ stabilization level (linear)

(b) gap to stability threshold (log–log)

Figure 18: **Level of stabilization of $\lambda_{\max}$.** Baseline network trained on a 32k-subset of CIFAR-10 subset with step size 0.002. (See Fig. 13 for sub-plot explanations.)

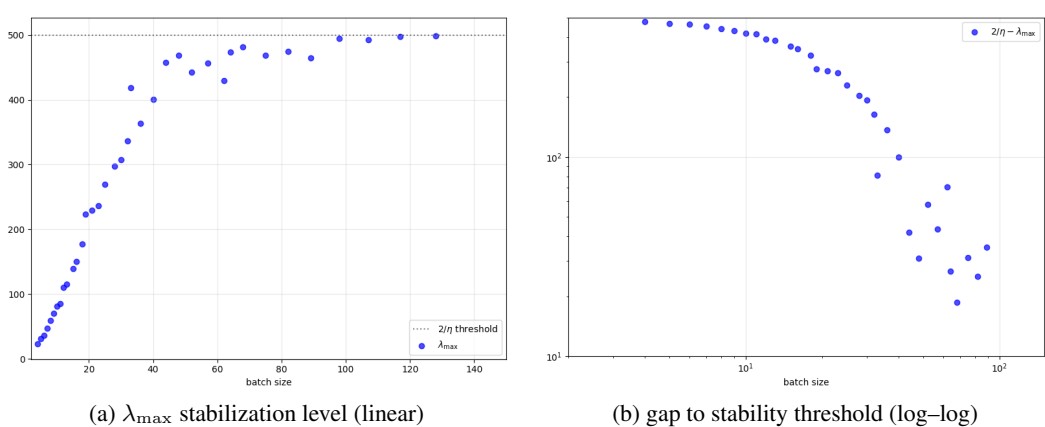

(a) $\lambda_{\max}$ stabilization level (linear)

(b) gap to stability threshold (log–log)

Figure 19: **Level of stabilization of $\lambda_{\max}$.** Deeper MLP (the `mlp_l`: 4 hidden layers, width 512) on the 8k-subset, step size 0.004. (See Fig. 13 for sub-plot explanations.)

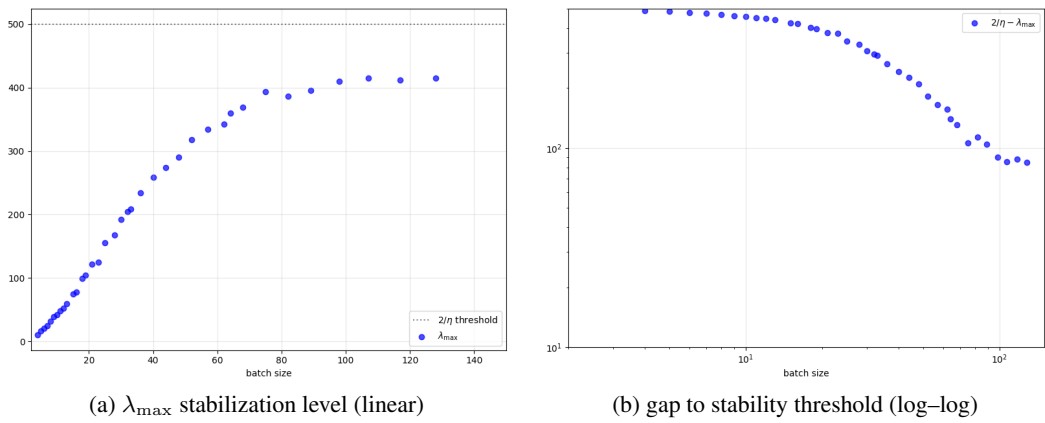

(a) $\lambda_{\max}$ stabilization level (linear)

(b) gap to stability threshold (log–log)

Figure 20: **Level of stabilization of $\lambda_{\max}$.** Same deeper MLP as in Fig. 19 but trained on a 32k subset. (See Fig. 13 for sub-plot explanations.)

## J  IMPLICATIONS: HOW NOISE-INJECTED GD DIFFERS FROM SGD

**SGD vs. Noisy Gradient Descent.**  A common belief is that SGD's regularization stems from its "noisy" gradients, which find flatter minima. However, our analysis points to the "noisy" Hessians as crucial. To test this, we compare mini-batch SGD (batch size 16) against three noisy GD variants: (see details in Appendix J)

- *Anisotropic Sampling Noise:* Gaussian reweighting on the samples (Wu et al., 2020), which is different from SGD but maintains the mini-batch structure (and injects noise in the Hessians).
- *Diagonal Noise:* Gaussian noise restricted to the diagonal part of the SGD noise covariance (Zhu et al., 2019).
- *Isotropic Noise:* Gaussian noise with isotropic covariance (Zhu et al., 2019).
- *SDE dynamics integration* (Li et al., 2017)

As shown in Figure 8, only noise which maintains the higher moments of the Hessian(s) (and thus preserves the mini-batch landscape structure) leads to an EoSS-like regime with $\lambda_{\max}$ stabilizing well below $2/\eta$. More generic (e.g., diagonal or isotropic) noise fails to reproduce this behavior. These experiments suggest that stability thresholds differ fundamentally between mini-batch SGD (governed by *Batch Sharpness*) and noise-injected GD (governed by $\lambda_{\max}$). Notably, these results are consistent with the findings of Zhu et al. (2019)—although their focus is on generalization. Unsurprisingly, in the case in which the noise affects only the gradients—not the Hessians—indeed, EoSS comes for $\lambda_{\max} = 2/\eta$ as for GD (Ma & Ying, 2021; Mulayoff & Michaeli, 2024). Even in the quadratic setting, the appearance of *Type-1* oscillations and *GNI* are not affected by the structure and distribution of the Hessian on the mini-batches, see Appendix E. The stability threshold, however, is affected. It depends on the Hessian's higher moments, see Theorem 1 or (Ma & Ying, 2021; Mulayoff & Michaeli, 2024).

**Challenges for SDE Modeling.**  Classical analyses of neural network optimization often assume a single, static landscape: (*i*) **Online** perspective, modeling each step's gradient as a noisy unbiased estimator of the expected gradient, or (*ii*) **Offline** perspective, treating the dataset as fixed and SGD as noisy GD on the empirical loss. In both views, it is the *full-batch* Hessian that supposedly drives curvature. Our results instead highlight that each update sees a Hessian $\mathcal{H}(L_{\mathbf{B}})$ that generally differs significantly from $\mathcal{H}$, leading to *Batch Sharpness* stabilizing at $2/\eta$ when $\lambda_{\max}$ is smaller.

> Standard SDE—or analogous—approximations of SGD cannot thus describe the location of convergence of SGD or its behavior for neural networks under the assumption of progressive sharpening. Indeed, they typically ignore any statistics of the Hessians except for the mean.

Prior works already note limitations of SDE-based approaches for SGD implicit regularization: they may be mathematically ill-posed (Yaida, 2018), fail except under restrictive conditions (Li et al., 2021), converge to qualitatively different minima (HaoChen et al., 2020), or miss higher-order effects (Damian et al., 2021; Li et al., 2022). Recent discrete analyses (Smith et al., 2021; Beneventano, 2023; Roberts, 2021) attempt to address some of these issues. Nonetheless, our findings expose a deeper gap: when batch sizes are small, the *geometry of the mini-batch Hessian* differs markedly from that of the full-batch, altering both eigenvalues and eigenvector alignments. Conventional SDE models, which assume a static or average Hessian, cannot easily capture these rapid fluctuations.

### J.1  NOISY GD

We are running a number of noisy GD implementations.

#### J.1.1  NOISY GD WITH ANISOTROPIC NOISE (GAUSSIAN RESAMPLING)

This version of noisy GD essentially preserves the mini-batch landscape structure by averaging the landscapes using Gaussian sampling noise. In particular, it takes a Gaussian sampling vector with the same first and second moments as the sampling vector of SGD. Now, this trivially forces the expectation of the mini-batch Hessians to be the same between SGD and Gaussian resampling (and essentially equal to the full-batch Hessians. Importantly, though, this also makes the covariance of the mini-batch Hessians to be the same between SGD and GD with Gaussian resampling noise (as per linearity of the mini-batch Hessians in the weights of the sampling vector). Together with the

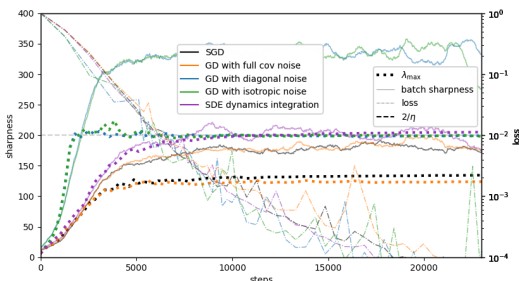

Figure 21: Version of Figure 8, with the loss curves added.

fact that GD with the Gaussian resampling behaves in the same manner as SGD from the point of view of stability, *Batch Sharpness*, and suppression of $\lambda_{\max}$—it is an indicator of the fact that it is the higher moments of the mini-batch Hessians that determine the dynamics SGD; and it is indeed the noise in the Hessians that creates the instability regime of EoSS and its consequences. As a weaker consequence, it also preserves the covariance of the noise of SGD.

For implementation details refer to Wu et al. (2020). In summary, we re-draw the sampling vector at each step with the corresponding covariance.

### J.1.2 NOISY GD WITH DIAGONAL NOISE

This implementation follows Zhu et al. (2019) — it recreates what they refer to as "GLD diagonal". This is essentially noisy GD with the noise covariance being equal to the diagonal of the covariance of the noise produced by SGD. This preserves each parameter's marginal variance while ignoring off-diagonal correlations. Conceptually, we are approximating SGD's noise by $\mathcal{N}\left(0, \frac{1}{b}\mathrm{diag}(\Sigma(\theta))\right)$ and add it to the full-batch gradient before the optimizer step. Essentially, this is one step further from a true SGD then the aforementioned GD with anisotropic noise. In particular, it does not preserve the mini-batch landscape structure. As a result, the behavior of GD with diagonal noise differs from SGD from the point of view of $\lambda_{\max}$ stabilizing below $2/\eta$, and instead stabilizing at $2/\eta$. We refer the reader to (Zhu et al., 2019) for the details of implementation. In our implementation, we compute the diagonal of the covariance every 30 steps and reuse it on those 30 steps (as it is too computationally expensive to compute it at every step).

### J.1.3 NOISY GD WITH ISOTROPIC NOISE

This implementation follows Zhu et al. (2019) — it recreates what they refer to as "GLD dynamic". This is essentially noisy GD with the noise covariance being identity (hence the "isotropic"), scaled such that the magnitude of the noise conincides with that of SGD. That is, this is isotropic gradient noise that matches the average variance of SGD noise but ignores both parameter-wise variability and correlations. Conceptually, we are approximating SGD's noise by $\mathcal{N}\left(0, \frac{\sigma^2}{b}I\right)$ add it to the full-batch gradient before the optimizer step, where $\sigma^2 = \frac{\mathrm{tr}(\Sigma)}{d}$ is the mean per-parameter variance from the per-sample gradient covariance $\Sigma$, $b$ is the target batch size, and $d$ is the number of parameters. This is one step "further" from SGD then the noisy GD with diagonal noise. Consequently, this sort of noisy GD does not preserve the regularization effect of SGD on $\lambda_{\max}$ either.

### J.2 SDE

We are taking the standard SDE approximation of SGD: (see e.g. Li et al. (2018))

$$d\theta_t \;=\; -\nabla f(\theta_t)\, dt \;+\; \sqrt{\eta}\,\Sigma^{1/2}(\theta_t)\, dW_t$$

where $dW_t$ is the standard d-dimensional Wiener process, and $\Sigma$ is the covariance matrix of mini-batch gradients.

To simulate its dynamics, we are using the Euler–Maruyama discretization with a step size of $0.0005$, chosen to be sufficiently small compared to $\eta$ (1/20th of $\eta = 0.01$ in this example). In Figure 22 we are showing a number of sample paths of the SDE trajectory illustrate the similarity in the properties

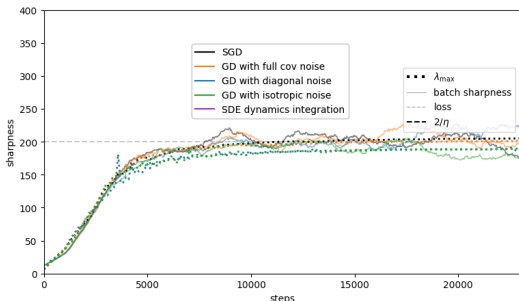

Figure 22: **SDE sample paths** Multiple realizations of SDE trajectory to showcase the similarity of the solutions found by SDE dynamics

of the solutions found by those dynamics – in particular, that $\lambda_{\max}$ stabilizes around $2/\eta$, rather than below as it does for SGD dynamics. In all the experiments, batch size is 16, and $\eta$ is 0.01.

# K    ALIGNMENT

The stability of mini-batch SGD is governed by the geometry of the mini-batch loss landscape, rather than solely the full-batch landscape. As further discussed in Sections B, 5, L, M, the stability of SGD depends not only on the magnitude of mini-batch Hessians but also critically on their alignment (both pairwise and with the loss gradients). This appendix offers a limited characterization of the alignment structure relevant to mini-batch Hessians.

We approximate both the full-batch and mini-batch Hessians by their Gauss–Newton matrices, an approximation commonly used in analyses of SGD (e.g., (Wu et al., 2018; Ma & Ying, 2021; Mulayoff & Michaeli, 2024)), valid at convergence, and supported empirically (e.g., (Papyan, 2019) and Appendix P). Concretely,

$$\nabla_\theta^2 L_B(\theta) \approx \frac{1}{b} \sum_{i=1}^{B} J_i^\top H_{z,i} J_i$$

where $J_i$ is the Jacobian of the model output with respect to $\theta$ and $H_{z,i}$ is the loss Hessian with respect to the output evaluated at the $i$-th sample. We work with MSE and, for simplicity, consider a two-class setting (i.e., a single model output), which yields:

$$\nabla_\theta^2 L_B(\theta) \approx \frac{1}{b} \sum_{i=1}^{b} \nabla_\theta f_i \nabla_\theta f_i^\top$$

where $\nabla f_i$ is the per-sample model gradient.

Under this structure, properties of mini-batch Hessians—including their top eigenvalues and the cross-batch alignment/commutativity—are controlled (though not fully determined) by the pairwise alignment of the per-sample model gradients. We report the empirical distributions of these pairwise alignments in Figure 23 (MLP) and Figure 24 (CNN), plotting pairwise dot products, plotting pairwise dot products, cosine similarity and individual norms. The evolution of the distributions throughout the training indicates the effects of progressive sharpening—for example, the growth of norm of model gradients corresponds to the increase of $\lambda_{\max}$ and $\lambda_{\max}^b$, see Appendix M. A complete description of the dynamics would require a precise account of progressive sharpening and falls outside the scope of this work.

The important observation for our purposes is that per-sample model gradients are only *weakly* aligned, with cosine similarities clustered around $0.1$—which is still much higher than random $d$-dimensional vectors would have. Two immediate implications follow. First, mini-batch Hessians are generically non-commuting (as the model gradients are not orthogonal or completely collinear)—an aspect that matters for linear stochastic stability (Appendix L). L. Second, if we fix the parameters $\theta$ and vary the batch size b, the eigenspaces of the mini-batch Hessian mix gradually, which induces a gap between *Batch Sharpness* and $\lambda_{\max}^b$ (see Appendix M). We leave full characterization of the mini-batch Hessians, which would depend on the higher moments of model gradients for future work. Still, these observations underscore that a comprehensive, training-time characterization of the structure of mini-batch Hessians is an important future direction of research for understanding SGD dynamics.

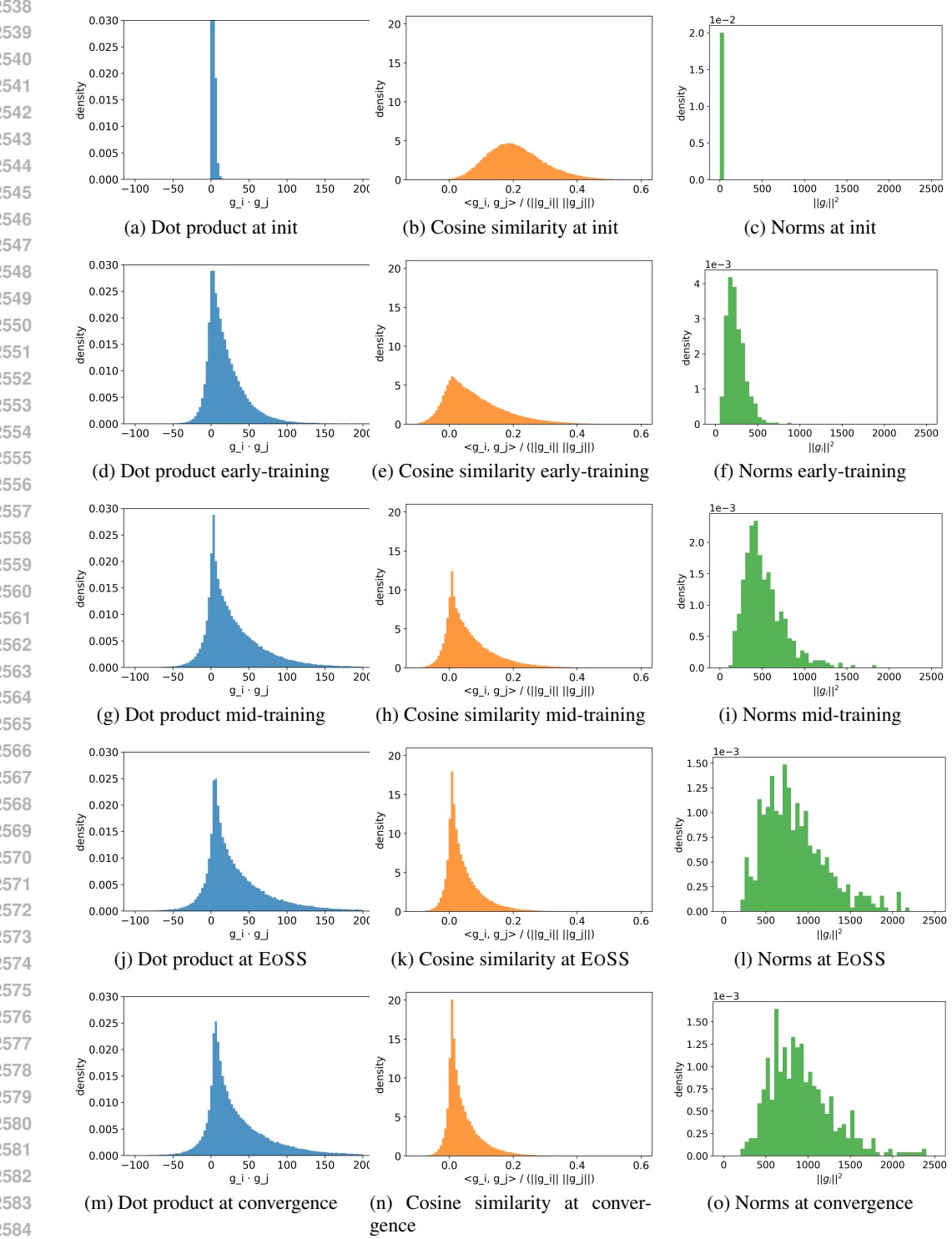

Figure 23: **Model gradients alignment.** Pairwise alignment between model gradients forming the Hessian as the training progresses. We show dot products ($i \neq j$), cosine similarities and the squared norms of the model gradients. Each row corresponds to a stage of training—from initialization, to mid-training (during progressive sharpening), to the later stages (at EoSS and convergence). Notice the gradients become weakly aligned throughout the training (with the cosine similarities clustered around $0.1$), but not completely orthogonal, as it would have been with random vectors. MLP, CIFAR-8k (2 classes), $\eta = 0.02$, batch size 32

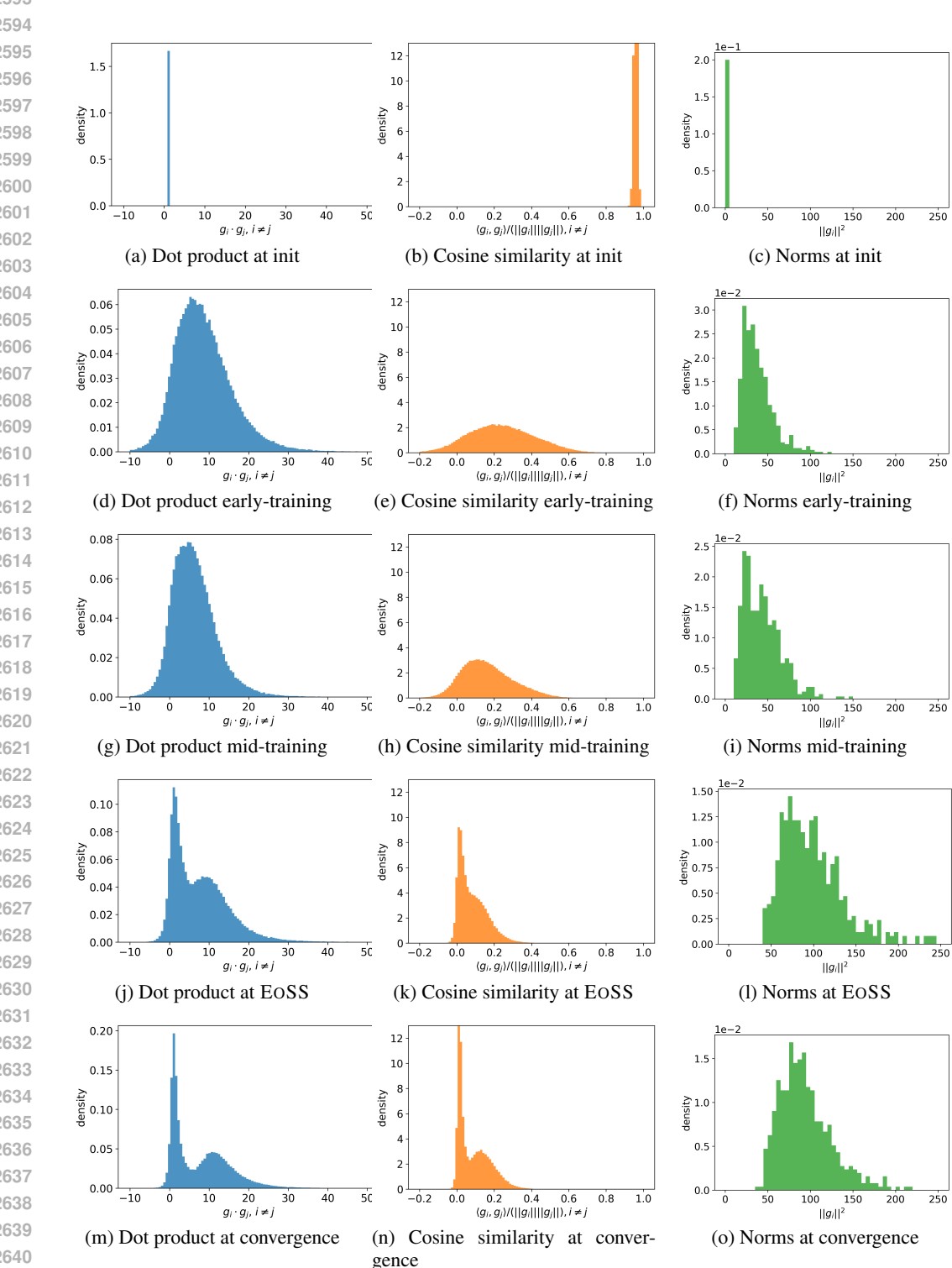

Figure 24: **Model gradients alignment.** Same setup as Figure 23, but for CNN. CIFAR-8k (2 classes), $\eta = 0.05$, batch size 8.

## L    LINEAR STOCHASTIC STABILITY

This appendix extends the discussion from Section B, particularly regarding works addressing SGD stability from a linear stability viewpoint, including an analysis of the behavior of the bound introduced by Wu et al. (2018).

### L.1    NOTIONS OF LINEAR STABILITY

As the discussion in Section B highlights, we need two conditions to establish a regime of instability in mini-batch training: (i) a valid notion of stability as per Definition 4—an inequality whose violation leads to divergence—and (ii) empirical saturation of this stability notion during SGD training, with continued training as the condition remains at saturation.

Wu et al. (2018) were the first to analyze linear stability of SGD, establishing a *sufficient* (but, in general, necessary) condition for SGD stability. In particular, they prove that mini-batch gradient descent is stable when

$$\lambda_{\max}\left((I - \eta\mathcal{H})^2 + \frac{\eta^2}{b}\mathbb{E}[\mathcal{H}_i^2 - \mathcal{H}^2]\right) \;=\; \lambda_{\max}\left(\mathbb{E}_B\left[(I - \eta\mathcal{H}_B)^2\right]\right) \leq 1. \tag{37}$$

This criterion upper-bounds the spectral radius of the second-moment update operator. Since this condition is only sufficient (and necessary solely when $d = 1$), it does not strictly satisfy our criteria for a valid stability notion (Definition 4). Importantly, while Wu et al. (2018) explicitly note this limitation, nothing a priori excludes this bound from being *empirically* tight—after all, EoSS is fundamentally an empirical phenomenon. We show here that this criterion does not govern the EoSS—that is, that the "not necessary" part is not vacuous.

**Necessary stability conditions.**    Conditions derived from linear stochastic stability theory that are indeed valid stability notions often suffer from computational intractability. Indeed, Ma & Ying (2021) proved that mini-batch gradient descent is 2nd order linearly stable *if and only if* the operator

$$T_k := \mathbb{E}_B\left[(I - \eta H(L_B))^{\otimes 2}\right] \tag{38}$$

is a contraction on the cone of PSD matrices. Importantly for us, this a necessary condition for stability, which would constitute a notion of stability. While this condition is necessary and would represent a valid stability notion, it is computationally infeasible in high-dimensional neural networks, as it involves operations on $d^2 \times d^2$ tensors (making even tensor-vector product unfeasible).

Mulayoff & Michaeli (2024) showed that the PSD condition is inactive, which reduces the criterion to one on a spectral norm of this operator. Moreover, they express this notion in the form of (notion of curvature) $\leq$ (step-size dependent threshold). Although potentially useful to find the corresponding maximum stable learning rate, this reformulation did not solve the incomputability problem. (Mulayoff & Michaeli, 2024) also construct elegant lower bounds, which therefore also serve as a necessary condition for stability, and thus a valid notion of stability. However, as their empirical results show, these bounds never saturate, and thus do not effectively capture the empirical presence of an instability regime in mini-batch training.

### L.2    EMPIRICAL BEHAVIOR OF WU ET AL. (2018) CRITERION

**Always above** 1.    In neural networks negative Hessian eigenvalues are typically present, thus the quantity in Equation (37), which we term *second moment contraction*, is always bigger than 1 (Figure 25). Ideally, when "the phase transition at EoSS" happens this quantity keeps being bigger than 1, but the highest singular value becomes the biggest eigenvalue instead of the smallest. In the deterministic full-batch algorithm case this can be seen cleanly:

$$\lambda_{\max}\left(\mathbb{E}_B\left[(I - \eta\mathcal{H}_B)^2\right]\right) = \lambda_{\max}\left((I - \eta\mathcal{H})^2\right) =$$

$$\begin{cases} 1 - \eta\lambda_{\min}(\mathcal{H}) & \sim 1 + \epsilon_1 & \text{when } \lambda_{\max} \leq 2/\eta \text{ and } \lambda_{\min} < 0 \\ |\eta\lambda_{\max} - 1| & \sim 1 + \epsilon_2 & \text{when } \lambda_{\max} \geq 2/\eta \text{ and } \lambda_{\min} < 2 - \eta\lambda_{\max} \leq 0. \end{cases}$$

Thus we can think of plotting the quantity

$$\lambda_{\max}\left(-2\eta\mathcal{H} + \eta^2\mathbb{E}_B\left[\mathcal{H}(L_B)^2\right]\right) \lesseqgtr 0. \tag{39}$$

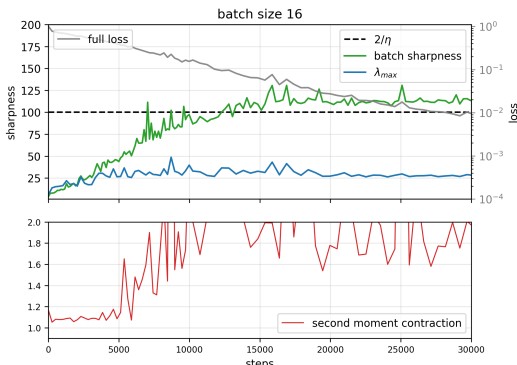

Figure 25: *Linear stochastic stability.* Tracking the condition of equation (37) ("second-moment contraction") during training. The value stays strictly above 1.

As illustrated in Figure 26, the condition (39) is initially slightly above zero (due to negative eigenvalues). Then it undergoes the aforementioned phase transition and starts growing—although the exact moment of that transition is unclear. After that it seems to stabilize around the time that $\lambda_{\max}$ stabilizes, and *Batch Sharpness* stabilizes around $2/\eta$.

**Not a stability measure empirically.** Empirically, the criterion (39) behaves in a way that precludes it from being a quantity that governs stability in a EOS-like fashion, as evidenced by the experiments in Figure 26:

- **The bound is not tight.** In particular, if *empirically* the *second moment contraction* was governing stability of SGD training, the condition of equation (37) would only be violated by a small margin, just like is the case with that condition in the full-batch GD—equivalently, as it is the case with $\lambda_{\max}$ being just slightly above $2/\eta$ during GD training. Instead, *second moment contraction* hovers at 2 (equivalently, the quantity of Equation (39) hovering around 1).

- **No up and down oscillations.** The higher-order-term driven EOS stabilization mechanism of Damian et al. (2023) and the dynamics in Cohen et al. (2024) prescribe a notion of stability to go up and down around the stability threshold. On top of the above point of being significantly above the threshold level, *second moment contraction* does not oscillate in a way prescribed by a stabilization that's based on higher order terms.

- **Inconsistent level of stabilization.** Finally, a notion that would govern SGD dynamics would present a consistent level of stabilization, independent of hyperparameters. This does not happen in the case of *second moment contraction*.

L.3 IMPLICATIONS

**Alignment matters.** The reason why the condition of Wu et al. (2018) (equation (37)) is only a sufficient one, while the condition of Ma & Ying (2021) (equation (7)) is necessary and sufficient, is that the mini-batch Hessians are not commuting/not simultaneously diagonalizable. In particular, they would have been simultaneously diagonalizable if either the model gradients forming their Gauss-Newton approximation were either all the same or all orthogonal, which is hypothetically possible as we have $N \ll d$. Now, in Appendix K it was show that neither is the case—they are misaligned, but not completely orthogonal. Now, the condition (37) not being a governing quantity of EoSS, and not being tight as an upper bound on instability condition, is evidence that this not-complete misalignment has non-trivial effects. That is, unlike in deterministic full-batch gradient descent settings, instability in SGD is dependent on the alignment between notions of curvatures. Therefore, a true notion of stability has to involve a notion of alignment, not only the magnitude of curvature—and, correspondingly, *Batch Sharpness* does consider the alignment, as opposed to, for example, $\lambda_{\max}$ or $\lambda_{\max}^b$.

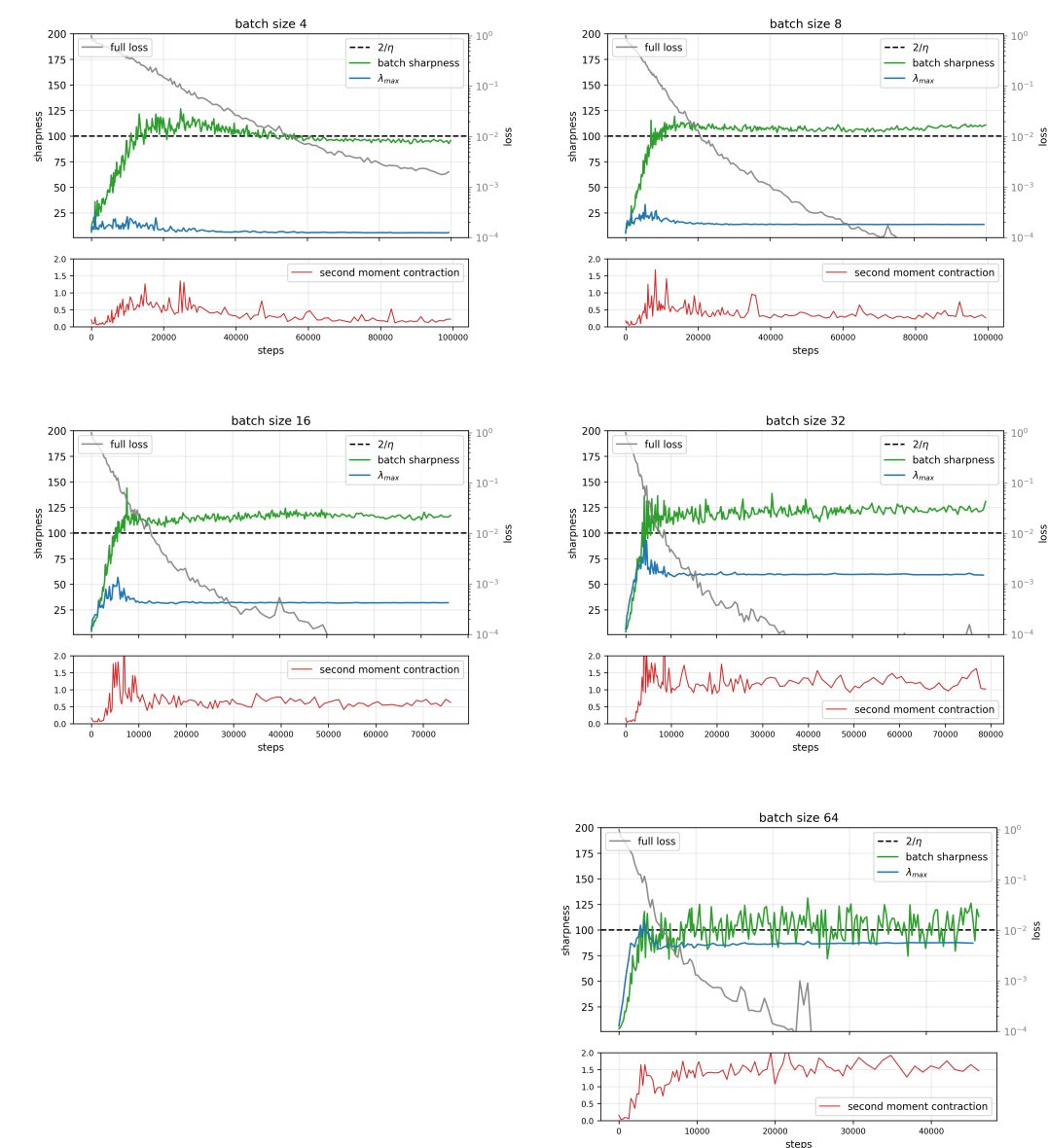

Figure 26: Tracking the condition of equation (39) (the one without the identity), *second moment contraction*, for different batch sizes. MLP, CIFAR-8k, $\eta = 0.02$. Note how the stabilization levels is significantly above the threshold, and inconsistent across batch sizes.

**Importance of instability (not "stability").** We define EOSS as being at the edge of instability. This quantity of equation (37) smaller than 1 is *only* a sufficient condition for stability. The fact that breaking Eq.(37) is not enough for assessing the behavior of SGD is a further proof that what matters is the instability, not the stability. What matters is an inequality that implies divergence if broken, not that implies convergence if satisfied.

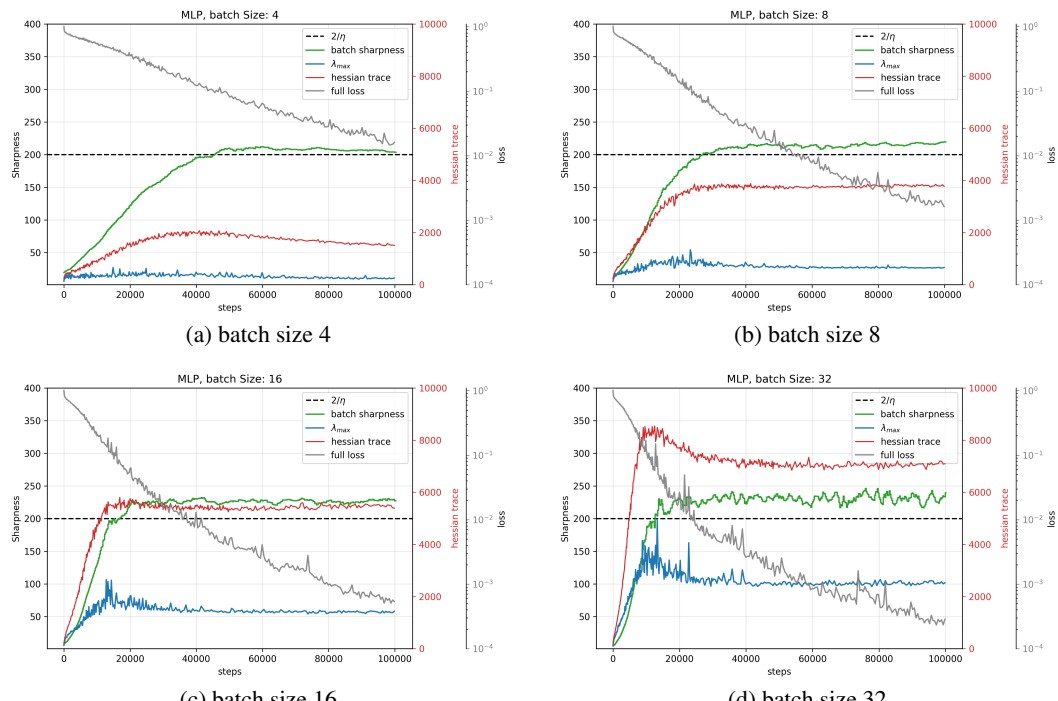

Figure 27: **Trace of the Hessian.** We plot the trace of the full-batch loss Hessian (red), together with the usual *Batch Sharpness* (green) and $\lambda_{\max}$ (blue). Notice that the scale of the trace of the Hessian is much bigger than the rest of the quantities, and it follows the axis on the right (in particular, has no particular relation to $2/\eta$. The plots showcase that trace behaves in a similar manner as $\lambda_{\max}$—its level of stabilization is highly dependent on the batch size, it raises as long as *Batch Sharpness* is rising, and it is stabilizes as batch sharpness stabilizes. Here, we are doing experiments with MLP on CIFAR-10-8k and $\eta = 0.01$

# M    OTHER QUANTITIES OF SGD DYNAMICS

In this Appendix we explore other quantities that describe the SGD dynamics, and discuss their role from the point of view of governing stability. In particular, we are covering the following quantities: trace of full-batch loss Hessian, $\lambda_{\max}^b$ (average max eigenvalues of mini-batch Hessians) and a modified version of *Batch Sharpness*.

## M.1    TRACE OF THE LOSS HESSIAN

A number of works Ma & Ying (2021); Wu & Su (2023); Agarwala & Pennington (2024) have linked the trace of the full-batch loss Hessian to implicit regularization by SGD. We plot in Figure 27 and 28: $\lambda_{\max}$, *Batch Sharpness*, and the trace of the Hessian along the training for a variety of models and batch sizes. We observe here that trace of the Hessian behaves very similarly to the previously studied $\lambda_{\max}$. In particular, it does not have a consistent stabilization level, and depends significantly on the batch size—with smaller batch sizes leading to lower stabilization level of the trace (aka flatter solutions). Also analogous to $\lambda_{\max}$, it undergoes progressive sharpening, as long as *Batch Sharpness* is under $2/\eta$. Analogously, the stabilization of *Batch Sharpness* leads to stabilization of the trace. All of this showcases that trace of the Hessian is not the quantity that governs stability of the SGD dynamics. Yet, it might be a useful indicator of the end phase of progressive sharpening—in the potential situation when we have $\lambda_{\max}$ stabilize, but other eigenvalues continue growing, as illustrated, for example, in Cohen et al. (2024).

It is noteworthy that, in the context of MSE loss combined with piecewise-linear activation functions (e.g., ReLU), the trace of the full-batch loss Hessian coincides with the trace of its Gauss–Newton approximation. Furthermore, under MSE loss, the trace of the Gauss–Newton matrix is equal to the

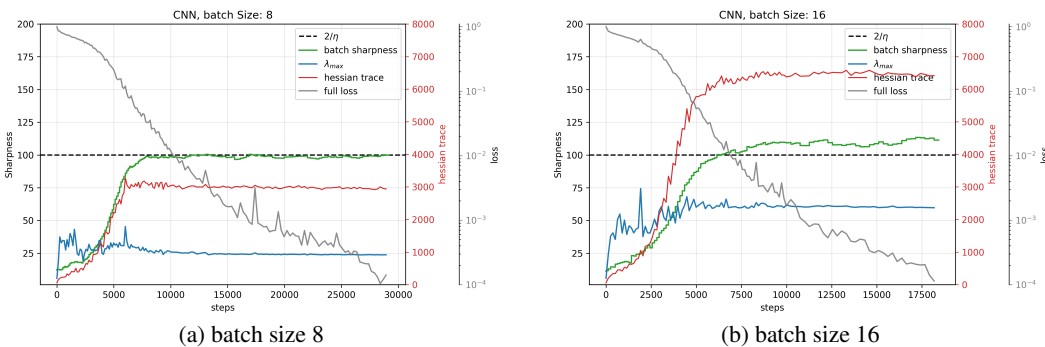

(a) batch size 8          (b) batch size 16

Figure 28: **Trace of the Hessian.** Similar to 27, but for CNN, and with $\eta = 0.02$

trace of the NTK. Consequently, in our setup (MLP with ReLU under MSE) evaluating the trace of the loss Hessian subsumes these cases.

### M.2 $\lambda^b_{\max}$: EXPECTED HIGHEST EIGENVALUE OF MINI-BATCH HESSIANS

In the early versions of this work we have looked at another promising quantity that we term $\lambda^b_{\max}$:

$$\lambda^b_{\max} := \mathbb{E}_{B \sim \mathcal{P}_b}\Big[\lambda_{\max}(\mathcal{H}(L_{\mathbf{B}}))\Big].$$

In particular, the significance of this quantity lies in its characterization of the worst-case sharpness of mini-batch loss landscapes. Yet, the reason why this quantity does not govern SGD dynamics arises from the very phenomenon distinguishing SGD dynamics from full-batch gradient descent—the misalignment of the mini-batch Hessians, see K. Specifically, while individual mini-batch Hessians may exhibit considerable sharpness in their individual directions, these directions typically fail to align, preventing the emergence of a single dominant sharp direction. This scenario closely mirrors the behavior of the operator analyzed in Appendix L, illustrating why *Batch Sharpness*, which dictates the stability of SGD dynamics, relies on both the size of the mini-batch Hessians together with their alignment with the step direction.

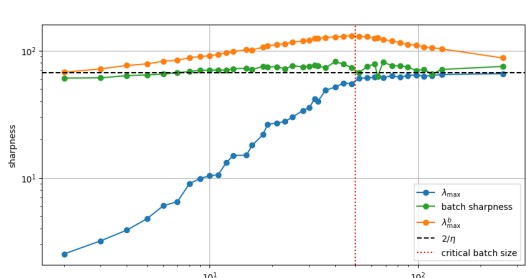

Figure 29: *Final* stabilization levels of *Batch Sharpness*, $\lambda_{\max}$ and $\lambda^b_{\max}$ *vs* batch size. Only the stabilization level of *Batch Sharpness* does not depend on batch size. Setting: MLP, CIFAR10-8k, $\eta = 0.03$

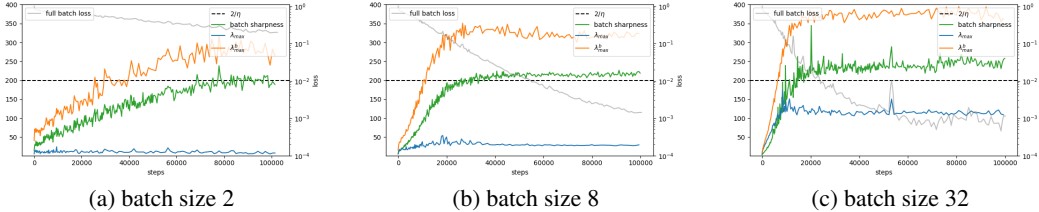

(a) batch size 2       (b) batch size 8       (c) batch size 32

Figure 30: **Behavior of $\lambda^b_{\max}$:** $\lambda^b_{\max}$ stabilizes higher than $2/\eta$, with its stabilization level dependent on the batch size. *Batch Sharpness* and $\lambda_{\max}$ also shown for comparison. Setting: MLP, $\eta = 0.01$, CIFAR10-8k

Consequently, $\lambda^b_{\max}$ stabilizes at a level higher than the threshold $2/\eta$ and *Batch Sharpness*. Moreover, the precise stabilization level is sensitive to the chosen batch size, as showcased in Figure 30, with the dependence of the level of stabilization on batch size shown in Figure 29. For additional

experiments with illustration of behavior of $\lambda_{\max}^b$ refer to Appendix S. Now, this inconsistency of stabilization means that $\lambda_{\max}^b$ does not govern the stability of SGD, and its stabilization is a by-product of EoSS and *Batch Sharpness* stabilization.

In particular, we establish that:

(i) $\lambda_{\max}^b$ also stabilizes.

(ii) $\lambda_{\max}^b$ stabilizes at a level ranging between $2/\eta$ and $4/\eta$. The level is lower for very small and very large batch sizes, and higher for intermediate batch sizes.

(iii) $\lambda_{\max}^b$ increases concurrently with *Batch Sharpness* and stabilizes simultaneously, indicating insights into the nature of *Batch Sharpness* growth and progressive sharpening.

(iv) $\lambda_{\max}^b$ is by construction greater than both $\lambda_{\max}$ and *Batch Sharpness*. The stabilization of *Batch Sharpness* around $2/\eta$ for SGD and $\lambda_{\max}$ for GD ensures that $\lambda_{\max}^b$ stabilizes at or above $2/\eta$ in the EoS/EoSS regime.

Concerning (iv), the inequality $\lambda_{\max}^b \geq$ *Batch Sharpness* follows directly from the definition of *Batch Sharpness* as an expectation of Rayleigh quotients. Furthermore, the inequality $\lambda_{\max}^b \geq \lambda_{\max}$ results from the following reasoning. The largest singular value of the Hessian matrix derived from single data points is positive. This observation is crucial in establishing the following well-known property of matrix eigenvalues.

**Lemma 6.** *Let $m, b \in \mathbb{N}$ and consider $m$ matrices $M_1, M_2, \ldots, M_b \in \mathbb{R}^{m \times m}$ satisfying $\lambda_{\max} > |\lambda_{\min}|$. Then, the largest eigenvalue of their sum satisfies*

$$\lambda_{\max}\left(\sum_{i=1}^b M_i\right) \leq \sum_{i=1}^b \lambda_{\max}(M_i) \tag{40}$$

*with equality only if all $M_i$ are identical.*

This lemma is a direct consequence of the convexity of the operator norm in matrices and the fact that the largest eigenvalue is positive in our setting. In our setting, it implies that with non-simultaneously-diagonalizeable matrices, the maximum eigenvalue of the sum is strictly less than the sum of the maximum eigenvalues of the individual matrices. To illustrate, consider eigenvalue sequences for batch sizes that are powers of four, though the result generalizes to any $b_1 < b_2$:

$$\lambda_{\max}^1 > \lambda_{\max}^4 > \lambda_{\max}^{16} > \lambda_{\max}^{64} > \lambda_{\max}^{256} > \ldots \tag{41}$$

Importantly, this ordering is the case only for "static" model – i.e. when we take a model, and without changing the weights, evaluate $\lambda_{\max}^b$ as we change the batch size.

As noted in point (ii) above, this ordering does not hold in the *trained* case, as different batch sizes affect also the progressive sharpening and the nature of the mini-batch Hessians. Specifically, since *Batch Sharpness* stabilizes at $2/\eta$ at EoSS, the level of stabilization of $\lambda_{\max}^b$ depends on its relation to *Batch Sharpness*. Since the two are quite similar, with the difference that *Batch Sharpness* also takes into account the alignment of the mini-batch landscapes sharpest directions with the mini-batch gradients—the gap between $\lambda_{\max}^b$ and *Batch Sharpness* is governed precisely by this alignment. As illustrated in Figure 29, the level of stabilization of $\lambda_{\max}^b$ is similar to that of *Batch Sharpness* for very small and very large batch sizes. For large batch sizes, this result is straightforward, as the dynamics approach full-batch GD, in which all relevant quantities equalize at EoS. Conversely, the small-batch case emerges because, for smaller batch sizes, the mini-batch Hessian (or its Gauss-Newton approximation) comprises averages of only a few per-sample *model* gradient outer products, causing mini-batch gradients to align closely with the largest eigenvalues.

This alignment diminishes as the batch size increases, leading to a widening gap between *Batch Sharpness* and $\lambda_{\max}^b$. Intriguingly, our experiments reveal that this gap only widens up to the afore-mentioned *critical batch size*, which also serves as a switch between SGD and GD dynamics from the point of view of $\lambda_{\max}$ stabilization. Beyond the *critical batch size* the gap begins to narrow again, as depicted in Figure 29. Clarifying this phenomenon fully would be a key outcome of a comprehensive theory of progressive sharpening and SGD stability.

Another significant consequence of the stabilization of $\lambda_{\max}^b$ is that it provides insights into the mechanisms underlying the progressive sharpening of *Batch Sharpness*. Specifically, the growth in

*Batch Sharpness* could be attributed either to a general increase in the sharpness of the mini-batch landscapes or to an increase in alignment between mini-batch Hessians and gradients. Notably, throughout the period of *Batch Sharpness* increase, both $\lambda_{\max}^b$ and the trace of the loss Hessian consistently rise and stabilize simultaneously with *Batch Sharpness*. This suggests that at least portion of the increase in *Batch Sharpness* arises from the overall sharpening of the mini-batch landscapes, rather than solely from alignment of the mini-batch gradients and Hessians. Consequently, *Batch Sharpness* appears closely linked with the progressive sharpening phenomenon itself, with its eventual stabilization marking the end of progressive sharpening. This points to the fact that *Batch Sharpness* is closely connected to progressive sharpening.

### M.3 MODIFIED BATCH SHARPNESS

In the earlier versions of this work we also looked at a modified definition of *Batch Sharpness*:

**Definition 7** (Modified *Batch Sharpness*). We call Modified *Batch Sharpness* the quantity defined as

$$\text{Modified } \textit{Batch Sharpness}(\theta) \quad := \quad \frac{\mathbb{E}_{B\sim\mathcal{P}_b}\Big[\nabla L_B(\theta)^\top \mathcal{H}(L_B)\nabla L_B(\theta)\Big]}{\mathbb{E}_{B\sim\mathcal{P}_b}\Big[\|\nabla L_B(\theta)\|^2\Big]}.$$

The difference from the definition of *Batch Sharpness* is that in this one the expectation over batches in taken inside the fraction. The intuition for this quantity comes from a notion of average stability on mini-batch landscapes. That is,

$$\frac{\mathbb{E}_{B\sim\mathcal{P}_b}\Big[\nabla L_B(\theta)^\top \mathcal{H}(L_B)\nabla L_B(\theta)\Big]}{\mathbb{E}_{B\sim\mathcal{P}_b}\Big[\|\nabla L_B(\theta)\|^2\Big]} \leq 2/\eta \quad \Longleftrightarrow \quad \mathbb{E}\Big[L_B(\theta_{t+1}^B) - L_B(\theta_t)\Big] \geq 0$$

where $\theta_{t+1}^B$ is the parameters that we are getting if we are stepping on the given mini-batch. This means that *Modified Batch Sharpness* $< 2/\eta$ is equivalent to "on average, the mini-batch loss does not increase when stepping the corresponding landscape". This formulation is an attempt to extend the descent lemma to the mini-batch landscapes that govern the SGD dynamics insted of the descent lemma on the full-batch landscape that govern GD. Empirically, it turns out that *Modified Batch Sharpness* also stabilizes, but its stabilization level is higher than that of the *Batch Sharpness* and therefore $2/\eta$, as illustrated in Figure 31. Moreover, its stabilization level is dependent on the batch size.

**Modified *Batch Sharpness* and mini-batch gradients.** Importantly, we show in Proposition 5 that Modified *Batch Sharpness* is a valid Instability Criterion and it governs the explosion of the expectation of the norm squared of the mini-batch gradients.

$$\text{Modified } \textit{Batch Sharpness}(\theta_t) > 2/\eta + c\eta \quad \Longrightarrow \quad \mathbb{E}_{B\sim\mathcal{P}_b}[\|\nabla L_B(\theta_{t+1})\|_2^2] > \mathbb{E}_{B\sim\mathcal{P}_b}[\|\nabla L_B(\theta_t)\|_2^2].$$

## N MODIFIED BATCH SHARPNESS IS A VALID INSTABILITY CRITERION

We show here that Modified Batch Sharpness (Definition 7) is a valid instability criterion (Definition 4).

> *Importantly, while it is a valid instability criterion, it does not stabilize at $2/\eta$ in practice, thus it is not the quantity that self-stabilization tames, but its stabilization is a byproduct of Batch Sharpness stabilizing.*

We now compute what the update of the norm of the gradients $\mathbb{E}_i[\|Y_i\|_2^2]$ after one step is. Precisely, with the notations of Appendix G, we are computing here the value of $\mathbb{E}_t\mathbb{E}_i[\|Y_i^{t+1}\|_2^2]$ so the average over the iterations of the update to the quantity $\mathcal{C}$ above. Precisely we here prove the following Proposition.

**Proposition 5.** *There exists an absolute constant $c > 0$ such that when Modified Batch Sharpness $> 2/\eta + c\eta$, then $\mathbb{E}\|\nabla L_B\|_2^2$ increases in size exponentially and the trajectory diverges (is quadratically unstable, see Definition 5).*

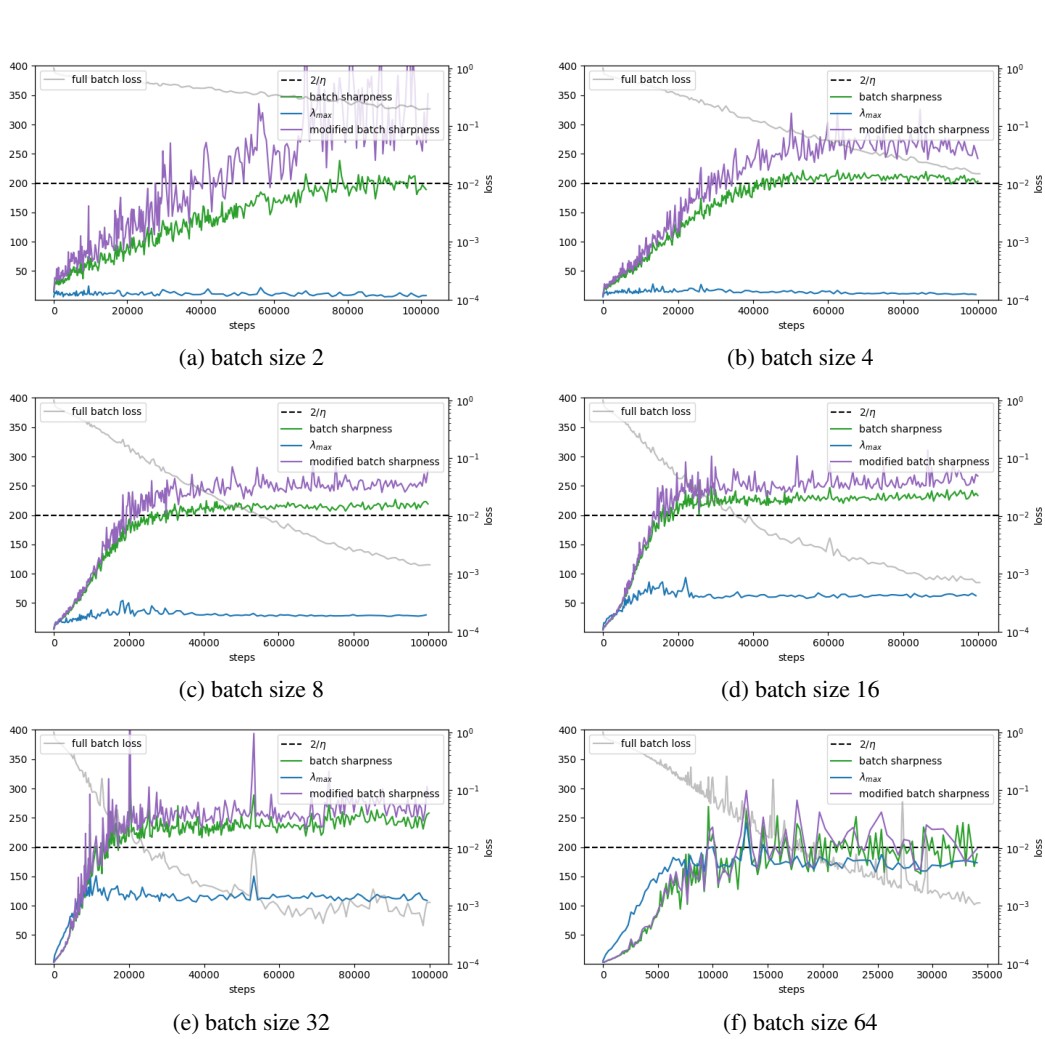

Figure 31: **Modified Batch Sharpness**: Behavior of *Modified Batch Sharpness*, definition of which is similar to *Batch Sharpness*, but with the expectation "inside" the fraction ($\mathbb{E}[\nabla L_B(\theta)^T H_B \nabla L_B(\theta)]/\mathbb{E}[\|\nabla L_B(\theta)\|^2]$). It stabilizes above $2/\eta$ with its stabilization level highly dependent on the batch size.

*Proof.* In the proof we use the notations of Appendix G.

**Step 1: One step on the gradient's second moment** . Remind that the SGD iterate satisfy

$$\theta_{t+1} \;=\; \theta_t - \eta\, Y_{j_t}(\theta_t), \quad i_t \overset{\text{i.i.d.}}{\sim} \mathcal{D},$$

and define a *fresh*, independent index $j$ used only for the outer expectation in $\mathcal{C}^{t+1}$. Because $j \perp i_t$ we may write

$$Y_i(\theta_{t+1}) \;=\; H_i\big(\theta_{t+1} - x_i\big) = Y_i(\theta_t) \;-\; \eta\, H_i Y_{j_t}(\theta_t).$$

Squaring, expanding, and averaging over $j$ gives

$$\mathcal{C}^{t+1} = \mathbb{E}_i \big\| Y_i(\theta_t) - \eta H_i Y_{j_t}(\theta_t) \big\|^2$$
$$= \mathcal{C}^t - 2\eta \underbrace{\mathbb{E}_{i,j_t}\big[Y_i(\theta_t)^\top H_i Y_{j_t}(\theta_t)\big]}_{\text{cross term}} + \eta^2 \underbrace{\mathbb{E}_{i,j_t}\big[Y_{j_t}(\theta_t)^\top H_i^2 Y_{j_t}(\theta_t)\big]}_{\text{variance term}}. \tag{42}$$

**Step 2: Decoupling the indices.** Note that

$$2\mathbb{E}_{i,j_t}\big[Y_i(\theta_t)^\top H_i Y_{j_t}(\theta_t)\big] \;=\; \mathcal{A} - \mathcal{B} - \widetilde{\Delta}. \tag{43}$$

This implies that we can rewrite

$$\mathcal{C}^{t+1} = \mathcal{C}^t - \eta\,(\mathcal{A} - \mathcal{B} - \widetilde{\Delta}) + \eta^2\,(\text{variance term}). \tag{44}$$

Next note that if we are at the EoSS, then $\mathcal{A} \approx \frac{2}{\eta}(1+\delta)\mathcal{C}$ for some $\delta \in \mathbb{R}$. This implies that we can rewrite the term above as

$$\mathcal{C}^{t+1} \;\approx\; -(1+2\delta)\,\mathcal{C}^t \;+\; \underbrace{\eta\mathcal{B} + \eta\widetilde{\Delta} + \eta^2\,(\text{variance term})}_{\text{rest}}. \tag{45}$$

Let us know understand the size of the rest, the trajectory diverges if and only if:

$$\eta\mathcal{B} + \eta\widetilde{\Delta} + \eta^2 \mathbb{E}_{i,j_t}\big[Y_{j_t}(\theta_t)^\top H_i^2 Y_{j_t}(\theta_t)\big] \;>\; 2(1+\delta)\,\mathcal{C}^t. \tag{46}$$

Next note that by applying Jensen inequality to the term multiplied by $\eta^2$ we obtain that

$$\sqrt{\underbrace{\mathbb{E}_{i,j_t}\big[Y_{j_t}(\theta_t)^\top H_i^2 Y_{j_t}(\theta_t)\big]}_{\text{variance term}} \cdot \underbrace{\mathbb{E}_i\big[Y_i(\theta_t)^\top Y_i(\theta_t)\big]}_{\mathcal{C}}} \;\geq\; \underbrace{\mathbb{E}_{i,j_t}\big[Y_{j_t}(\theta_t)^\top H_i \cdot Y_i(\theta_t)\big]}_{\mathcal{D}}. \tag{47}$$

**Step 3: Final algebra.** Plugging this above, we obtain that the trajectory diverges when

$$\eta\mathcal{B} + \eta\widetilde{\Delta} + \eta^2 \frac{\mathcal{D}^2}{\mathcal{C}} \;>\; 2(1+\delta)\,\mathcal{C}. \tag{48}$$

Again applying (43) we obtain that this is equivalent to

$$\eta\mathcal{B} + \eta\widetilde{\Delta} + \eta^2 \frac{(\mathcal{A} - \mathcal{B} - \widetilde{\Delta})^2}{4\mathcal{C}} \;>\; 2(1+\delta)\,\mathcal{C}. \tag{49}$$

Since $\eta\mathcal{A} = 2(1+\delta)\mathcal{C}$, then $\eta^2\mathcal{A}^2 = 4(1+\delta)^2\mathcal{C}^2$ to asking

$$\eta\mathcal{B} + \eta\widetilde{\Delta} + \eta^2 \frac{\mathcal{B}^2 + \widetilde{\Delta}^2 - 2\mathcal{A}\widetilde{\Delta} - 2\mathcal{A}\mathcal{B} + 2\mathcal{B}\widetilde{\Delta}}{4\mathcal{C}} \;>\; 2(1+\delta)\,\mathcal{C} - \frac{4(1+\delta)^2\mathcal{C}^2}{4\mathcal{C}}. \tag{50}$$

Furthermore, equivalent to asking

$$\eta\mathcal{B} + \eta\widetilde{\Delta} - \frac{2(1+\delta)}{2}\eta\widetilde{\Delta} - \frac{2(1+\delta)}{2}\eta\mathcal{B} + \eta^2 \frac{\mathcal{B}^2 + \widetilde{\Delta}^2 + 2\mathcal{B}\widetilde{\Delta}}{4\mathcal{C}} \;>\; (1 - \delta + \delta^2)\,\mathcal{C} \tag{51}$$

or, even further simplified

$$\eta\delta(\mathcal{B} + \widetilde{\Delta}) + \eta^2 \frac{(\mathcal{B} + \widetilde{\Delta})^2}{4\mathcal{C}} \;>\; (1 - \delta + \delta^2)\,\mathcal{C}. \tag{52}$$

We can rewrite this as

$$\eta\delta(\mathcal{A} - 2\mathcal{D}) + \eta^2 \frac{(\mathcal{A} - 2\mathcal{D})^2}{4\mathcal{C}} \;>\; (1 - \delta + \delta^2)\,\mathcal{C}. \tag{53}$$

By plugging, as before, $\eta\mathcal{A} = 2(1 + \delta)\mathcal{C}$ we obtain

$$2\delta(1 + \delta)\mathcal{C} - 2\eta\delta 2\mathcal{D} - 2\eta(1 + \delta)\mathcal{D} + \eta^2 \frac{\mathcal{D}^2}{\mathcal{C}} \;>\; \left(1 - \delta + \delta^2 - (1 + \delta)^2\right)\mathcal{C} \tag{54}$$

which simplifies as

$$\underbrace{2\eta(1 + 2\delta)\mathcal{D}}_{\mathcal{O}(\eta^2)} - \underbrace{\eta^2 \frac{\mathcal{D}^2}{\mathcal{C}}}_{\mathcal{O}(\eta^4)} \;<\; \delta(5 + 2\delta)\,\underbrace{\mathcal{C}}_{\mathcal{O}_\eta(1)}. \tag{55}$$

Thus there exists a constant $c > 0$, such that if $\delta > c\eta^2$ the trajectory diverges exponentially, if $\delta < c\eta^2$ the trajectory is stable.

$\square$

## O   HARDWARE & COMPUTE REQUIREMENTS

All experiments were executed on a single NVIDIA A100 GPU (80 GB) with 256 GB of host RAM. The software stack comprises Python 3.12 and PyTorch 2.5.1 (built with the default CUDA toolchain supplied by the wheel).

**Baseline MLP (2M parameters, Section Q)**   Training for 100k steps on the 8 k-image CIFAR-10 subset finishes in $\approx 5$ min wall-clock while computing step sharpness every 8 steps, batch sharpness every 128 steps and $\lambda_{\max}$ every 256 steps. Peak device memory is 14 GB during ordinary training and $\approx 70$ GB while estimating $\lambda_{\max}$ on a 32k subset, comfortably fitting the 80 GB card.

**Algorithmic caveats.**   We rely on power iteration for $\lambda_{\max}$; while Lanczos would reduce the number of Hessian–vector products, the official PyTorch implementation remains CPU-only. To offset the extra memory incurred by double backward, we cache the first forward pass; batching $\lambda_{\max}$ is left to future work.

## P   THE HESSIAN AND THE FISHER INFORMATION MATRIX OVERLAP

In the theoretical analysis of stability of SGD dynamics it is assumed that the loss Hessian can be well-approximated by its Gauss-Newton approximation—in particular, it is often assumed we are at the minima, where there is an equality between the two. Concretely, having $C$ classes, we have:

$$L(\theta) \;=\; \frac{1}{N} \sum_{i=1}^{N} \ell\big(z_i(\theta),\, y_i\big), \qquad z_i(\theta) \;=\; f_\theta(x_i) \in \mathbb{R}^C.$$

$$J_i := \frac{\partial z_i(\theta)}{\partial\theta} \in \mathbb{R}^{C\times d}, \quad g_{z,i} := \nabla_z \ell\big(z_i(\theta), y_i\big) \in \mathbb{R}^C, \quad H_{z,i} := \nabla_z^2 \ell\big(z_i(\theta), y_i\big) \in \mathbb{R}^{C\times C},$$

and for $j = 1, \ldots, C$, let $\nabla_\theta^2 f_j(x_i) \in \mathbb{R}^{d\times d}$ denote the Hessian (w.r.t. $\theta$) of the $j$-th output component. With this notation, we have:

$$\nabla_\theta L(\theta) \;=\; \frac{1}{N} \sum_{i=1}^{N} J_i^\top g_{z,i}.$$

and

$$\nabla_\theta^2 L(\theta) = \frac{1}{N} \sum_{i=1}^{N} \left( J_i^\top H_{z,i} J_i \;+\; \sum_{j=1}^{C} \big[g_{z,i}\big]_j \nabla_\theta^2 f_j(x_i) \right)$$

$$= \underbrace{\frac{1}{N} \sum_{i=1}^{N} J_i^\top H_{z,i} J_i}_{\text{Gauss-Newton approx}} \;+\; \underbrace{\frac{1}{N} \sum_{i=1}^{N} \sum_{j=1}^{C} \big[g_{z,i}\big]_j \nabla_\theta^2 f_j(x_i)}_{\text{remainder / model-curvature term}}.$$

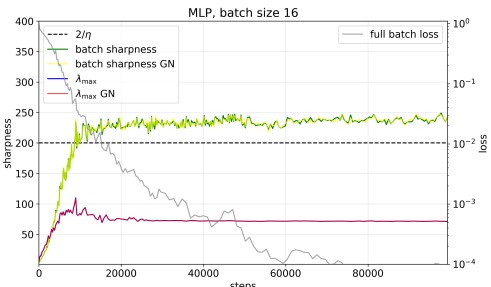 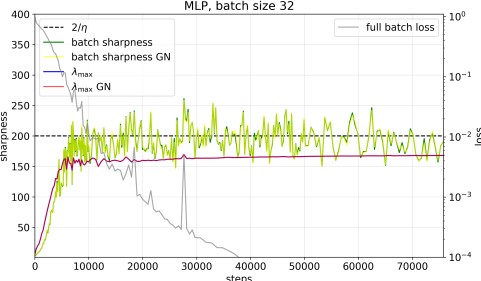

Figure 32: **Gauss–Newton approximation (MLP).** Comparison of *Batch Sharpness* and $\lambda_{\max}$ computed with the true loss Hessian and its Gauss–Newton approximation, showing the validity of approximation. Both of the lines overlap almost perfectly

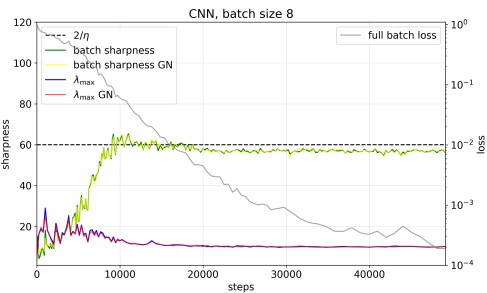 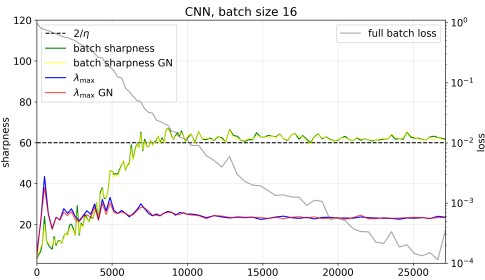

Figure 33: **Gauss–Newton approximation (CNN).** Comparison of *Batch Sharpness* and $\lambda_{\max}$ computed with the true loss Hessian and its Gauss–Newton approximation, showing the validity of approximation. Both of the lines overlap almost perfectly

For MSE, this simplifies to:

$$\nabla_\theta^2 \mathcal{L}(\theta) = \underbrace{\frac{1}{N}\sum_{i=1}^{N} J_i^\top J_i}_{\text{Gauss-Newton for MSE}} + \underbrace{\frac{1}{N}\sum_{i=1}^{N}\sum_{j=1}^{C} r_{i,j}\,\nabla_\theta^2 f_j(x_i)}_{\text{remainder / model-curvature term}}.$$

in particular, if we are at minima, the residuals are 0, so the second terms completely disappears.

Yet, the dynamics enter the EOSS regime away from minima (which is showcased by the continued decrease of the loss), where the Gauss-Newton approximation might not hold. In this Appendix we illustrate empirically that the Gauss-Newton approximation is close to the actual loss Hessian—at least from the perspective of EOSS and SGD stability. In particular, we compare *Batch Sharpness*, $\lambda_{\max}$ and $\lambda_{\max}^b$ when computed on the actual loss Hessian and on its Gauss-Newton approximations. Figures 32 to 34 illustrate that the computed quantities coincide throughout the *whole* training. Notice that due to the fact that the Gauss-Newton approximation and the NTK have the same spectrum, the $\lambda_{\max}$ and $\lambda_{\max}^b$ results also apply to the highest eigenvalues of the full-batch and mini-batch NTKs ($\frac{1}{B} J_B J_B^\top$). Note that this agrees with the findings in literature, see e.g. Papyan (2019), but it was not clear whether the Gauss-Newton approximation holds before convergence. Our experiments demonstrate that it does the throughout the whole training, in particular, during EOSS.

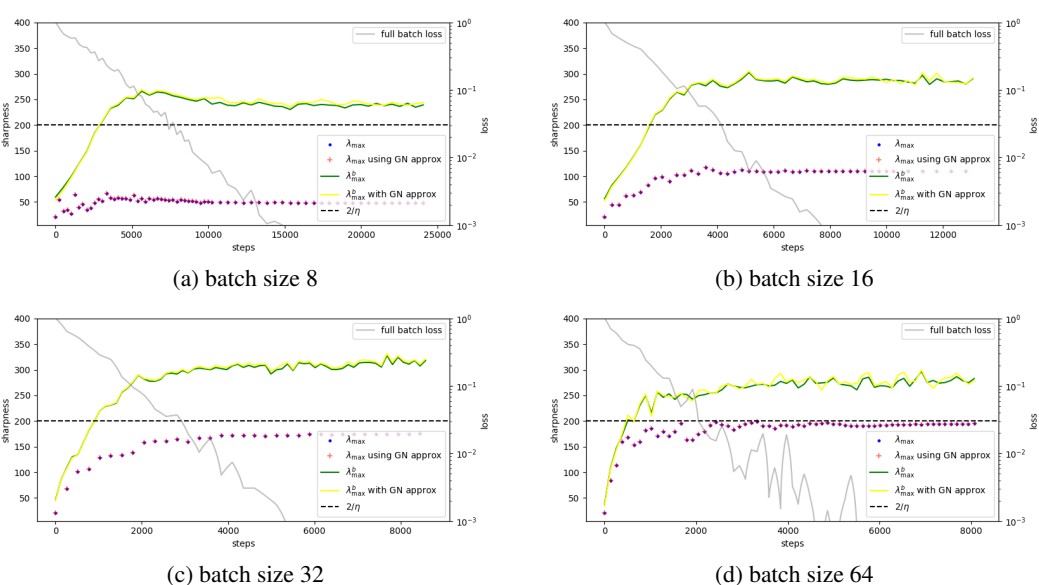

(a) batch size 8      (b) batch size 16

(c) batch size 32      (d) batch size 64

Figure 34: **Gauss-Newton approximation.** Showcasing the correctness of using the Gauss-Newton approximation to the loss Hessian by comparing the $\lambda_{\max}$ and $\lambda_{\max}^b$ computed with the true loss Hessian and its GN-approximation across different batch sizes.

## Q    ILLUSTRATION OF EOSS IN VARIETY OF SETTINGS: *Batch Sharpness*

In this appendix, we provide further empirical evidence that EOSS arises robustly across a variety of architectures, step sizes, and batch sizes. For each experiment, we plot three quantities: $\lambda_{\max}$, *Batch Sharpness*, and *step sharpness* as a point cloud, which constitutes *Batch Sharpness* without expectation, and measured only on the *current* batch. Notice that time-averaging *step sharpness* is approximately the same as taking expectation over batches (albeit with slowly changing parameters), so it is approximately equal to *Batch Sharpness*, which takes this expectation at a point. Consistent with our main observations, we find that *Batch Sharpness* invariably stabilizes around $2/\eta$. Also refer to Section S for additional experiments illustrating EOSS

**MLP (2-Layer) Baseline.**    Figure 36 illustrates EOSS for our baseline network, an MLP with two hidden layers of dimension 512, trained on an $8192$-sample subset of CIFAR-10 with step size $\eta = 0.01$. As the training proceeds, *Batch Sharpness* stabilizes around $2/\eta$, whereas $\lambda_{\max}$ plateaus strictly below *Batch Sharpness*.

**5-Layer CNN.**    We further confirm the EOSS regime in a five-layer CNN. As depicted in Figures 37, *Batch Sharpness* continues to plateau near the instability threshold for two distinct step sizes, while $\lambda_{\max}$ once again settles at a lower level. Notably, as we vary the batch size, the gap between *Batch Sharpness* and $\lambda_{\max}$ increases for smaller batches, mirroring the patterns described in Section I.

**ResNet-14.**    Finally, we demonstrate that the EOSS regime also emerges for a deeper, residual architectures. In our case we are using RESNET-14 without BatchNor. Figure 41 highlights the same qualitative behavior, with *Batch Sharpness* stabilizing at $2/\eta$.

Overall, these experiments provide further confirmation that EOSS is a robust phenomenon across different architectures, step sizes, and batch sizes.

**CNN with Full CIFAR-10.**    We also demonstrate in Figure 35 the emergence of EOSS when training on the full CIFAR-10 dataset. Consistent with the rest of the experiments, *Batch Sharpness* consistently stabilizes at $2/\eta$. Notably, in these experiments we also include a plot of the accuracy on the training set, to illustrate that EOSS happens away from the manifold of minima, and thus cannot be attributed solely to the structure around the manifold of minima.

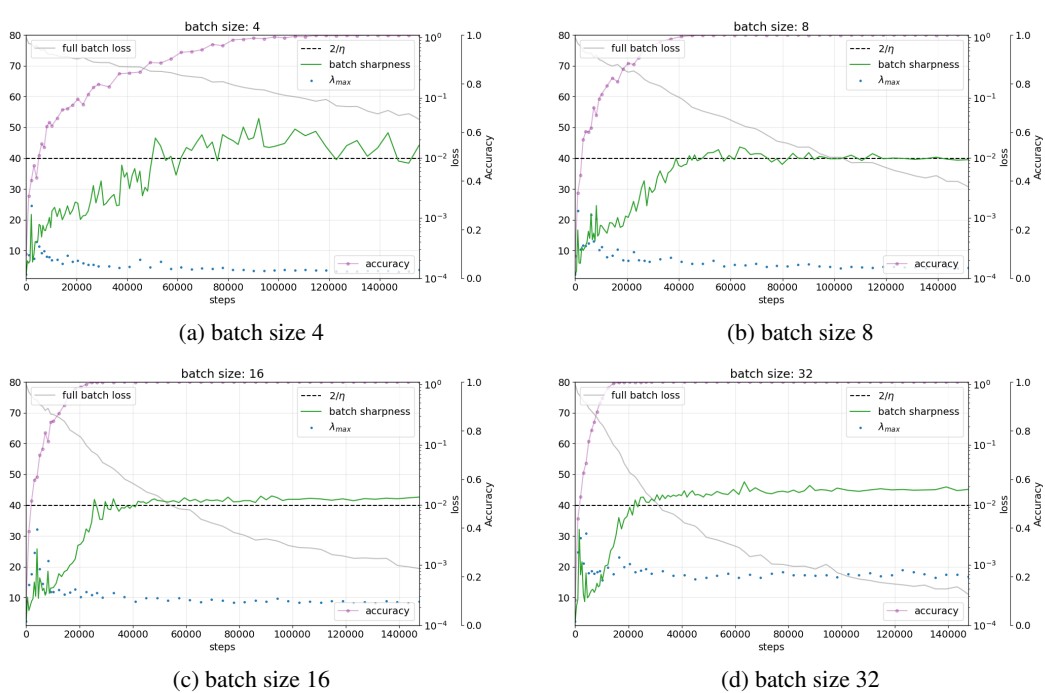

(a) batch size 4

(b) batch size 8

(c) batch size 16

(d) batch size 32

Figure 35: **CNN on Full CIFAR-10:** Same architecture as in Figure 37, but trained at a larger $\eta = 0.05$ on the full CIFAR-10 dataset, illustrating the emergence of EoSS away from manifold of minima.

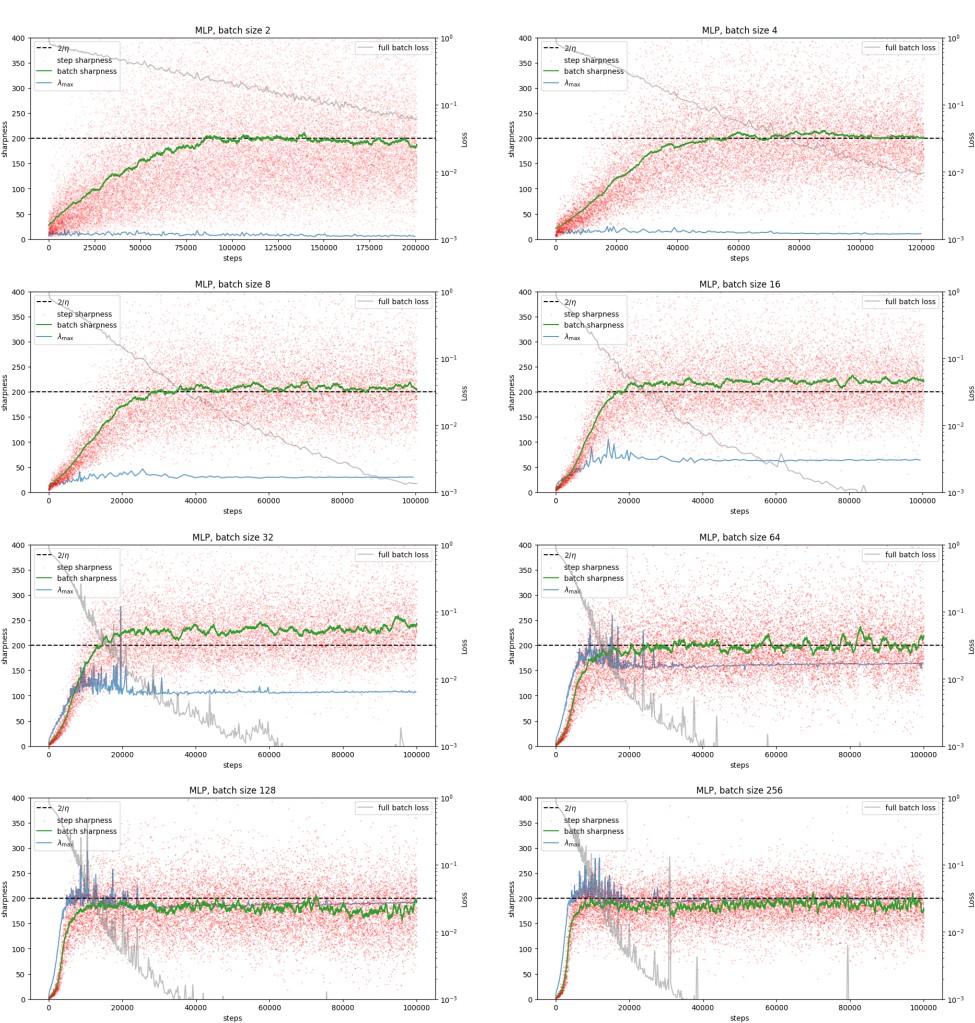

Figure 36: **MLP**: 2 hidden layers, hidden dimension 512; **step size 0.01**, 8k subset of CIFAR-10. Comparison between: step sharpness, aka batch sharpness without expectation over batches and measured on the current batch (red dots, time-smoothing would be $\approx$ *Batch Sharpness*), the empirical *Batch Sharpness* (green line), the $\lambda_{\max}$ (blue line).

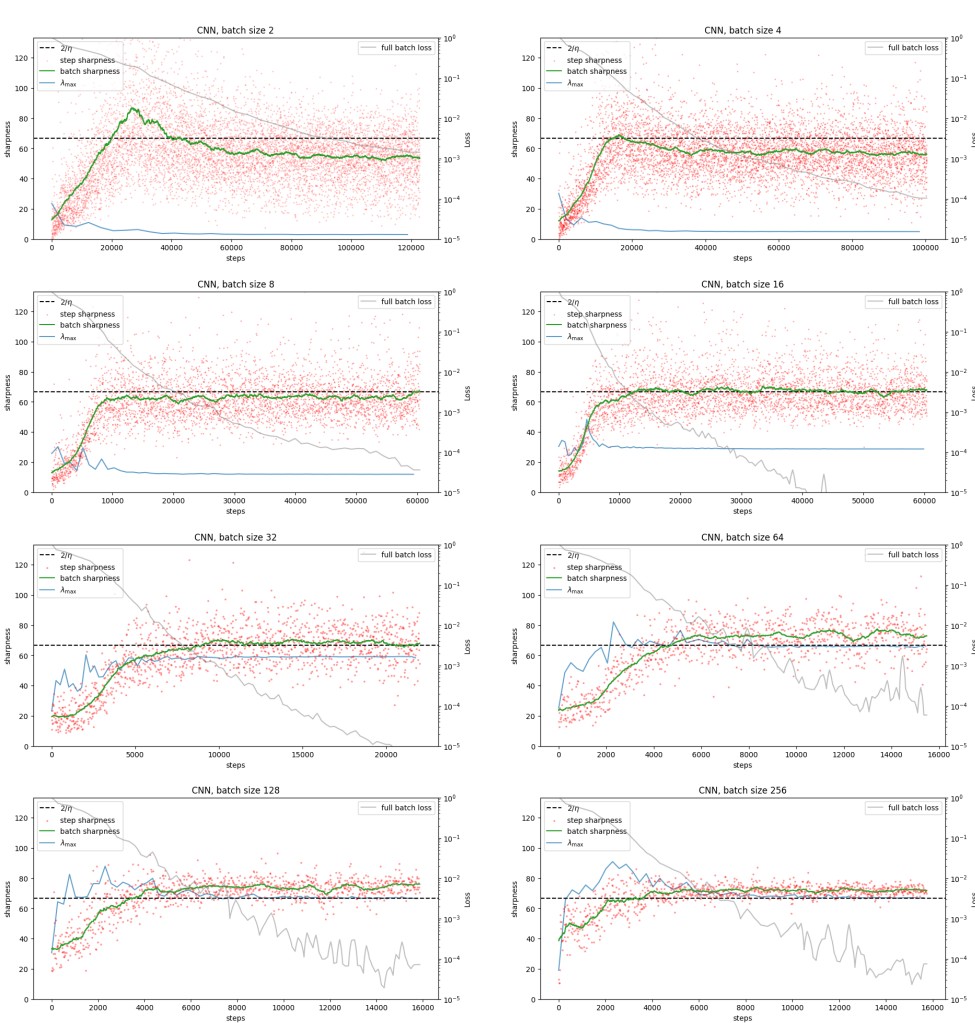

Figure 37: **CNN**: 5 layers (3 convolutional, 2 fully-connected), **step size 0.03**, 8k subset of CIFAR-10. Comparison between: step sharpness, aka batch sharpness without expectation over batches and measured on the current batch (red dots, time-smoothing would be $\approx$ *Batch Sharpness*), the empirical *Batch Sharpness* (green line), the $\lambda_{\max}$ (blue line).

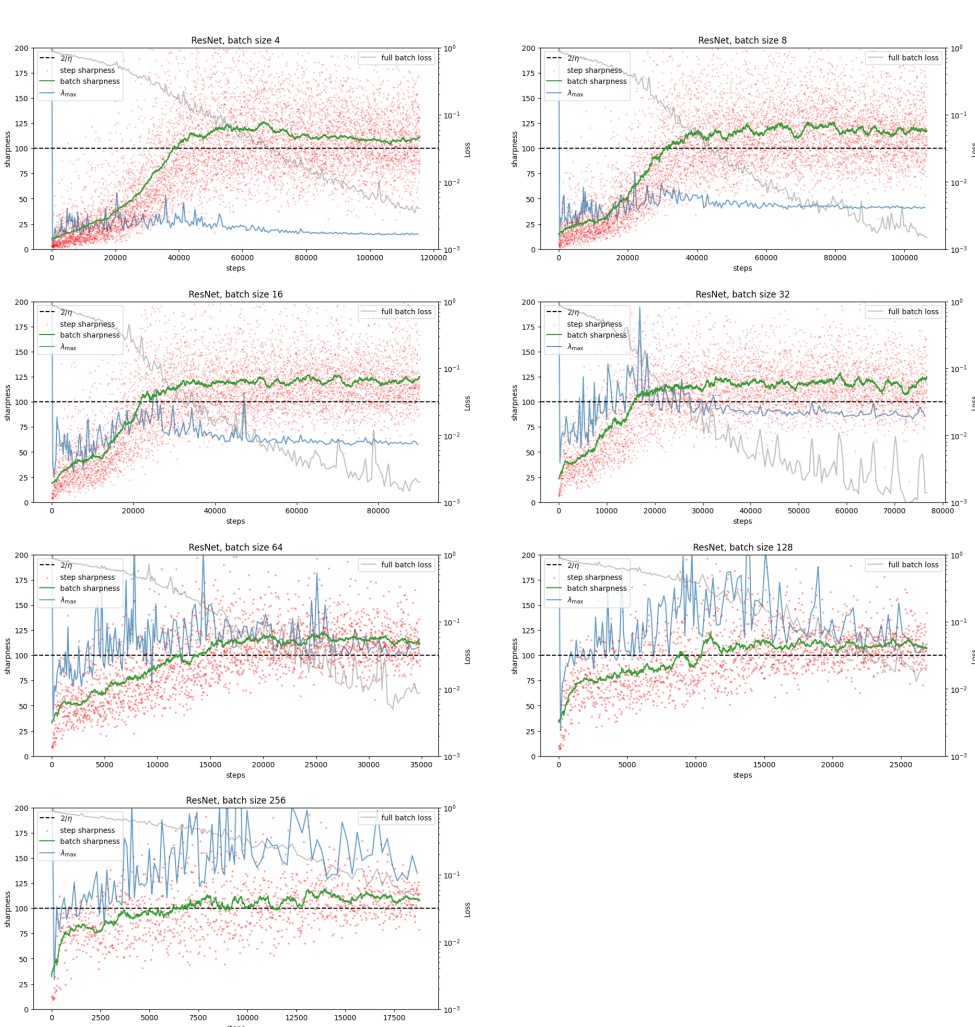

Figure 38: **ResNet-10**, step size 0.005, 8k subset of CIFAR-10. Comparison between: step sharpness, aka batch sharpness without expectation over batches and measured on the current batch (red dots, time-smoothing would be $\approx$ *Batch Sharpness*), the empirical *Batch Sharpness* (green line), the $\lambda_{\max}$ (blue line).

# R ILLUSTRATION OF EOSS FOR THE SVHN DATASET

This appendix complements Appendix Q by verifying that the EoSS phenomena are not specific to CIFAR-10 but persist under a change of dataset. We repeat the experiments of Appendix Q—sweeping architectures (MLP, CNN, ResNet) and batch sizes—on an 8k subset of the SVHN dataset, and track step sharpness, Batch Sharpness, and the full-batch $\lambda_{\max}$ along the training trajectory. Across all settings we again observe progressive sharpening followed by stabilization of Batch Sharpness at $2/\eta$, catapult-like spikes, and suppression of $\lambda_{\max}$ below $2/\eta$, mirroring the behavior seen on CIFAR-10 and supporting the claim that EoSS is a robust feature of mini-batch SGD on standard vision benchmarks.

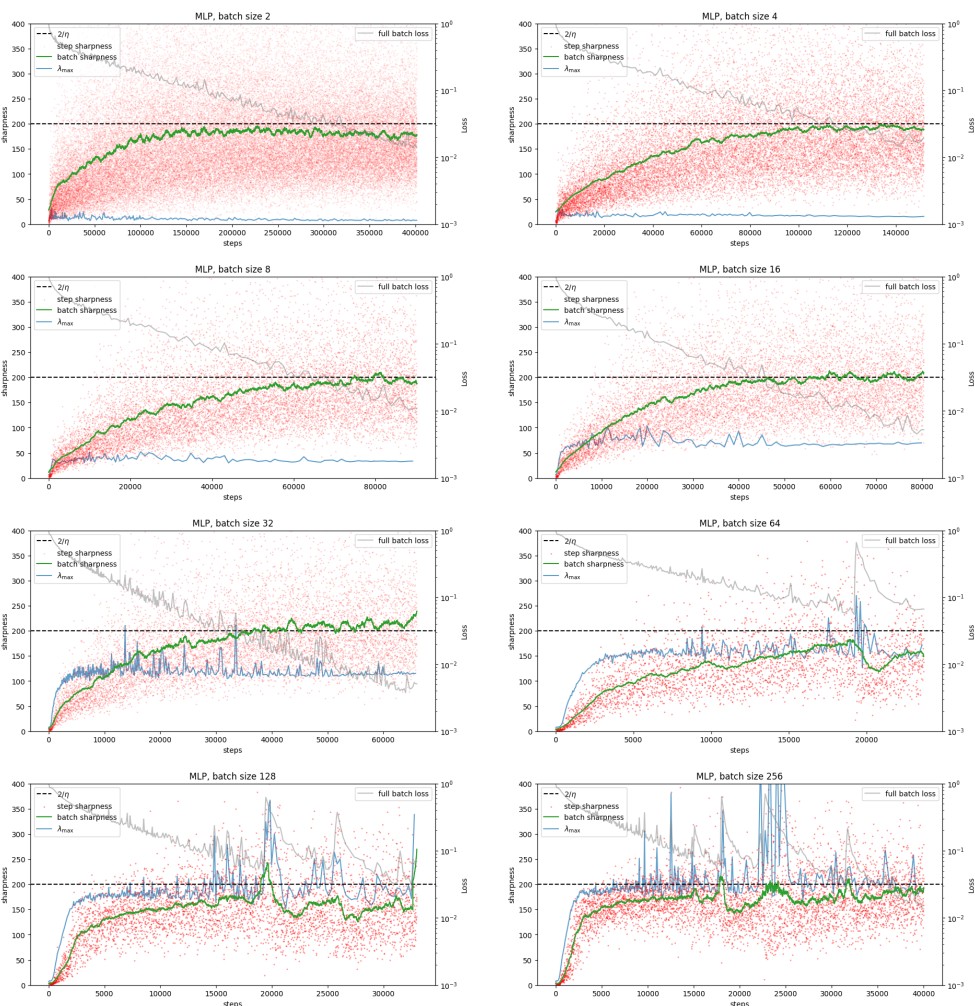

Figure 39: **MLP**: 2 hidden layers, hidden dimension 512; **step size 0.01**, 8k subset of **SVHN**. Comparison between: step sharpness, aka batch sharpness without expectation over batches and measured on the current batch (red dots, time-smoothing would be $\approx$ *Batch Sharpness*), the empirical *Batch Sharpness* (green line), the $\lambda_{\max}$ (blue line).

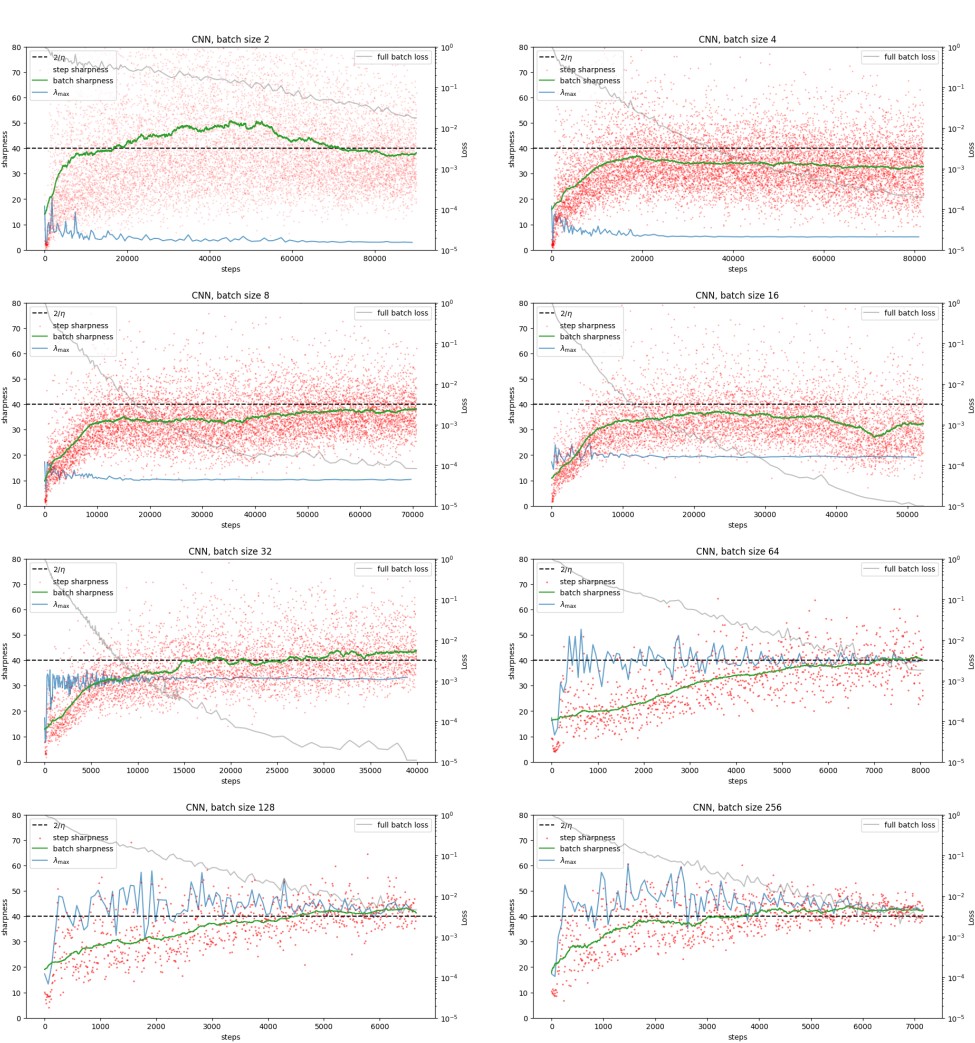

Figure 40: **CNN**: 5 layers (3 convolutional, 2 fully-connected), **step size 0.05**, 8k subset of **SVHN**. Comparison between: step sharpness, aka batch sharpness without expectation over batches and measured on the current batch (red dots, time-smoothing would be ≈ *Batch Sharpness*), the empirical *Batch Sharpness* (green line), the $\lambda_{\max}$ (blue line).

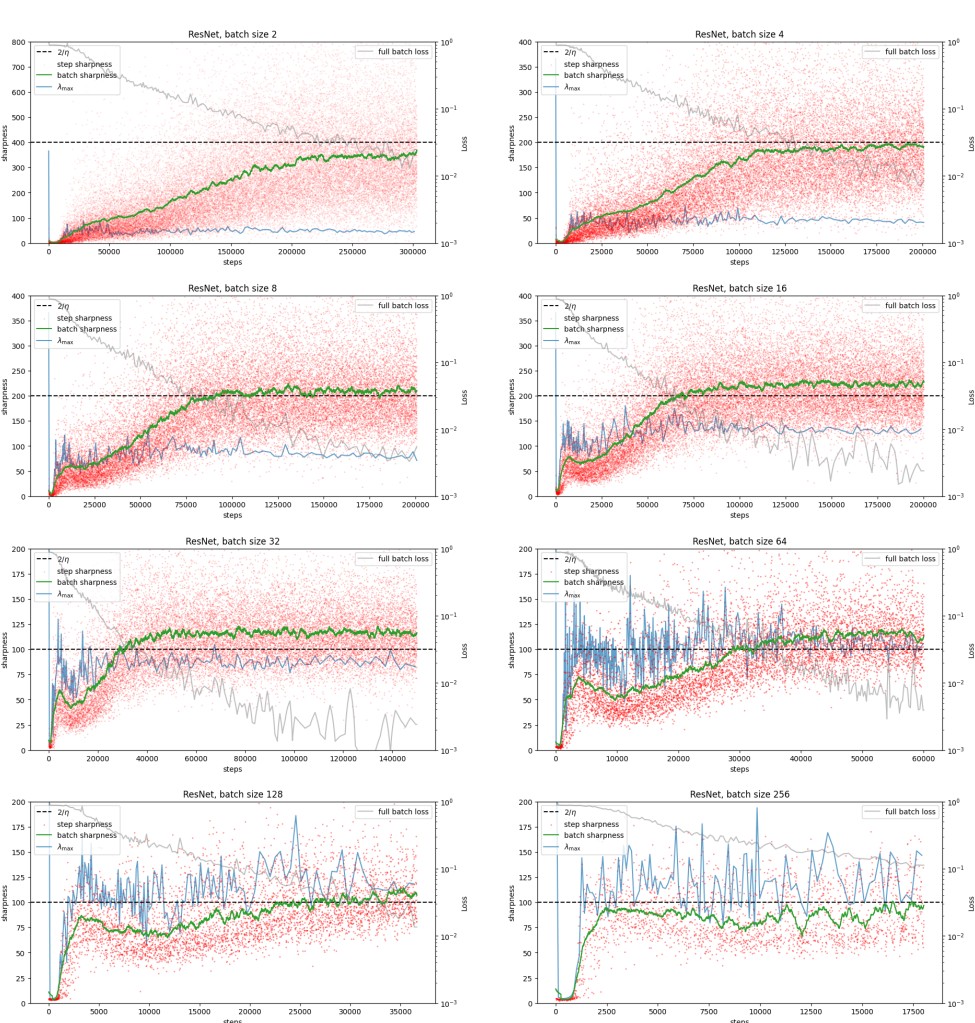

Figure 41: **ResNet-10**, step size 0.005, 8k subset of **SVHN**. Comparison between: step sharpness, aka batch sharpness without expectation over batches and measured on the current batch (red dots, time-smoothing would be $\approx$ *Batch Sharpness*), the empirical *Batch Sharpness* (green line), the $\lambda_{\max}$ (blue line).

## S    ILLUSTRATION OF EoSS IN VARIETY OF SETTINGS: $\lambda_{\max}^b$

In this appendix, we provide additional empirical evidence for both emergence of EoSS and to Appendix M.2 add!!!, varying across models, step sizes, and batch sizes. Consistent with our primary findings, we observe that $\lambda_{\max}^b$ consistently stabilizes within the interval $\left(2/\eta,\ 2 \times 2/\eta\right]$, in particular always higher than *Batch Sharpness* and $\lambda_{\max}$. We are conducting experiments on MLP, CNN and ResNet-20 in Figures 42, 43, 44 respectively.

Note that the fact that $\lambda_{\max}^b$ consistently stabilizes above $2/\eta$ implies that the supremum of Lipchitz constants of the gradient of individual mini-batch losses also stabilizes above $2/\eta$, thus clearly indicating that the usual assumptions in theory works on SGD about step size break. Same applies for supremum of Lip constants of gradients of per-sample losses. The former fact trivially follows from the inequality between sup and mean, and the second one from the same plus Lemma 6.

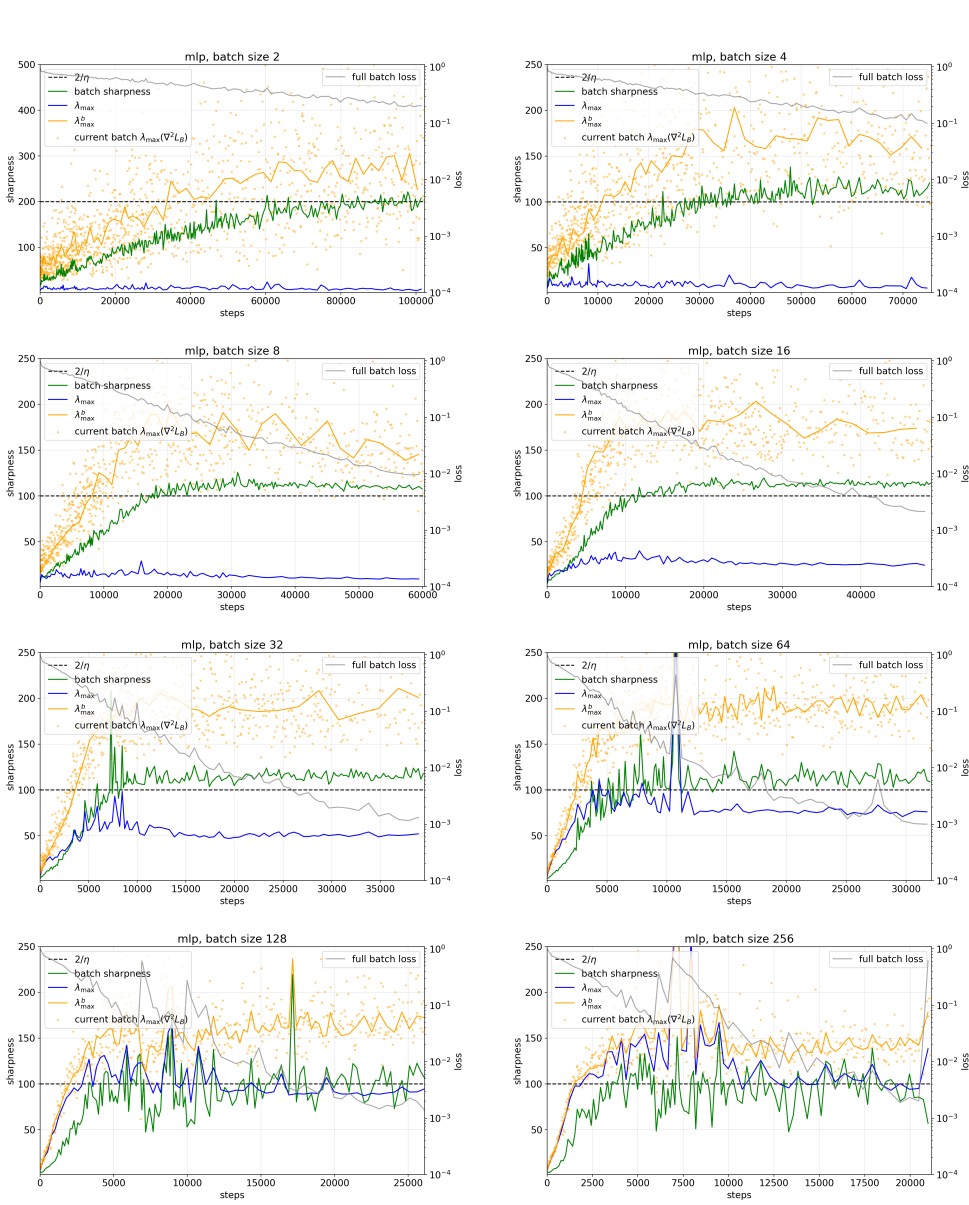

Figure 42: **Tracking $\lambda_{\max}^b$, MLP.** MLP with 2 hidden layers of width 512, step size 0.02, trained on an 8k subset of CIFAR-10. Comparison between the highest eigenvalue of the Hessian of the current mini-batch loss (orange dots, time-smoothed $\approx \lambda_{\max}^b$), *Batch Sharpness* (green line), $\lambda_{\max}$ (blue line), and $\lambda_{\max}^b$ (orange line). Note that *Batch Sharpness* stabilizes as $2/\eta$, while $\lambda_{\max}^b$ is above it, and $\lambda_{\max}$ is below for small batch sizes. Note that for batch size 2 we were using a lower step size of 0.01, as otherwise the network wasn't converging.

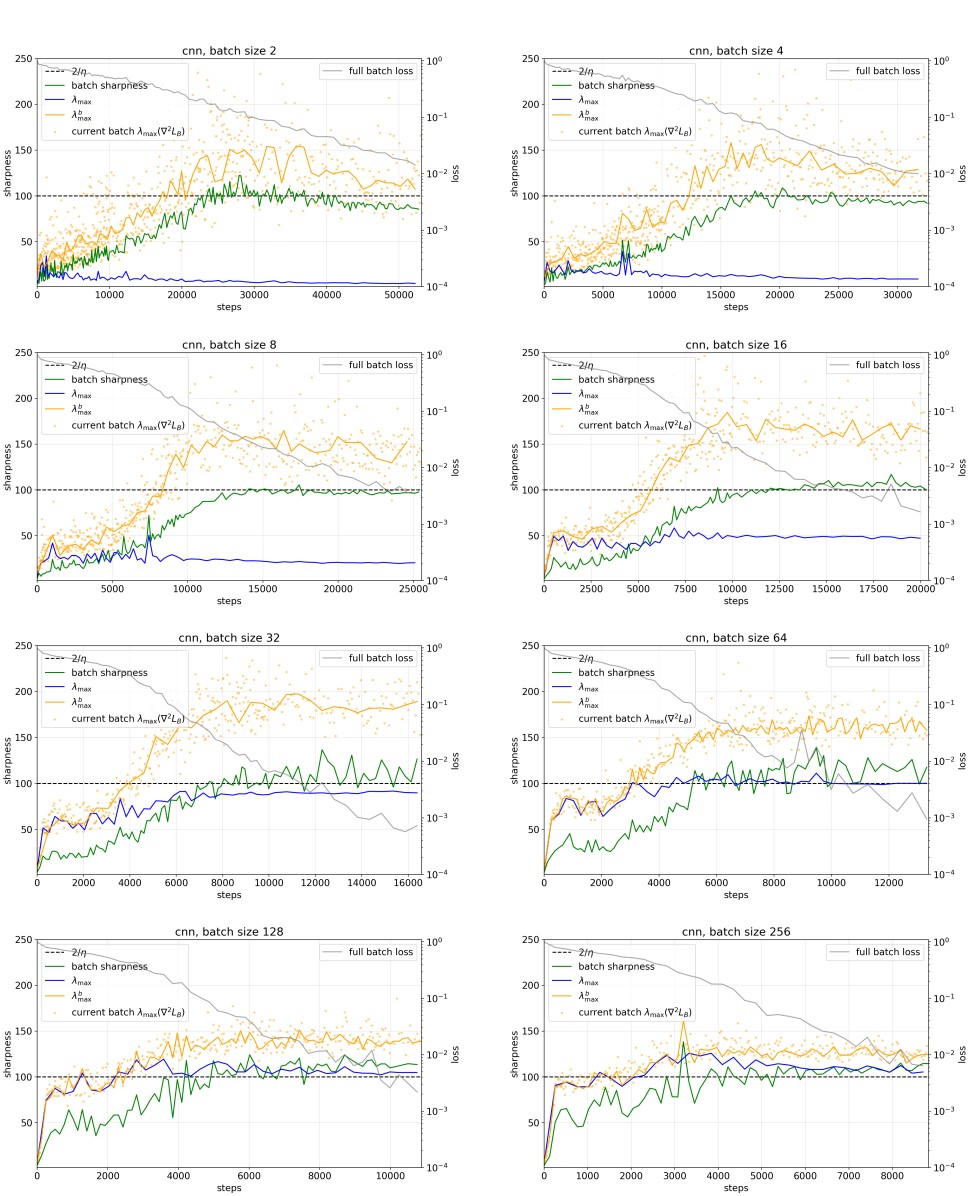

Figure 43: **Tracking $\lambda^b_{\max}$, CNN.** CNN with 5 layers (3 convolutional, 2 fully connected), step size 0.03, trained on an 8k subset of CIFAR-10. Comparison between the highest eigenvalue of the Hessian of the current mini-batch loss (orange dots, time-smoothed $\approx \lambda^b_{\max}$), *Batch Sharpness* (green line), $\lambda_{\max}$ (blue line), and $\lambda^b_{\max}$ (orange line). Note that *Batch Sharpness* stabilizes as $2/\eta$, while $\lambda^b_{\max}$ is above it, and $\lambda_{\max}$ is below for small batch sizes.

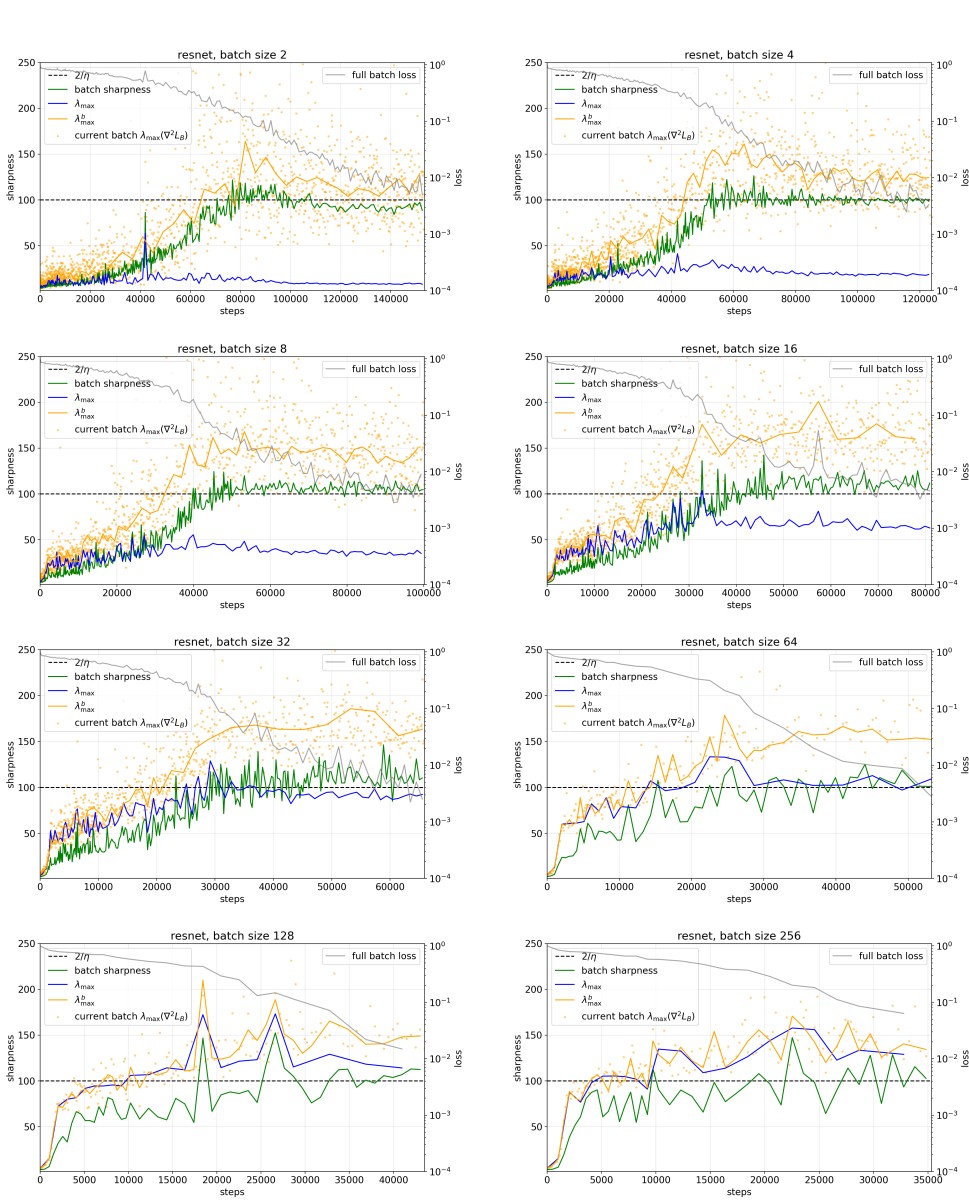

Figure 44: **Tracking $\lambda_{\max}^b$, ResNet-20.** ResNet-20 on CIFAR-10 with step size 0.02, trained on an 8k subset of the dataset. Comparison between the highest eigenvalue of the Hessian of the current mini-batch loss (orange dots, time-smoothed $\approx \lambda_{\max}^b$), *Batch Sharpness* (green line), $\lambda_{\max}$ (blue line), and $\lambda_{\max}^b$ (orange line). Note that *Batch Sharpness* stabilizes as $2/\eta$, while $\lambda_{\max}^b$ is above it, and $\lambda_{\max}$ is below for small batch sizes.

