# OpenReview forum: "Edge of Stochastic Stability: Revisiting the Edge of Stability for SGD"
_ICLR.cc/2026/Conference — Submitted to ICLR 2026_

### Official Review · Reviewer_d8mW · 2025-10-28

**Soundness:** 1
**Presentation:** 2
**Contribution:** 2
**Rating:** 2
**Confidence:** 5

**Summary:**

In the past, it has been shown that gradient descent (GD) exhibits a phenomenon known as the edge of stability. This refers to the empirical observations that the iterates of GD typically experience a progressive sharpening effect, where the sharpness (maximal eigenvalue of the Hessian) increases until it hits 2/step size, which is the stability threshold of GD, and then it hovers just above it with some oscillations until the end. This paper aims to generalize the edge of stability phenomenon to the stochastic setting (SGD), where it finds a new notion of sharpness that exhibits similar properties to the deterministic setting. They verify this with experiments.

**Strengths:**

The main strength of this paper lies in the following insight. Prior work [R1] has established that stable training for SGD with a constant step size $ \\eta $ occurs when the Gradient Noise Interaction (GNI) satisfies $ \\text{GNI} = \\frac{2}{\\eta} $. This paper introduces a new observation, arguing that GNI may not be the appropriate measure of stability. Specifically, whenever $ \\theta\_t $ converges in probability (*i.e.*, converges to a stationary distribution), for sufficiently large $ t $ we have
$ \\mathbb{E}[L(\\theta\_t)] \\approx \\mathbb{E}[L(\\theta\_{t+1})] = \\mathbb{E}[\\mathbb{E}[L(\\theta\_{t+1}) \\mid \\theta\_t]]. $
In other words, the stability condition implied by the GNI definition holds in expectation, so that, on average, $ \\text{GNI} \\approx \\frac{2}{\\eta} $. However, this relationship is independent of curvature, which is central to the deterministic EoS.

Additionally, I think that classifying the origin of instabilities (gradient noise or curvature) in SGD is interesting.

**References:**\
[R1] - Sungyoon Lee and Cheongjae Jang. A new characterization of the edge of stability based on a sharpness measure aware of batch gradient distribution.

**Weaknesses:**

**Main issues:**
* The paper claims to generalize the edge-of-stability (EoS) phenomenon observed in gradient descent (GD), *i.e.* $ \\lambda\_{\\max} \approx \\frac{2}{\\eta}  $, to the stochastic setting of SGD. However, this claim is incorrect. A straightforward way to see this is by considering the full-batch case: under this setting, the proposed measure fails to recover the deterministic EoS condition. This discrepancy stems from the fact that batch sharpness is derived from the descent lemma on a random batch loss, which characterizes a single gradient step. In contrast, the deterministic EoS condition in GD emerges from multi-step dynamics, where the iterates align with the sharpest curvature direction near the minimum. Such long-term dynamical behavior cannot be captured by the descent lemma alone. Therefore, the purported generalization of EoS to the stochastic case is not merely an overstatement—it is a false claim.

* The paper does not provide a clear explanation or justification for why the proposed batch sharpness measure is of practical or theoretical interest.
The only result presented in the main paper regarding this measure is Theorem 1. According to Theorem 1 if the batch sharpness happens to be too large, then consecutive steps of SGD using the same batch lead, on average, to a rapid increase in the gradient of the loss over that batch. However, in standard SGD, the probability of sampling the same batch in consecutive iterations is effectively zero. This raises the question: why should this result matter in practice? The natural assumption is that the batches across iterations are independent. Furthermore, the theorem does not specify the exact condition—it is stated only up to an unspecified absolute constant, which could potentially be large. This makes the claim that batch sharpness $ \\approx \\frac{2}{\\eta} $ rather peculiar. Finally, it is hard for me to see how the batch sharpness can govern the stability of SGD.

* This paper considers a quadratic approximation of the loss (see Def. 1), namely, linearized dynamics of SGD. This **precise** setting was already studied in the past, and the exact stability condition in a closed-form expression was given in [R2]. Relating to the first point here, in contrast to this paper, [R2] gives a multi-step analysis. Importantly, the stability threshold in [R2] does recover the stability condition in the deterministic case, *i.e.*, it generalizes the stability threshold of GD to SGD (while the approach here fails). Moreover, many calculations in the appendix overlap with [R2]. For example, the evolution of the covariance matrix in App. C and D. Therefore, it seems that the novelty of the current manuscript is limited, and the presented results are weaker than those of published literature. Moreover, the use of the descent lemma to characterize stability in SGD was done by prior work [R1], and the proposed measure of batch sharpness is very similar to the terms in [R1]. This fact further diminishes the novelty of this paper.

**Presentation and writing issues:**
The paper lacks clarity and rigor. Here are a few examples that illustrate these issues:
* In Definition 3, we have the following expression
$$ \\text{Batch Sharpness}:= \\mathbb{E} \\left[ \\frac{( \\nabla L\_B)^T \\nabla^2 L\_B \nabla L\_B  }{\\| \nabla L_B \\|  } \\right] $$
However, in the appendix line 1867, also in equation (49), and in the definitions and the derivations in App. F where $ \\text{Batch Sharpness}:= \\frac{\\mathcal{A}}{\\mathcal{C}} $ we have
$$ \\text{Batch Sharpness}:= \\frac{\\mathbb{E} [ ( \\nabla L\_B)^T \\nabla^2 L\_B \nabla L\_B ] }{\\mathbb{E} [ \\| \nabla L_B \\| ] }$$
These two expressions are different. I tend to believe that the latter is the correct expression.
* Although the stability measures (GNI and batch sharpness) have clear mathematical formulas, no mathematical definitions are given to Type 1 and Type 2 oscillations.
* Definition 1 is unclear (maybe a fault?)


**Reference**:\
[R2] - Rotem Mulayoff, Tomer Michaeli. Exact Mean Square Linear Stability Analysis for SGD

**Questions:**

1)  [R1] used a notion of stability defined as $\\mathbb{E} [L(\\theta\_{t+1}) | \\theta\_t ] \leq L(\\theta\_t) $. This translates to a stability condition of (if and only if)
$$ \\text{GNI}: = \\frac{\\mathbb{E} [ ( \\nabla L_B)^T \\nabla^2 L \nabla L_B ] }{\mathbb{E} [ \\| \\nabla L \\| ]  } \leq \frac{2}{\eta},$$
where the edge of stability is obtained in equality. In the paper, you showed that this notion of stability (condition or definition, they are equivalent) is misleading and fails to capture the right characteristics of EoS.\
On the other hand, your underlying definition of stability is $ \\mathbb{E} [L\_B(\\theta\_{t+1}) | \\theta\_t ] \\leq \\mathbb{E} [L_B(\\theta\_t) | \\theta\_t ] $, and the corresponding stability condition is (if and only if)
$$ \\text{Batch Sharpness}:= \\frac{\\mathbb{E} [ ( \\nabla L\_B)^T \\nabla^2 L\_B \nabla L\_B ] }{\\mathbb{E} [ \\| \nabla L_B \\| ] } \\leq \\frac{2}{\\eta}.$$
My question is the following. Why does your underlying definition of stability make sense? Please base your answer on the definition.

2) Why do you claim that batch sharpness governs the stability of SGD?

---

> ### Author Response · Authors · 2025-11-27
>
> First, a moment to thank you for the fact that you appreciated our effort to classify oscillations and distinguish instabilities from stable expected oscillatory behaviors. We put great efforts in this conceptual part, as we believe that one can not really speak of EoS in mini-batch settings without disentangling different phenomena that simplify to the same one in the deterministic case.
>
> Thank you also for the long line of comments and for taking the time to properly read our article. We will deep dive now into trying to address your comments one by one.
>
> All the line number reference are on the old version.
>
> ---
>
> ## Weakness 1
>
> > Under this setting, the proposed measure fails to recover the deterministic EoS condition
>
> You are right that in the full-batch case \(B = N\), batch sharpness does not reduce to \(\lambda_{\max}\); instead, it becomes the Rayleigh quotient of the Hessian in the direction of the full-batch gradient. We already comment on this in the original submission in “Relation to earlier notion” (L257–271, though the wording there was confusing, it is now rephrased in the new version in Section 5.2) and explicitly at L995 in Appendix B.2. However, as you mention in your question, this is not the point, what we need is that EoSS reduces to EoS, not that Batch Sharpness reduces to $\lambda_{\max}$.
>
> That is, we explicitly now state *in what sense* EoSS generalizes EoS, which is probably the more rigorous and correct way to put it. Indeed, “Batch Sharpness generalizes \(\lambda_{\max}\)” (L263) in the sense that it causes divergence if the corresponding quantity crosses $2/\eta$, and not directly through the fact that when $B=N$, we have the same quantity. I.e., Batch Sharpness generalizes $\lambda_{\max}$ in the sense that it is the quantity governing EoSS which generalizes EoS.
> In Appendix B we properly formalize what an unstable behavior is now, and at the end of B.1 we explicitly say how this relates to the deterministic case.
>
> Given the important comment above, in a sense one can also relate Batch Sharpness to $\lambda_{\max}$ directly.
> **(1)** In the full-batch regime, the Rayleigh quotient we obtain also a quantity that reaches (2/\eta) at the edge of stability (and is at $2/\eta$ iff $\lambda_{\max}$ is), which is theoretically expected and empirically observed by Lee and Jang (2023); this is why we refer to their result in L258. As we now, thanks to your comment, mention in Section 5.2, in the full-batch setting, $\lambda_{\max} > Batch\ Sharpness$, thus if Batch Sharpness is at 2/\eta, also $\lambda_{\max}$ is. Viceversa, if $\lambda_{\max}$ crosses $2/\eta$ the gradient aligns with the top eigenvectors, thus a few steps later, also the Rayleigh quotient Batch Sharpness hovers at $2/\eta$.
> **(2)** This alignment perspective helps us connect to an other point (also now made in Section 5.2): that thus in EoS $\lambda_{\max}$ can be seen as a (time-)averaged direction curvature in the step direction. Importantly---and we added Appendix K to show that empirically---the gradients on the mini-batches are never properly aligned, thus it is natural to consider the same quantity, the averaged directional curvature, which is now Batch Sharpness. This is exactly the point of view from which Batch Sharpness "converges" to $\lambda_{\max}$ when batch size is increased.
>
> We hope this, the new Appendix B, and the new Section 5.2 completely address this point you raised.

---

> ### Author Response · Authors · 2025-11-27
>
> ## Weakness 2 - The Descent Lemma & Proof of Theorem 1
>
> ### Concerning your comment about same batch on Theorem 1.
>
> We believe your weakness 2 was due to a **misunderstanding**---we are referring to the submission version you reviewed here.
> > SGD using the same batch lead, on average, to a rapid increase in the gradient of the loss over that batch
>
> This is *not* what we argue, in Theorem 1, you see an average over the batches. In the proof you see that we use **(1)** an index $i$ for the batch we compute the norm of the gradients, **(2)** an *independently sampled* index $j_t$ for the batch used by SGD at step $t$, and **(3)** we take the expectation over sampling $i$ and $j_t$.
> We explicitly state in L1757 of the old version that the index over which we take the expectation, $i$, is independent of the batch index $j_t$  used to perform the SGD step.
>
> Thus all the arguments holds for:
>  - Mini-batch SGD where the batch are step $t$ are either sample independently or without replacement. In particular, we do not use the same batch at every step, we use batches sampled with whatever batch sampling procedure your algorithm uses.
>  - The rapid gradient increase is, as we state in Theorem 1, on $\mathbb E_{B \sim \mathcal P_b} [ \| \nabla L_B \|^2]$ thus on the expectation over our batch sampling procedure of the norm squared of the mini-batch gradients.
>
> > The natural assumption is that the batches across iterations are independent.
>
> Indeed, our theorem holds *both* under that, and *also* mini-batch without replacement assumption. As explained.
>
> Therefore, we are not assuming that SGD repeatedly picks the same batch, and the interpretation that our result relies on consecutive reuse of the same batch is incorrect.
>
> ### Discrepancy and Descent Lemma
>
> > This discrepancy stems from the fact that batch sharpness is derived from the descent lemma on a random batch loss, which characterizes a single gradient step.
>
> We assume you are referring here to Eq. (4). However, Eq. (4) is only meant as intuition/interpretation for the definition of Batch Sharpness. We never use Eq. (4) in any of our instability proofs, nor do we equate that one-step condition with EoSS. In fact, the proof of Theorem 1 is explicitly not based on the descent lemma.
>
> In particular, our use of the descent lemma is precisely part of our critique of the approach of Lee and Jang (2023) :) As we mention in L229, Appendix B.2, and L1533. We intentionally do not foreground this critique in the main body, since our goal in this work is not to focus on criticizing the method of Lee and Jang (2023).
>
> ### Up to a constant
>
> For the following discussion, it is important to note that we have substantially reworked the proof of Theorem 1 in the revised manuscript, see Appendix G. While the original proof did have some issues, these are all addressed in the new version, and the revised proof no longer suffers from them.
>
> That said, let us anyways respond to this critique of our previous Theorem 1, as we believe it was not a limitation of the theorem, actually a strenght.
>
> > “Furthermore, the theorem does not specify the exact condition - it is stated only up to an unspecified absolute constant, which could potentially be large.”
>
> The general such statements of induced instability usually depends on assumptions of the form:
> \[
> \text{curvature notion} \geq \frac{2 + \epsilon}{\eta} = \frac{2}{\eta} +  \frac{\epsilon}{\eta}.
> \]
> This is also as we rewrote the statement now, however, before by stating
> \[
> \text{curvature notion} \geq \frac{2}{\eta}+c\eta
> \]
> we thought we were stating a stronger result, as in $c\eta \leq \epsilon/\eta$ for small $\eta. Now we framed it this way.
>
> ### Not providing a clear explanation
>
> > The paper does not provide a clear explanation or justification for why the proposed batch sharpness measure is of practical or theoretical interest.
>
> We respectfully disagree with this one, as the other reviewers, by the way. We do it in the implications, the intro, etc.
> E.g., the fact that it is a valid instability criterion means that classical optimization proof do not apply, that noise injected SGD does not explain mini-batch SGD, etc.
>
> We believe the reason why you made this strong statmeent is that you generalized to much the fact that Theorem 1 was not strong enough to sustain our empirical findings. On one hand we added more mathematical framework and treatment with Appendix B---and we believe this criticism does not apply anymore in general. On the other hand, we believe you meant this because of the misunderstanding on Theorem 1. We completely agree that if Theorem 1 was as you understood it was not enough, but importantly it was very different.

---

> ### Author Response · Authors · 2025-11-27
>
> ## W3 - Comparison to Other Works
>
> ### **General: Misunderstanding of what our paper does**
>
> > This paper considers a quadratic approximation of the loss (see Def. 1), namely, linearized dynamics of SGD. This precise setting was already studied in the past [...]  Therefore, it seems that the novelty of the current manuscript is limited, and the presented results are weaker than those of published literature.
>
> **We believe there is an important misunderstanding of what our paper actually is, were it stems from, what it shows**:
>
> - We completely agree that previous mathematical work was present on linear stochastic stability, it is important, necessary, and solid mathematical work which *we are not* rewriting, we are complementing! Our article is exactly **not** about that!
>
> - **Our message is:** (We hope it is now clear with the adition of new Appendix B) There are many possible notions of stability for *non-linear* *stochastic* dynamical systems, as SGD when training NN. We search empirically, for that valid notion of stability which *does describe* the training of neural networks.
>
> That could be linear stability as [R2], that could be stable oscillations as [R1], that could be similar to logistic cycles or whatever. That could be based on high-probability bounds of curvature, could be based on expectations, on any possible one dimensional statistics of the distribution of the Hessians *or* the higher order derivatives. We do find one, which we formalize, which explains empirically the behavior. Our paper is about claiming that many are possible, unlike for deterministic GD, and finding a possible one. It is not about invalidating or re-inventing linear stability!
>
> ### **Technical, Detailed, Response**
>
> We've already discussed how our work differs from [R2] in the Related Work section (**L153–157 + footnote**) and in ** old Appendix B.1** and more clearly layed out in **new Appendices B, L**.
> To better address your criticism and the relation with Linear Stability, we also added Appendix L, “Linear Stochastic Stability”, which directly compares our results to works that analyze SGD using linear stability. Even in the original manuscript, however, we already clarified the key differences. Let us summarize them here:
>
> (a) [R2]—more precisely, [R3] Ma and Ying (2021), on which [R2] is based—does derive a stability condition for a chosen Lyapunov function (the distance of the iterates from 0). However, since this is done under a linearization/approximation of the dynamics, there is no a priori definition of stability for the true SGD dynamical system itself, and without empirical evidence it is not clear that the corresponding instability regime is ever actually saturated/becomes a limitation.
>
> (b) Indeed, none of these works show that SGD on neural networks actually *trains* continuously in an \textsc{EoS}-like instability regime along its real trajectory—that is, that the instability condition they derive is in fact saturated in practice.
>
> (c) Moreover, the conditions in [R3] and [R2] involve operators on (d^2)-dimensional spaces, which makes them effectively incomputable for modern neural networks—and this intractability is exactly what underlies point (b). In practice, [R2] therefore works with lower bounds on this quantity, but even in their own experiments these bounds do not consistently saturate at the minima (let alone along the training trajectory).
>
>
> Regarding your comment:
>
> > “Moreover, many calculations in the appendix overlap with [R2]. For example, the evolution of the covariance matrix in App. C and D.”
>
> these calculations concern GNI and Proposition 1, which we do *not* constitute a novel contribution of our work. Their role in the paper is purely to illustrate the nature of Type-1 oscillations. We never claim novelty there. So criticizing these parts specifically for lack of novelty feels somewhat misplaced, since they are only meant as supporting analysis. Furthermore, the techniques we use for those derivations are indeed classical and similar to those in [R2], but only in the same way that many standard optimization proofs share common tools and arguments.. The theoretical novelty of our work lies in the treatment of Batch Sharpness, not in the GNI calculations.
>
> Lastly, regarding
>
> > “Moreover, the use of the descent lemma to characterize stability in SGD was done by prior work [R1], and the proposed measure of batch sharpness is very similar to the terms in [R1].”
>
> as we explained above, we do not use the descent lemma to characterize stability in SGD. Its role in our paper is solely to provide intuition/interpretation for the definition* of Batch Sharpness, not to derive or justify our stability results.

---

> ### Author Response · Authors · 2025-11-27
>
> ## Definition of Batch Sharpness
>
> Concerning the definition of Batch Sharpness, the correct definition is the one we have in Definition 3:
> $$
> \textit{Batch Sharpness}(\theta)\quad:=\quad
> \mathbb{E}
> \bigg[\frac{
> \nabla L_B(\theta)^\top\,\mathcal{H}(L*\mathbf{B})\,\nabla L_B(\theta)}{\|\nabla L_\mathbf{B}(\theta)\|^2}
> \bigg]
> $$
>
> This is the quantity we measure in our experiments, and the quantity that is in the updated proof in Appendix G is exactly about this quantity. Our original proof was about the related quantity, which we term “modified Batch Sharpness”. Indeed, it is a quantity with the expectation “inside the fraction”:
>
> $$
> \text{Modified \textit{Batch Sharpness}}(\theta)
> \quad := \quad
> \frac{
> \mathbb E\Big[
> \nabla L_B(\theta)^\top\,\mathcal{H}(L_B)\,\nabla L_B(\theta)\Big]
> }{
> \mathbb E\Big[
> \|\nabla L_B(\theta)\|^2
> \Big ]
> }.
> $$
>
> Our original proof showed that this is also a valid instability criterion for a specific Lyapunov function, and relating the two quantities. We moved this proof for modified batch sharpness into a separate Appendix N and proved Theo 1 directly in the new proof. Yet, as we show in the Appendix M.3, its stabilization level is inconsistent and is above $2/\eta$.
>
>
> ---
>
> > Definition 1 is unclear (maybe a fault?)
>
> We unfortunately don't understand the exact meaning of "maybe a fault", what do you mean by unclear?
> We added Appendix B and H to make that discussion rigorous, we hope now it's clear.
>
>
> ----
> ## Questions
>
> > Why does your underlying definition of stability make sense? Please base your answer on the definition.
>
> We hope the discussion above already addresses this, but let us briefly summarize. First, as discussed, the correct definition of Batch Sharpness is the one with the expectation “outside.” The whole paper is essentially about why this definition makes sense, but to give a compact answer: intuitively, Batch Sharpness captures the average stability properties over mini-batch landscapes. Formally, it yields a valid instability criterion (Batch Sharpness (< 2/\eta)), and empirically we observe that it saturates during training. Taken together, these two points also address the related question “Why do you claim that batch sharpness governs the stability of SGD?": That is the whole point of the paper, it is an instability criterion, and that such one that we can compute that stabilizes at 2/\eta.
>
> ---
>
> ## Concluding
>
> To conclude, thanks again for appreciating our conceptualization of oscillations and instability in SGD, and a warm thank you again for the detailed and thoughtful feedback. Your feedback and the misconceptions on instability helped us identify places where our presentation and proofs were confusing or incomplete, and we have substantially revised the manuscript in response: in particular, we significantly reworked the proof of Theorem 1, prompted by you, we hope that now the connections and the conceptual points are more clear and less inclined to misconceptions.
>
> Prompted also by you, we further improved this part with rigor adding Appendix B, which together with Section 3, we believe now completely frames what unstable oscillations are, rigorously. On one hand we hope this could be basis for increase in the score, but independently, on the other hand, we hope you appreciate it and you may give a look and tell us what you think and if it properly addresses the general framework of instability for optimizers.
>
> Thanks again, and looking forward to the rest of our discussion.
>
> The authors

---

### Official Review · Reviewer_YyPY · 2025-10-29

**Soundness:** 3
**Presentation:** 2
**Contribution:** 3
**Rating:** 4
**Confidence:** 3

**Summary:**

This paper introduces the regime that is coined the edge of stochastic stability for the mini-batch SGD.  In this regime, what stabilizes at $2/\eta$ is batch sharpness, defined as the expected directional curvature of mini-batch Hessians along their corresponding stochastic gradients. This leads to the observation that is in line with the well-known fact that smaller batches and larger stepsizes favor flatter minima.

**Strengths:**

The paper contains many fine details and extensive discussions on many different aspects, as well as the literature background. The research problem that is proposed in the paper is an interesting and important topic worth investigating.

**Weaknesses:**

(1) The presentation of the paper needs to be improved. I find it not that easy to follow. The Appendix is super long, and contains many random topics that do not seem to capture the essence of the major contributions of the paper.

(2) When I read the proofs in the Appendix, there are too many places you used $\approx$ which should be made more rigorous by using Big O notation or other notations that can be made rigorous or at least you should make the meaning of $\approx$ more transparent and explicit.

**Questions:**

(1) In terms of literature review, since previous works on SGD stability is most relevant. I actually did not see you have more discussions until I see Appendix B. Perhaps you should mention in the main paper that more details will be provided in Appendix B.

(2) The statement of Lemma 1 is not very rigorous. You should specify the meaning of $\approx$ or simply avoid using $\approx$.

(3) I would suggest you to add footnote 7 at the end of the sentence as the ratio, instead of inside equation (3).

(4) On page 22, you discussed by Taylor expansion $\nabla L(x)\approx\mathcal{H}(x-x^{\ast})$, but if you use an approximation here, it is not clear to me why you get an equality in $\mathbb{E}_{k}[\Vert\nabla L(x_{k})\Vert^{2}]=\text{Tr}(\mathcal{H}\Sigma_{x}\mathcal{H})$.

(5) On page 25, you should make the meaning of $\approx$ in $\Sigma_{x}\approx\frac{\eta}{b}\mathcal{K}^{\dagger}(\Sigma_{g})$
more transparent or simply avoid using $\approx$ notation.

(6) In the statement of Theorem 1, a absolute should be an absolute.

(7) On page 27, in the last equation, you had an extra $)$ which is a typo.

(8) On page 33, Remind that should be Recall that.

---

> ### Author Response · Authors · 2025-11-27
>
> Thank you for the comments! We're glad you appreciate our efforts in commenting the fine details and addressing this relevant open puzzle of optimization for deep learning.
>
> ## On Weakness 1
>
> ### Flow and presentation issues
> > The presentation of the paper needs to be improved. I find it not that easy to follow.
>
> We apologize for the lack of clarity in the original version and have revised the paper to improve readability: incorporating several changes (including those suggested by reviewer mnHT), tightening Section 3, moving some of its material to Section 5, and expanding Section 5 accordingly. We also added an Appendix B to formalize the arguments that seemed sloppier.
>
> ### Appendix Length
> > The Appendix is super long...
>
> We agree that the appendix in the original version was overly long. Our initial idea was to also cover adjacent topics that might be useful for readers wishing to extend our results, but in light of your comment we have streamlined some of those parts (although we added the new B and and new H to formalize catapults). In particular, we removed Appendices O (“Exemplification Through a Simplified Models”), H ("Pure Gradient-Noise Oscillations"), which were relevant but not strictly necessary to understand the main results. We also added explanations at the beginning of the appendices to clarify what is done there. Finally, we consolidated and significantly shortened some parts: the relevant material from “K – On Largest Eigenvalues of Sums of Matrices” and “L – Dependence of $\lambda_{\max}^b$–$\lambda_{\max}$ Gap on the Batch Size” has been merged into the new Appendix M. Linear Stochastic Stability was moved into a dedicated Appendix L. We moved Appendix I ("Mini-Batch Without Replacement") as a part of Appendix G. We hope these changes make the appendix more accessible, while still aligning with our goal of covering adjacent, relevant topics.
>
> ---
>
> ## Weakness 2
>
> > When I read the proofs in the Appendix, there are too many places you used $\approx$...
>
> Agreed, this was an issue. We address the concrete examples you brought up below. We have substantially reworked the proof of Theorem 1 to avoid such uses (see Appendix G).
> Importantly, whenever we used $\approx$ we meant "up to higher order terms in $\eta$ which would anyways disappear in the error term". We made that precise.
>
> ---
>
> # Questions
>
> ## Question 1 - Literature Review: Previous Works on SGD Stability
> Regarding your question about the discussion of “previous works on SGD stability” in the related work: we did already point to Appendix B in a footnote in that paragraph, but we agree this was not prominent enough. In the revision, we moved this reference into the main text for better visibility, rewrote that paragraph, and moved and expanded the discussion of linear stochastic stability into a separate Appendix L. We also added a general discussion to contestualize instability for stochastic dynamical systems in the new Appendix B, which indirectly connects our results to other works on SGD stability and we reference now multiple times in the main body.
>
> ## Question 2
>
> > The statement of Lemma 1 is not very rigorous....
>
> This was actually deliberate in the main text, we agree though. For matter of space we do not define what oscillations rigorously are (the Loss update changes sign) and thus we do not formalize properly Lemma 1 there. It was a choice, but happy (now that we have also one more page) to be formal. In the meantime, to address your concern, we added in Appendix D.2 the formal version (Proposition 2) of Lemma 1 and properly referenced there. We hope that this is enough to address your issues with Lemma 1 and sotry of *Type-1* oscillations, let us know if we can do more in this direction.
>
>
> Thanks again for the questions and suggestions!

---

> > ### Author Response · Authors · 2025-11-27
> >
> > # Questions (continued)
> >
> > ## Other comments
> > Please note that all the references we give here to lines' numbers are for the old version.
> >
> > * “I would suggest you to add footnote 7” - thanks; changed!
> > * “On page 22, you discussed...” - the argumentation on page 22 is just the intuition for the proof of Proposition 1 (as stated on L1130), with the proof itself in the Appendix D, so the statements there are deliberately non-rigorous. The rigorous statements are in appendix D. Still, we will add the “up to higher orders in the deviation” there per your request.
> > * “On page 25, you should make…” - we wrote it in such way for readability sake. On L1307 we had the precise statement that $\Sigma_x = \tfrac{\eta}{b}\,\mathcal{K}^\dagger(\Sigma_g) + \mathcal{O}(\eta^2)$. Now, when we use $\Sigma_x\approx \tfrac{\eta}{b}\,\mathcal{K}^\dagger(\Sigma_g)$, it is therefore clear that this means “up to \mathcal{O}(\eta^2)”, which we do for ease of reading; the reason we are justified to drop that term is that our final result is up to $\mathcal{O}(\eta)$.
> > * “(6) In the statement of Theorem 1...” - thanks, fixed!
> > * “On page 27, in the ...” - thanks, fixed
> > * “On page 33, Remind...” - thank you, fixed! Although that proof is fully moved to a different Appendix N.
> >
> > # Concluding
> > We agree with you that we could have done a better job in the presentation. We appreciate, though, that you found we provide extensive detailed explanations and the topic is worth investigating.
> > Please do let us know if you think this completely addresses your concern and whether there is something else we can do for you to raise your score.
> >
> > Thanks again for your review.
> >
> > The authors

---

### Official Review · Reviewer_mnHT · 2025-10-31

**Soundness:** 3
**Presentation:** 2
**Contribution:** 3
**Rating:** 6
**Confidence:** 3

**Summary:**

The authors discover a counterpart of the edge-of-stability (EoS) phenomenon by Cohen et al. 2021 in the case of training under SGD, which they call the edge of stochastic stability (EoSS). The authors' augment the curvature-based oscillations of EoS with (gradient) noise-driven oscillations. In this context they name these type-1 (noise-driven) oscillations and type-2 (curvature-driven) oscillations. According to the authors, only the latter is what drives EoS-like behavior under SGD: batch sharpness (BS) stabilizes at $\approx2/\eta$, and sudden increases in BS lead to catapult-like behavior that puts back the iterates in a different basin with BS $\approx2/\eta$. Decreasing the batch size (through increasing BS) or increasing the learning rate (through decreasing $2/\eta$) lead to similar catapult effects, and make the iterates settle in a sharper basin.

**Strengths:**

- The paper is very well motivated. Its contributions are solid and important. The breakdown of the oscillation in the SGD case is simple, elegant, and very useful. The comparisons with previous work seem sufficient.
- In addition to their main findings, the authors' conclusions re. SGD vs. noisy gradient descent is very useful, and speaks to the potential of their findings to progress the field.

**Weaknesses:**

- Although the paper is about extending EoS to SGD, and involves a lot of comparisons between the two, the authors do not have an authoritative explanation why SGD does not follow EoS. That is, their results show why SGD follows EoSS, but it does not show why/when following EoSS corresponds to not following EoS - in the form of a more specific relationship between batch sharpness and $\lambda_{\max}$. In this sense the paper parallels Cohen et al. 2021 but does not complement it. Experiments at Section 4.1 and Appendix J are attempts at this, but I believe a more explicit discussion of this in the main paper is warranted.
- The paper is sometimes very difficult to read. This is due to a range of issues from conceptual conflicts to more mundane errors. I will provide a main example here and defer the rest to the following section. In Section 4, part 3. Catapults, the authors highlight the batch stochasticity-based spikes in batch sharpness to account for the "catapults". However, their original understanding of catapults seem to include the sharpness-recovering behavior in both EoS and EoSS.

**Questions:**

- There seems to be a problem with the style file, and the fonts are not compiled as intended
- Alphabetical ordering of citations needed
- Footnotes (3, 4, 7) are made following right after equations, disrupting the reading of the formulas.
- L014: Cohen et al. (2021) consistently states that sharpness stabilizes *above* this value.
- L015: Cohen et al. (2021) explicitly ignores generalization concerns.
- L039: Items (2) and (3) are unclear.
- Boxed summary:
	- L046: I think at least a brief mention of the notation should be made before the batch sharpness equation is presented.
	- L046: Similar to? The boxed summary should be mostly self-contained
- L052: "implicitly functions as sharpness", "stability for SGD is stability on the mini-batch landscape" sound confusing. I would encourage the authors to slightly expand on these, as their vagueness defeats the purpose of this highlight.
- L063: $\eta>0$?
- L077: What is $\tilde{L}$? Please define.
- L093: First couple of sentences make me think that another paragraph with the title e.g. "Learning rate and gradient-based optimization" can be used.  As Jastrzebski et al. 2020 is about SGD, for example.
- L115: How is this paragraph title different than L091?
- L121: I feel this lengthy quote from Cohen et al. is unnecessary.
- L130: "the answer" to?
- L186: What does "step size does not vanish" mean? Please explain within the text.
- L191: "leaves all the compacts in which..." -> "exits all compact subset of the region in which..."
- L225: Why not $\mathcal{H}(L)$ to be consistent with $\mathcal{H}(L_B)$?
- L229: "descent lemmas" are unintroduced.
- L266: I do not understand this footnote.
- L270: "Importantly, a reason why..." I do not understand this sentence.
- L302: What does step sharpness mean?
- L313: Referring to a future figure for notation seems suboptimal
- L324: Please increase the axis and legend fonts for figures dramatically
- Minimal typos/errors:
	- L039: Seemingly mistaken use of $L$ instead of $\lambda_{\mathrm{max}}$.
	- L044: "as $\lambda_{\mathrm{max}}$" -> "such as $\lambda_{\mathrm{max}}$"
	- L176: "constraints" -> "constrains"
	- L181: The word "oscillations" neglected from the title?
	- L215: "Dyanmics"
	- L285: "traind"

---

> ### Author Response · Authors · 2025-11-27
>
> We want to deeply thank you for carefully reading the manuscript and apologize for the difficulty in parsing parts of it!
>
> ---
>
> ## Response to W1
>
> > Although the paper is about extending EoS to SGD, and involves a lot of comparisons between the two, the authors do not have an authoritative explanation why SGD does not follow EoS. That is, their results show why SGD follows EoSS, but it does not show why/when following EoSS corresponds to not following EoS - in the form of a more specific relationship between batch sharpness and $\lambda_{\max}$.
>
> Thanks a lot for raising this point — we have added Subsection 5.2 and Appendix K to clarify it. Please let us know if you feel this addresses your question sufficiently.
>
> On a quadratic (deterministic) landscape, when $\lambda_{\max} \geq 2/\eta$, the gradients align exponentially fast with the eigenvectors corresponding to $\lambda_{\max}$. In this sense, $\lambda_{\max}$ can be viewed as a notion of curvature along the steps—essentially, a time-average of curvature along the gradient directions.
>
> In the stochastic case (we added Appendix K for this purpose), the gradients on the mini-batch landscapes remain misaligned. Batch Sharpness captures the same idea of averaged curvature along the steps, but now the steps themselves are misaligned, and this needs to be taken into account.
> In this way, our work parallels and complements Cohen et al. (2021), and makes more precise which property of their quantity is actually relevant.
>
> Regarding the gap between these two quantities, we attempt an empirical comparison in Appendix J. Batch Sharpness inherently depends on the second and third cumulants of the distribution of per-datapoint Hessians.
>
> For small batch sizes, $\lambda_{\max}$ is smaller because these cumulants are significant; they vanish as the batch size increases. However, we cannot compute the exact value of the gap between the two, as it depends on the dynamics of progressive sharpening (and even on things like the scale of initialization)---and in the absence of a full theory of progressive sharpening, we are unable to derive this gap analytically. Or better, we know it at initialization, it scales with 1/b, but we are unable to derive it analytically during training, which does not follow 1/b.
>
> ---
>
> ## Response to W2
>
> Thank you for pointing out that some parts of the paper were hard to read. We agree that several passages, including Section 4, were confusingly worded, and we have revised the manuscript accordingly. In particular, we went through your specific comments one by one and implemented the corresponding textual changes in the new version. We appreciate your effort for going thoroughly through the manuscript pointing out bad wording/typos/spots for improvement.
>
> > However, their original understanding of catapults seem to include the sharpness-recovering behavior in both EoS and EoSS.
> To be honest, we are not sure what you meant by "the sharpness-recovering behavior".
>
> In any case, Section 4 wasn't written well, and we rephrased it in the updated version. In particular, by catapults we mean an event when the dynamics become unstable in the quadratic approximation, and, almost tautologically, the only options after that is (a) the trajectory diverges, or (b) it will become stable again. Now, for the latter to happen, trivially, we have to leave the neighborhood where the quadratic approximation applies (because in the quadratic we were not stable). Now, to not diverge, the trajectory has to re-stabilize in a stable region, where the sharpness might be low, which causes for the progressive sharpening to restart.
>
> These are the ideas we had in mind in the original Section 4, but in trying to keep the section short, we didn’t phrase this well. We believe the re-phrasing explains it better. In particular, see how we redefined Catapults in Appendix B much more rigorously.
>
> Thanks again! We hope this concerns of yours were addressed, please let us know!

---

> ### Author Response · Authors · 2025-11-27
>
> ## Response to Questions
> Below are the one-by-one answers to the points that you raised.
>
> * “Footnotes (3, 4, 7) are made following right after equations, disrupting the reading of the formulas.” - thank you, corrected!
> * "Alphabetical ordering of citations needed” - thank you, fixed
> * "There seems to be a problem with the style file, and the fonts are not compiled as intended” - turned out to be an issue with us importing “lmodern”; thank you for catching!
> * “L014: Cohen et al. (2021) consistently states that sharpness stabilizes above this value.” - good point! (we had = there, which is strictly speaking incorrect); although we would argue that when it comes to the level of stabilization (instead of momentarily) it is not *always* above, so corrected to the more factual “around”, which also captures the Self-Stabilization of Damian et al. and Cohen et. al later Central Flow.
> * "L015: Cohen et al. (2021) explicitly ignores generalization concerns.” - agreed, although in our wording this is a separate sentence, and this rather refers to the other works, so we think it is suitable to keep as is, as we don’t think it implies that Cohen implies that; keeping as-is
> * "L039: Items (2) and (3) are unclear.” - this makes sense; we reworded it!
> * “L046: I think at least a brief mention of the notation should be made before the batch sharpness equation is presented.” - added the missing (non-trivial) notation.; "L046: Similar to? The boxed summary should be mostly self-contained” - agreed, reworded
> * "L052: "implicitly functions as sharpness", "stability for SGD is stability on the mini-batch landscape" sound confusing. I would encourage the authors to slightly expand on these, as their vagueness defeats the purpose of this highlight.” - although we agree this is somewhat vague, yet we think the implications would be clear for readers familiar with EoS literature, so decided to keep it like that
> * “L063: $\eta > 0$” - we are putting it explicitly there to indicate a difference from gradient flow, we agree that it is somewhat excessive though
> * $\tilde L$ - explained the notation
> * “L093: First couple of sentences make me think that another paragraph with the title e.g. "Learning rate and gradient-based optimization" can be used. As Jastrzebski et al. 2020 is about SGD, for example.” - the main point of the paragraph is about full-batch EoS, so we think the name is suitable, so keeping as is; the reason to bring up Jastrzebski is that although that work is primarily about SGD, they also discover the full-batch EoS, and the stabilization at 2/eta
> * “L115: How is this paragraph title different than L091?” - good point, rephrased to make it clearer
> * "L121: I feel this lengthy quote from Cohen et al. is unnecessary.” - we thought this quote was actually important for demonstrating that it was an open problem, we agree that ultimately it is not that necessary; removed
> * “L130: "the answer" to?” - agreed, removed
> * “L186: What does "step size does not vanish" mean? Please explain within the text.” - changed to “is not annealed”
> * "L191: "leaves all the compacts in which..." -> "exits all compact subset of the region in which..."” - yeah, this is more precise, changed
> * “L225: Why not $\mathcal H(L)$…” - good point; it adds clarity, especially compared to Batch Sharpness; changed
> * "L229: "descent lemmas" are unintroduced.” - true; yet, we think this is a classical result in theory of optimization, and is unambiguous; still, we added “classical”, and removed capitalization to clarify
> * “L266: I do not understand this footnote.” - it was indeed confusing; we removed it, as there is now a full Appendix B talking about that
> * "L270: "Importantly, a reason why..." I do not understand this sentence.” - agreed, this wasn’t phrased well. expanded it, and moved this whole paragraph to Section 5.2
> * "L302: What does step sharpness mean?” - we refer to the notation in Figure 6; adding the explicit wording “step sharpness” in Figure 6
> * “L313: Referring to a future figure for notation seems suboptimal:” - agreed about suboptimatily, but the caption unfortunately makes sense in a later figure, and the “size” of that figure forces it to be placed later
> * “L039: Seemingly mistaken use of $L$ instead of $\lambda_{\mathrm{max}}$.” - the way this is written is actually confusing without referencing prior works; we used $L$ as a standard notation in the “$L$-smooth” functions; changing it to $\lambda_{\max}$.
> * “L044: "as $\lambda_{\mathrm{max}}$" -> "such as $\lambda_{\mathrm{max}}$" - thanks, corrected!
> * “L176: "constraints" -> "constrains””, “L181: The word "oscillations" neglected from the title?”, "L215: "Dyanmics"”, “L285: "traind" - all good catches, fixed!
>
> Thanks a lot for raising all these, this was extremely valuable for us!
> Although they do not really conceptually change the paper, we think you really helped improving the presentation.

---

### Official Review · Reviewer_JB7m · 2025-11-05

**Soundness:** 2
**Presentation:** 2
**Contribution:** 3
**Rating:** 4
**Confidence:** 4

**Summary:**

This paper investigates the instability regime of stochastic gradient descent (SGD) in neural network optimization, which has not been sufficiently explained by the existing edge of stability (EoS) framework for gradient descent (GD). The authors distinguish two types of oscillatory behavior during SGD instability: noise-driven and curvature-driven oscillations. The latter, of particular interest, is characterized through batch sharpness, defined as the expectation of directional sharpness of the mini-batch loss along its gradient direction. Experiments demonstrate that, within the edge of stochastic stability (EoSS) regime, batch sharpness consistently hovers around 2/η and provides a potentially better explanation of the catapult effect than gradient-noise interaction (GNI).

**Strengths:**

- The paper addresses a fundamental and underexplored aspect of deep learning optimization—the instability regime of SGD—and makes an original attempt to distinguish between two forms of oscillation. The proposed concepts of curvature-driven oscillation and batch sharpness are interesting and potentially valuable for understanding SGD dynamics.
- Extensive experiments on CIFAR-10 provide empirical evidence that batch sharpness remains close to 2/η, supporting the proposed hypothesis.

**Weaknesses:**

- The Introduction and Background sections are disproportionately long (occupying over one-third of the main text), while the intuitive motivation or justification for the definition of batch sharpness, as well as its theoretical connection to the catapult effect, are insufficiently developed. Reducing the background in favor of more focused explanations on these points would improve the paper’s overall clarity and intuition.
- The current definition of the catapult effect is somewhat ambiguous. It is difficult to interpret what exactly constitutes the catapult phenomenon or its observable behavior from the experimental figures. A clearer and more formal definition, along with an analysis of its relationship to batch sharpness and GNI, is needed.

**Questions:**

- The paper provides a theoretical justification (Theorem 1) showing that when batch sharpness exceeds 2/η, the expected batch gradient squared norm can locally increase exponentially. However, the intuitive motivation for defining batch sharpness as the average of directional sharpness values is not clearly articulated. Why should the average—rather than other statistics such as the maximum or the proportion exceeding 2/η—be considered the most meaningful indicator of instability? Some discussion on this intuition would help readers better understand the rationale behind the definition.
- In Figure 4, batch sharpness drops sharply at several points, yet no explanation is provided. What mechanism causes these abrupt decreases?

---

> ### Author Response · Authors · 2025-11-27
>
> We thank the reviewer for appreciating the effort, for the detailed feedback and helpful suggestions. Below we address the two main weaknesses and then respond to the specific questions.
>
> ---
>
> ## W1 — Length and focus of Introduction / Background
>
>
> * The *Introduction* itself is only about one page, and we believe this is a reasonable length for setting up the problem and contributions.
> * We agree that the *Background / Related Work* section is relatively long. This is partly because we use it not only to review prior work but also to introduce and contextualize the specific problem we working on.
> * Following your suggestion, we shortened the related work by removing a long quote and tightening the discussion.
> * We would also be open to remove the paragraph about generalization, since our work is explicitly not about generalization, and only flatness comes up. Yet, we still think it is important to mention, as it was historically important for the community.
>
> ## W1 -- Intuition for Batch Sharpness
> > while the intuitive motivation or justification for the definition of batch sharpness is insufficiently developed
>
> Concerning your question about the intuitive motivation/justification for Batch Sharpness, we had one in the “Relation to earlier notions”. We expanded this discussion, and moved it to Section 5.2. We added an Appendix B and reframed Section 5 in a way to completely address this weakness. Please let us know what you think and if this completely addresses your concern.
>
>
> ---
>
> ## W2 -- Definition and role of the catapult effect
>
> You brought up multiple issues with the catapult effect (together with Figure 4, addressed below)
> > The current definition of the catapult effect is somewhat ambiguous. It is difficult to interpret what exactly constitutes the catapult phenomenon or its observable behavior from the experimental figures. A clearer and more formal definition, along with an analysis of its relationship to batch sharpness and GNI, is needed.
> > ...as well as its theoretical connection to the catapult effect
>
> We agree that our original exposition did not explain catapults clearly enough. We have revised the manuscript to address this at both the intuitive and formal levels:
>
> Concerning your request for theoretical connection to the catapult effect, we added explanation in Section 3.3 that connects Type-2 oscillations, perturbations and catapults. Importantly, Section 3.3 summarizes the rigorous formalization added in Appendix B and H. We formalize there what catapults are and that catapults on the 2nd-order Taylor approximation are analogous to having an instability criterion saturated.
> Let us know if this enough to clarify how we think of catapults and why they are related to EoSS.
>
> To summarize what we are saying in those parts:
> * Along the lines of our response to Reviewer mnHT, by catapults we mean an event when the dynamics become unstable in the quadratic approximation, and, almost tautologically, the only options after that is (a) the trajectory diverges, or (b) it will become stable again. Now, for the latter to happen, trivially, we have to leave the neighborhood where the quadratic approximation applies (because on the quadratic we were not stable). Now, this might cause the dynamics to re-stabilize at a stable region, where the sharpness might be low, which causes for the progressive sharpening to restart.
> * Now, the question is how to detect these catapults, and for this we added Appendix B and Appendix H to rigorously address these points in the revised version of our manuscript. In particular, we show that we have a catapult, if and only if we have an instability criterion (as Batch Sharpness) saturating, if and only if the loss has sizeable spikes.
> * Also, we re-worded the part about “Catapults” in Section 4 too, better explaining the phenomenon in SGD training. We also changed the caption in Figure 4, we hope it makes it clear why the sharpness drops, and then recovers - thus answering your Question #2 about Figure 4, this is a manifestation of catapults happening. The fact that Batch Sharpness drops just means that the region where dynamics stabilized has low batch sharpness, and we keep having the: instability -> loss spike -> transition to a flatter region -> renewed sharpening

---

> ### Author Response · Authors · 2025-11-27
>
> ## Q1 — Intuitive motivation for defining *batch sharpness* as an average
>
> **Comment:** Why is batch sharpness defined as the *average* of directional sharpness values, rather than, e.g., the maximum or the proportion exceeding (2/\eta)? What makes the average the meaningful indicator of instability?
>
> Thanks for the question! There are mutliple reasons why it has to be the average, as we explain in the updated Section 5.2, but will also expand on them here:
> * First, note that even \(\lambda_{\max}(\text{Hessian})\) can be viewed as the (time-)averaged directional sharpness along the gradient direction: once it crosses \(2/\eta\), the gradients align with the eigenvector corresponding to the largest eigenvalue. From this point of view, *Batch Sharpness* is a natural generalization to the case where the stochasticity of SGD prevents the steps from perfectly aligning. In this setting, the time average is replaced by an average over mini-batches, since at every iteration we observe a fresh batch.
> * More generally, over many steps, if an instability criterion has bounded variance, then when its mean is above the threshold, divergence occurs with high probability, and when its mean is below the threshold, convergence occurs with high probability. This suggests that the relevant quantity should be some form of average.
> * From a more technical perspective, the expectation naturally appears because, classically, one often takes as a Lyapunov function an expectation of Lyapunov functions (and even in full-batch, the full-batch Hessian is the expectation of the per-batch Hessians).
> * To sum up, SGD is a stochastic dynamical system. When the behavior of interest is time-averaged or mean behavior (e.g., whether a time average diverges), when you take Taylor expansions you keep having the expectation.
> * Most importantly, our goal is to describe an empirical phenomenon: our mathematics is built to capture what we observe, namely that the *expected* alignment between Hessians and gradients stabilizes over time at \(2/\eta\). This is not true, for instance, for (\lambda_{\max}^b = \mathbb{E}*B[ \lambda*{\max}(H(L_B))]), as we show in Appendix M.2.
>
> ## Q2
> We mentioned this before. However, let us add that in those cases we believe SGD is sampling a number of outlying batches that catapult the dynamics.
>
> ## Concluding
> We completely agree with you that we could have done a better job in the treatment of catapults and in explaining what batch sharpness actually is. We believe we may have turned too formal now, e.g., in Appendix B or H, but this was necessary to properly clarify these points.
> Please do tell us if you think this completely addresses your concern and whether there is something else we can do for you to raise your score.
>
> Thanks again for your review.
>
> The authors

---

### Author Response · Authors · 2025-11-28
**TLDR for Area Chair**

Dear Area Chair,

For your ease, we would like to write a TLDR on the state of rebuttals so far, feel free to discard it.

### **Main weaknesses of this work**:
The most shared weaknesses that reviewers pointed out were, imo:
- **(1)** the manuscript was hard to read and the math informal at times,
- **(2)** that the concept of catapults seemed confusing, and
- **(3)** no clear intuition of the meaning of Batch Sharpness and how that relates to $\lambda_{\max}$.

We are sorry for that and:
- **(1)** We heavily cleaned the text and we made every proof and statement more rigorous, e.g., we added the formal version of the results in the appendix (the proofs were already rigorous of course). We also changed the statement of Theorem 1 and we found a proof that is more direct, as the previous lead Reviewer 4 (d8mW) to misunderstandings.
- **(2)** We added Appendix B and H which now rigorously introduce a framework for instability for stochastic dynamical systems, derive an equivalence between instability and catapults on the quadratic approximation of the landscape. This formalizes the concept of catapults and show why our approach was correct and our questions were valid.
- **(3)**  We then added Section 5.2 on the meanings and intuitions of Batch Sharpness, which explains how EoSS generalizes EoS and how Batch Sharpness generalizes $\lambda_{\max}$. Critically, one needs to understand what stability is to understand in what sense the quantity generalize, their generalization is not about oscillations of Hessians per se, but about the instability perspective.

This pretty much summarizes the comments from the first three reviewers.

#### **Reviewer 4:**

We are sorry that we could not engage in discussion with Reviewer 4 (d8mW) because of the events happening. We do not want to bias your opinions but we believe that the poor soundness, fair contribution, and reject grade was due to a misunderstanding of Theorem 1 and of the conceptual steps in its proof and the misunderstanding of the whole point of our paper, which is that: **those in the previous literature were correct approaches to instability of SGD, but (see Appendix B) we show there are actually infinitely possible approaches, not only that one. We search empirically for which one is *the one* that captures neural networks training**.
More so, with the interpretation of our work as outlined by Reviewer 4 (with the assumptions that we relied solely on the descent lemma, and the same batch repeated etc) we would agree with the reject rating. However, as we detailed in our rebuttal, that interpretation does not accurately reflect our work.
While we completely understand that this was probably due to lack of clarity of our first draft—and we are sorry for that—we believe that bad review does not really assess our theoretical and empirical efforts, and rather assesses the insufficient quality of our initial writing which did not allow Reviewer 4 to understand what the paper was about.

Thanks a lot for your AC work,

We remain at your disposal for any comment,

The authors

---

### Meta-Review · Area_Chair_hReq · 2025-12-15

**Summary:**

The reviewers indicated the following concerns in their reviews:
- The paper is not well organized as the introduction and background sections are too long
- Current definition of the catapult effect is somewhat ambiguous
- Batch sharpness as the average of directional sharpness values is not clearly articulated
- A range of issues from conceptual conflicts to more mundane errors
- An authoritative explanation why SGD does not follow EoS is missing
- The presentation is poor as there are various typos
- The theoretical analysis is not rigorous as the authors use $\approx$ in many places
- The claim of edge-of-stability (EoS) phenomenon to SGD is not correct
- It is not clear whether the proposed batch sharpness measure is of practical or theoretical interest
- The proposed measure of batch sharpness is very similar to the terms in existing studies, which limits the novelty
- The presented results are weaker than those of published literature
- The paper lacks clarity and rigor

**Reviewer Concerns:**

After reading the authors' response, I think the following concerns are still outstanding
- The theoretical analysis is not rigorous as the authors use $\approx$ in many places. The paper lacks clarity and rigor. (the authors agree that the initial submission lacks rigor and clarity. In the revision, the authors have made a lot of changes to fix the mistakes. However, this significant change makes another full round of review necessary before the submission gets accepted. It remains unclear whether the authors make all the correct changes)
- The proposed measure of batch sharpness is very similar to the terms in existing studies, which limits the novelty (While the authors added some discussions regarding the difference, it is not quite clear key difference of their batch sharpness to the terms in the existing studies)

I think the following concerns may be well addressed
- The paper is not well organized as the introduction and background sections are too long
- Current definition of the catapult effect is somewhat ambiguous
- Batch sharpness as the average of directional sharpness values is not clearly articulated
- A range of issues from conceptual conflicts to more mundane errors
- An authoritative explanation why SGD does not follow EoS is missing
- The presentation is poor as there are various typos

**Reviewer Scores:**

Reviewer JB7m and reviewer mnHT are unlikely to change their scores as both reviewers identified several issues on the correctness of the analysis. While the authors made a major change in the rebuttal phase, this significant change makes another full-round of review process necessary.

Reviewer mnHT is unlikely to change his/her score as the current score is already high ( 6: marginally above the acceptance threshold). This reviewer also spotted various typos and errors in the presentation.

Reviewer JB7m may change his/her score as his comments on the organization and ambiguous definition are well addressed.

---

### Decision · Program_Chairs · 2026-01-26

Reject